# Supraglacial debris-cover changes in the Greater Caucasus from 1986 to 2014

Levan G. Tielidze[1,6,7], Tobias Bolch[2], Roger D. Wheate[3], Stanislav S. Kutuzov[4], Ivan I. Lavrentiev[4], Michael Zemp[5]

[1]Department of Geomorphology, Vakhushti Bagrationi Institute of Geography, Ivane Javakhishvili Tbilisi State University, 6 Tamarashvili st., 0177, Tbilisi, Georgia

[2]School of Geography and Sustainable Development, University of St Andrews, North Street, Irvine Building, St Andrews, KY16 9AL, Scotland, UK

[3]Natural Resources and Environmental Studies, University of Northern British Columbia, 3333 University Way, Prince George, V2N 4Z9, BC, Canada

[4]Department of Glaciology, Institute of Geography of the Russian Academy of Sciences, 29 Staromonetniy Pereulok, 119017, Moscow, Russia

[5]Department of Geography, University of Zurich, 190 Winterthurerstrasse, CH-8057, Zurich Switzerland

[6]Antarctic Research Centre, Victoria University of Wellington, PO Box 600, 6140, Wellington, New Zealand

[7]School of Geography Environment and Earth Sciences, Victoria University of Wellington, PO Box 600, 6140, Wellington, New Zealand

*Correspondence to:* Levan G. Tielidze (levan.tielidze@tsu.ge)

## Abstract

Knowledge of supraglacial debris-cover and its changes remain incomplete in the Greater Caucasus, in spite of recent glacier studies. Here we present data of supraglacial debris-cover for 659 glaciers across the Greater Caucasus based on Landsat and SPOT images from the years 1986, 2000, and 2014. We combined semi-automated methods for mapping the clean-ice with manual digitization of debris-covered glacier parts and calculated supraglacial debris-cover area as the residual between these two maps. The accuracy of the results were assessed by using high resolution Google Earth imagery and GPS data for selected glaciers. From 1986 to 2014, the total glacier area decreased from $691.5\pm29.0$ km$^2$ to $590.0\pm25.8$ km$^2$ ($15.8\pm4.1\%$ or $\sim0.52\%$ yr$^{-1}$) while the clean-ice area reduced from $643.2\pm25.9$ km$^2$ to $511.0\pm20.9$ km$^2$ ($20.1\pm4.0\%$ or $\sim0.73\%$ yr$^{-1}$). In contrast supraglacial debris-cover increased from $7.0\pm6.4\%$ or $48.3\pm3.1$ km$^2$ in 1986 to $13.4\pm6.2\%$ ($\sim0.22\%$ yr$^{-1}$) or $79.0\pm4.9$ km$^2$ in 2014. Debris-free glaciers exhibited higher area and length reductions than debris-covered glaciers. The distribution of the supraglacial debris-cover differs between northern and southern and between western, central and eastern Greater Caucasus. The observed increase of supraglacial debris-cover is significantly stronger in the northern slopes. Overall, we have observed up-glacier average migration of supraglacial debris-cover from about 3015 m to 3130 m above sea level (asl) during the investigated period.

## 1 Introduction

Supraglacial debris-cover affects surface melt with increasing ablation in cases of thin debris-cover (< a few cm), or decreasing ablation under continuous thick debris-cover (Östrem, 1959; Nicholson et al., 2018). Obtaining information about debris-cover is relevant not only with respect to its impact on glacier ablation but also because it is an important part of the sediment transport system (supraglacial, englacial, and subglacial) in cold and high mountains (Kellerer-Pirklbauer, 2008), which ultimately affect the overall dynamics, and mass balance of the glaciers. Several studies show an increase in debris-covered area with overall glacier shrinkage and mass loss (Deline, 2005; Stokes et al., 2007; Kirkbride and Deline; 2013; Glasser et al., 2016).

For regions where the local population is dependent on glacial meltwater supply, detailed knowledge of glacial hydrology is important to ensure the sustainable use of water resources (Baraer et al., 2012). One difficulty of such investigations is associated with limited knowledge of the large-scale extent, thickness, and properties of the supraglacial debris-cover. Field measurements of debris layers have practical difficulties on a large scale, and methods for estimating supraglacial debris thickness using remote sensing and modelling remain in development (Zhang et al., 2016; Rounce et al., 2018). Several studies have also reported the role of debris-cover in promoting the formation of supraglacial lakes (Thompson et al., 2016; Jiang et al., 2018), which are directly related to glacial hazards (Benn et al., 2012). Therefore, it is necessary to take supraglacial debris-cover into account when assessing temporal change of mountain glaciers.

Ice and snow melt in the Greater Caucasus are major sources of runoff for populated places in many parts of the Caucasus region (Tielidze, 2017). Previous studies have also shown that in this region, supraglacial-debris cover is an important control for ice ablation (Lambrecht et al., 2011), and a component in glacier mass balance (Popovnin and Rozova, 2002). Thus, correct delineation of supraglacial debris-cover in the Greater Caucasus is vital to correctly model future glacier development. A recent global study (Scherler et al., 2018) concluded that supraglacial debris-cover is abundant in the Caucasus and Middle East (more than 25% of glacier area) and that this region shows the highest percent of supraglacial debris-cover worldwide. However, earlier studies indicated lower relative supraglacial debris-cover than our study in the Greater Caucasus but are restricted to smaller regions (Stokes et al., 2007) or individual glaciers (Lambrecht et al., 2011; Popovnin et al., 2015). Scherler et al. (2018) used the glacier outlines from the Randolph Glacier Inventory (RGI) v6 database (RGI consortium, 2017) with some geolocation errors and nominal glaciers, representing glacier area by an ellipse and included in RGI to achieve global coverage in case no other information were available (Pfeffer et al., 2014). Hence, there is a clear need to provide an improved estimate of supraglacial debris-cover and its changes for this region.

Based on a recently published glacier inventory (Tielidze and Wheate, 2018), we present the first regional assessment of the spatial distribution of supraglacial debris-cover and related glacier changes between 1986, 2000 and 2014 for the Greater Caucasus.

## 2 Study area

The Greater Caucasus is one of the world's highest mountain systems, and the major mountain unit of the Caucasus region containing about 96% of contemporary glacier area of the Caucasus and Middle East glacier region according to RGI (Pfeffer et al., 2014). The range stretches for about 1300 km from west-northwest to east-southeast, between the Taman Peninsula of the Black Sea and the Absheron Peninsula of the Caspian Sea. Using morphological and morphometric characteristics, the Greater Caucasus can be divided into three parts - western, central and eastern. Moreover, the Greater Caucasus can also be subdivided the northern and southern part according to the location relative to the main crest of the range (Fig. 1). The central Greater Caucasus is the highest part of the main range represented by the following summits exceeding 5000 m asl: Dykh-Tau - 5205 m, Shkhara - 5203 m, Jangha - 5058 m, and Pushkin Peak - 5034 m. The western and eastern sections are relatively lower with highest summits of Mt. Dombai-ulgen (4046 m) and Mt. Bazardüzü (4466 m) respectively. Contemporary glaciers are concentrated mostly along the main mountain range, as well as in the sub-ranges of the Greater Caucasus and separate massifs such as Elbrus and Kazbegi-Jimara (Fig. 1). Elbrus is a separate dormant volcanic mountain on the border between western and central Greater Caucasus about 10 km north from the main watershed divide. It is the highest summit of the region with two peaks - western (5642 m) and eastern (5621 m).

According to the recent inventory (Tielidze and Wheate, 2018), this mountain range contains over 2000 glaciers with a total area of about 1200 km$^2$. The northern slopes of the Greater Caucasus contain more glaciers than the southern slopes. The altitude of the glacier equilibrium line (ELA), increases from 2500–2700 m in the west to 3700-3950 m in the eastern sector of the northern slope of the Greater Caucasus (Mikhalenko et al., 2015). The ELA was

1  determined to range from ~3030 m in the west to ~3480 m in the eastern section of the southern slope of the Greater
2  Caucasus (Tielidze, 2016). The ELA is ~1000 m higher on the northern slopes of the Elbrus than the southern slopes
3  of the central Greater Caucasus (Mikhalenko et al., 2015).

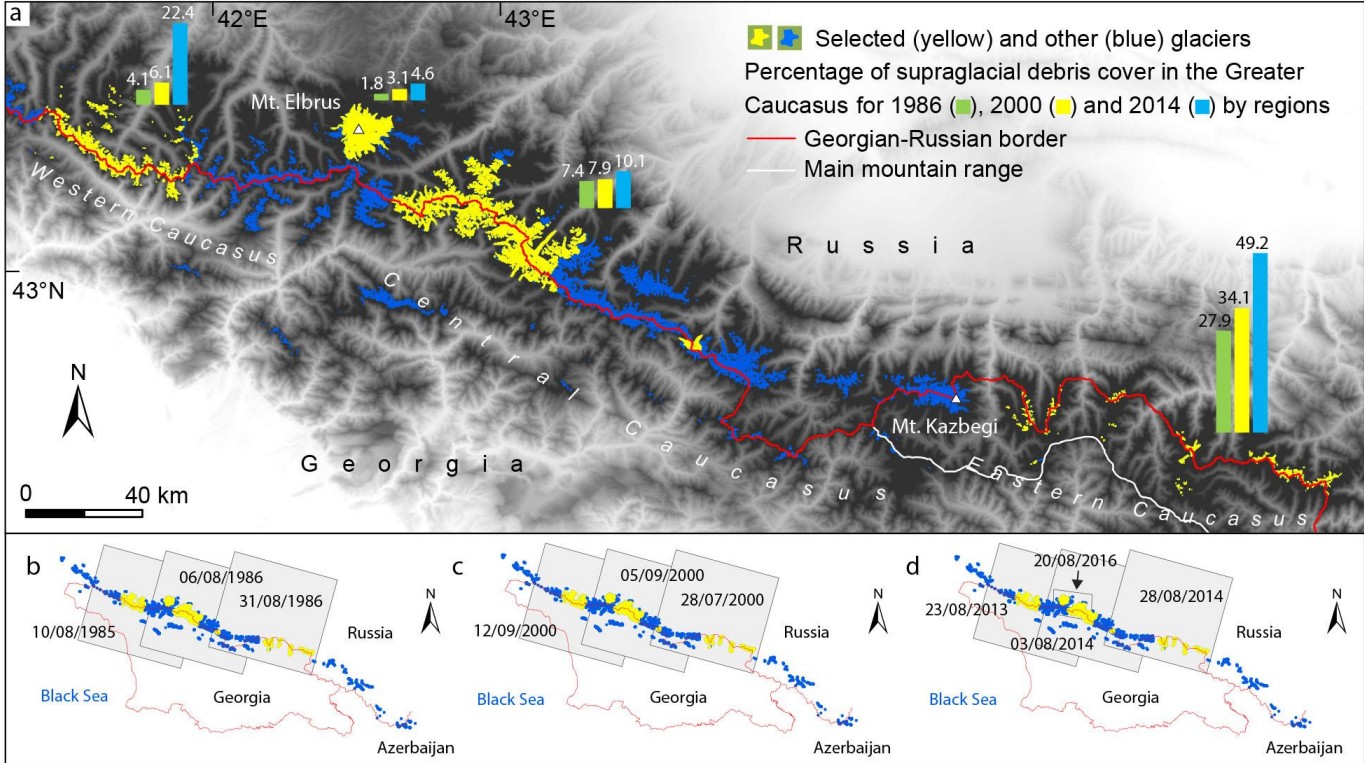

**Figure 1.** a – Investigated area and selected glaciers by regions. b – Three Landsat 5 TM satellite scenes 1985-1986. c – Three Landsat 7 ETM+ satellite scenes from 2000. d – Three Landsat 8 OLI satellite scenes from 2013-2014 and one (smaller) SPOT satellite scene from 2016.

10  We used the Glacier Classification Guidance from the Global Land Ice Measurements from Space (GLIMS)
11  initiative for remote sensing observations (Rau et al., 2005) to define debris-free and debris-covered glaciers.
12  According to this guideline we consider all glaciers with a debris-coverage of more than 10% as debris-covered and
13  otherwise as debris-free glaciers.
14  As the Greater Caucasus range forms the boundary between temperate and subtropical climatic zones, the
15  orientation and height of the range determines the contrasts between the northern and southern slopes. The mean
16  annual temperatures at the northern slopes are usually 1-2°C cooler than those in the south (Tielidze and Wheate,
17  2018). The average regional lapse rate is minimum in winter (-2.3°C per 1000 m) and maximum (-5.2°C per 1000 m)
18  in summer (Kozachek et al., 2016).
19  Annual precipitation varies between 2000-2500 mm in the west and declines to 800-1150 mm in the east on the
20  northern slope. The central section of the southern slope receives over 2000 mm of precipitation while in the east, the
21  annual total is 1000 mm. The south-western section of the region is humid with annual precipitation about 3200 mm
22  (Volodicheva, 2002; Mikhalenko et al., 2015). The south-western slopes of the Greater Caucasus experience heavy
23  snowfall and snow avalanches (from November to April). Snow cover may reach 5-7 meters in several regions of the
24  western part of the Greater Caucasus, such as the Upper Svaneti and northern Abkhazeti regions in Georgia (Sylvén
25  et al., 2008).
26
27

## 3 Data and methods

### 3.1 Datasets

We selected 659 glaciers with a total area of 590.0±25.8 km$^2$ (based on the "Greater Caucasus Glacier Inventory" and representing the year 2014 (Tielidze and Wheate, 2018)): 223 glaciers in the western Greater Caucasus (145 - northern slope, 78 - southern slope); 285 in the central Greater Caucasus (173/112); and 130 in the eastern Greater Caucasus (130/0). In addition, all 21 glaciers on Elbrus (5 - northern slope, 8 - southern slope, 5 - western slope, 3 - eastern slope) - the largest glacierised massif in the whole region - were selected (Fig. 1a). Overall, this equals 49.5% and 32.6% of the Greater Caucasus total glacier area and number respectively. The size of the selected glaciers varied between 37.5 km$^2$ and 0.01 km$^2$.

A total of nine Landsat images were used in this study (Fig. 1 b-d; Table 1), downloaded from the Earthexplorer website (http://earthexplorer.usgs.gov/). We also used an orthorectified high resolution (spatial resolution 1.5 m) SPOT-7 satellite image from 2016, and the Shuttle Radar Topography Mission Digital Elevation Model (SRTM DEM 30 m) (Table 1). The Landsat scenes served as a basis for supraglacial debris-cover assessment while the SPOT image was used for corrections of supraglacial debris-cover areas of Elbrus. All imagery was captured between 28[th] July and 12[th] September, when glacier tongues were mostly free of seasonal snow under cloud-free conditions.

The Advanced Spaceborne Thermal Emission and Reflection Radiometer (ASTER) Global Digital Elevation Model (GDEM, 30 m), version 2 (http://asterweb.jpl.nasa.gov/gdem.asp) was used to measure mean upper limit of the supraglacial debris-cover, and to calculate supraglacial debris-cover by 500 m elevation bands. We used these elevation bands to intersect our digitized debris-covered areas for 1985/86 to 2013/14, with the total area per elevation band. Further datasets used in this study include high resolution images from Google Earth, and GPS measurements.

**Table 1.** Satellite images and digital elevation models used in this study.

| Date | UTM zone | Sensor | Region/Section | Resolution | Scene ID |
|---|---|---|---|---|---|
| 10/08/1985 | 37N | Landsat 5 TM | Western Greater Caucasus | 30 m | LT51720301985222XXX04 |
| 06/08/1986 | 38N | Landsat 5 TM | Central Greater Caucasus | 30 m | LT51710301986218XXX02 |
| 31/08/1986 | 38N | Landsat 5 TM | Eastern Greater Caucasus | 30 m | LT51700301986243XXX03 |
| 12/09/2000 | 37N | Landsat 7 ETM+ | Western Greater Caucasus | 15/30 m | LE71720302000256SGS00 |
| 05/09/2000 | 38N | Landsat 7 ETM+ | Central Greater Caucasus | 15/30 m | LE71710302000249SGS00 |
| 28/07/2000 | 38N | Landsat 7 ETM+ | Eastern Greater Caucasus | 15/30 m | LE17003020000728SGS00 |
| 23/08/2013 | 37N | Landsat 8 OLI | Western Greater Caucasus | 15/30 m | LC81720302013235LGN00 |
| 03/08/2014 | 38N | Landsat 8 OLI | Central Greater Caucasus | 15/30 m | LC81710302014215LGN00 |
| 28/08/2014 | 38N | Landsat 8 OLI | Eastern Greater Caucasus | 15/30 m | LC81700302014240LGN00 |
| 20/08/2016 | 37N | SPOT-7 | Elbrus | 1.5 m | DS_SPOT7201608200751063 |
| 17/11/2011 | 37N | ASTER GDEM | Western Greater Caucasus | 30 m | ASTGTM2_N43E041/E040_DEM |
| 17/11/2011 | 38N | ASTER GDEM | Central Greater Caucasus | 30 m | ASTGTM2_N43E042/E043_DEM |
| 17/11/2011 | 38N | ASTER GDEM | Eastern Greater Caucasus | 30 m | ASTGTM2_N42E045/E046_DEM |
| 23/09/2014 | 37N | SRTM DEM | Elbrus | 30 m | SRTM1N43E042V3_DEM |

### 3.2 Methods

The widely used band ratio segmentation method (RED/SWIR with a threshold of $\geq$ 2.0) was used as the first step in delineating clean-ice outlines for 2014 (Bolch et al., 2010; Paul et al., 2013) followed by intensive manual improvements (removing misclassified areas, e.g. due to snow and shadows). In the next step supraglacial debris-cover was classified as the residual between these semi-automatically derived clean-ice outlines and the existing manually mapped outlines which include the debris-covered parts (cf. Tielidze and Wheate, 2018) (Fig. 2a). To assess temporal change, we calculated the area of supraglacial debris-cover for individual glaciers in a similar way for the earlier years 1986 and 2000.

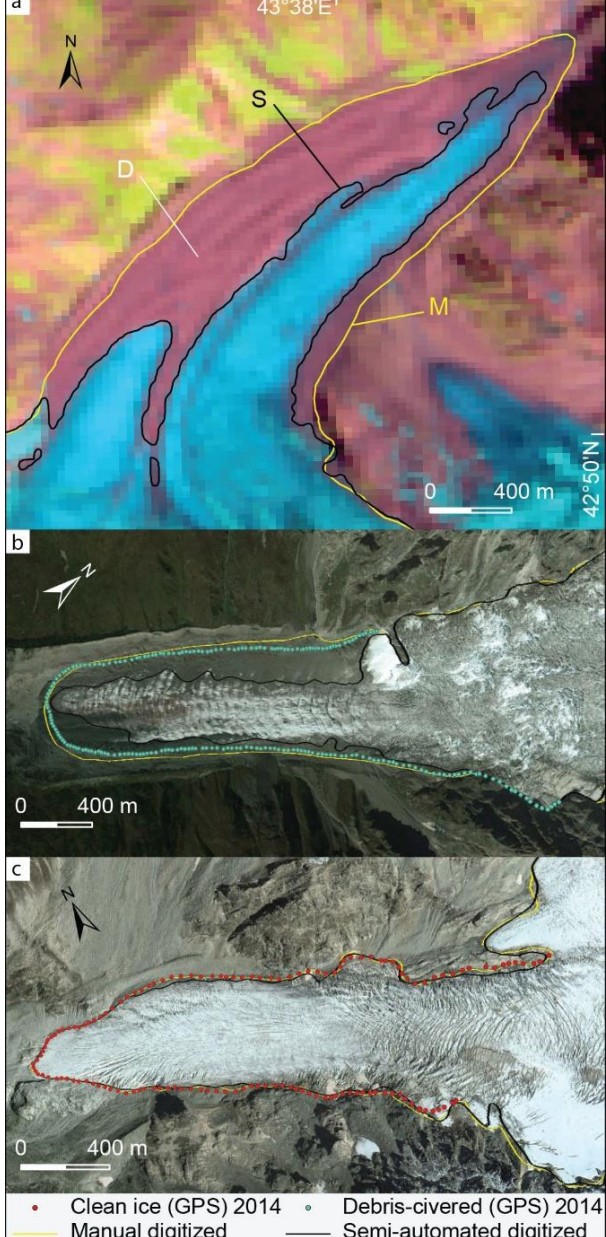

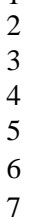

| | |
|---|---|
| • Clean ice (GPS) 2014 | ○ Debris-civered (GPS) 2014 |
| — Manual digitized | — Semi-automated digitized |

**Figure 2.** Mapping examples: a – supraglacial debris-cover (D) assessment with comparison of different methods: fully manual mapping (M) and semi-automated mapping (S) (ratio TM 3/5 threshold ≥ 2.0 followed by manual improvement). The Landsat image from 06/08/1986 is shown in the background. Examples of glacier outline accuracy assessment by GPS measurements: b – Adishi Glacier; c – Kirtisho Glacier. A Google Earth image from 19/09/2011 is shown in the background.

The buffer method (Granshaw and Fountain, 2006; Bolch et al., 2010) was used for uncertainty estimation for both clean-ice and debris-covered glacier parts. For clean-ice we used a 15 m (1/2 pixel) buffer (Bolch et al., 2010) and for debris-covered parts 60 m (two pixels) (Frey et al., 2012). We used the standard deviation of the uncertainty distribution for the estimate following Mölg et al. (2018) as a normal distribution can be assumed for this type of mapping error. This method is applied to glacier complexes excluding overlapping areas, as well as the boundary between clean- and debris-covered ice of the same glacier. The calculated uncertainty for the clean-ice/debris-covered parts was on average 4.0%/6.4% for 1986, 4.1%/6.3% for 2000, and 4.1%/6.2% for 2014. The uncertainty estimates for all Caucasus glaciers are described in previous studies (Tielidze, 2016; Tielidze and Wheate, 2018).

As an independent assessment of these uncertainty estimates, three randomly selected glacier outlines from Landsat 8 (03/08/14) (including clean-ice and debris-covered parts) were imported into Google Earth and manually adjusted using the available high-resolution Quickbird images (19/09/11) superimposed upon the SRTM3 topography (Raup, et al., 2014). These glacier outlines were then compared with original outlines from Landsat 8 image (03/08/14). The area differences between the two resulting sets of outlines were ±5.2% for supraglacial debris-cover and ±3.4% for clean-ice which confirms our uncertainty estimate based on the buffer method. For Elbrus the high-resolution SPOT-7 imagery was used to estimate an extra uncertainty of the clean-ice and debris-covered area. The area difference between SPOT-7 and Landsat 8 outlines was ±7.4% for supraglacial debris-cover and ±4.1% for clean-ice.

An additional uncertainty assessment was performed using GPS (Garmin 62stc) measurements of glacier margins (>1200 points) obtained during field investigations in 2014. In total seven glaciers (Ushba, Chalaati, Lekhziri, Adishi, Shkhara, Zopkhito, Kirtisho) were surveyed. The horizontal accuracy of these measurements varied from ±4 to ±10 m, Fig. 2b-c shows the results of comparison between GPS measurements and Landsat based supraglacial debris-cover/clean-ice outlines. The average accuracy based on all seven glacier measurements was ±30 m for supraglacial debris cover and ±15 m for clean-ice, hence again confirming the suitability of the selected buffer method.

## 4 Results

We found an absolute increase of supraglacial debris-cover for all investigated glaciers from 48.3±3.1 km$^2$ in 1986, to 54.6±3.4 km$^2$ in 2000 and 79.0±4.9 km$^2$ in 2014, in contrast to a reduction of the total glacier area. This equates to a total increase in the proportion of supraglacial debris-cover surface area from 7.0±6.4% in 1986, to 9.1±6.3% in 2000, and to 13.4±6.2% in 2014 (Table 2; Fig. 3). Supraglacial debris-cover was highest in the glacier area classes 1.0-5.0 km$^2$ and 5.0-10.0 km$^2$ for both northern and southern slopes (Fig. S1). The number of debris-covered glaciers (i.e. with an areal percentage of debris larger than 10% - see methods) also increased from 122 in 1986, to 143 in 2000, and to 172 in 2014.

For all regions investigated in the Greater Caucasus the rate of increase in supraglacial debris-cover varied between northern and southern slopes, and between the western, central and eastern sections of the mountain range (Table 2; Fig. 3, 4). The western Greater Caucasus experienced a supraglacial debris-cover area increase from 4.1±6.8% in 1986, to 22.4±6.4% in 2014, while the increase was significantly lower in the central Greater Caucasus over the last 30 years - from 7.4±6.3% to 10.1±6.2% (Table 2). The eastern Greater Caucasus with fewer glaciers almost doubled in supraglacial debris cover over the last 30 years from 27.9±6.2% to 49.2±5.7%.

Supraglacial debris-covered area increased from 13.5±6.3% to 29.3±6.0% for all three sections (western, central and eastern) on the northern slope (not including Elbrus), and from 4.0±6.8% to 9.2±6.9% on the southern slope of the Greater Caucasus between 1986 and 2014.

The Elbrus Massif contained the least percentage of supraglacial debris-cover in all our study regions, but it more than doubled between 1986 and 2014 (from 1.8±6.9% to 4.6±6.6%). Supraglacial debris-cover percentage increase and total glacier area decrease were both highest on the eastern slope of the Elbrus between 1986 and 2014, and lowest on the western slope (Fig. 5a-c).

Hypsometric profiles show that supraglacial debris-cover is most commonly found in the 2500-3000 m zone for Elbrus and the 1900-2500 m zone for the other regions (Fig. 6). The supraglacial debris-cover doubled from 6.4% to 12.2% (0.21% yr$^{-1}$) in the 3000-3500 m zone for all selected glaciers in 1986-2014 (Fig. 6d) and increased in the 3500-4000 m zone for all regions and selected glaciers during the investigated period.

1  **Table 2.** Supraglacial debris-cover and clean-ice in the Greater Caucasus for 1985/86, 2000 and 2013/14 by regions and slopes. The error values are derived by a
2  buffer approach.

| Section and river basin | Selected glacier number | Landsat 5 TM, 1985/86 | | | | | Landsat 7 ETM+, 2000 | | | | | Landsat 8 OLI, 2013/14. SPOT-7, 2016 | | | | |
|---|---|---|---|---|---|---|---|---|---|---|---|---|---|---|---|---|
| | | Total glacier area km² | Clean-ice area km² | Debris-covered area | | | Total glacier area km² | Clean-ice area km² | Debris-covered area | | | Total glacier area km² | Clean-ice area km² | Debris-covered area | | |
| | | | | Glacier number | Area km² | %* | | | Glacier number | Area km² | %* | | | Glacier number | Area km² | %* |
| **Western Caucasus** | | | | | | | | | | | | | | | | |
| Northern slope (Kuban) | 145 | 91.7±3.4 | 87.1±3.1 | 15 | 4.6±0.31 | 5.0±6.7 | 87.2±3.4 | 80.8±3.0 | 21 | 6.2±0.41 | 7.1±6.6 | 78.3±3.4 | 57.9±2.1 | 33 | 20.4±1.3 | 26.1±6.4 |
| Southern slope (Kodori) | 78 | 35.5±1.7 | 34.8±1.6 | 1 | 0.7±0.05 | 2.0±7.1 | 32.8±1.6 | 31.7±1.5 | 1 | 1.1±0.078 | 3.5±7.1 | 26.1±1.3 | 23.1±1.1 | 3 | 3.0±0.21 | 11.5±7.1 |
| **Sum** | **223** | **127.2±5.1** | **121.9±4.7** | **16** | **5.3±0.36** | **4.1±6.8** | **119.8±5.0** | **112.5±4.5** | **22** | **7.3±0.48** | **6.1±6.6** | **104.4±4.4** | **81.0±3.2** | **36** | **23.4±1.5** | **22.4±6.4** |
| **Central Caucasus** | | | | | | | | | | | | | | | | |
| Northern slope (Baksan, Chegem, Cherek) | 173 | 211.0±8.6 | 194.7±7.6 | 28 | 16.3±1.0 | 7.7±6.1 | 203.2±8.6 | 184.5±7.5 | 37 | 18.7±1.1 | 9.2±6.1 | 185.3±8.3 | 161.9±6.9 | 42 | 23.4±1.4 | 12.6±6.0 |
| Southern slope (Enguri) | 112 | 178.8±7.4 | 168.1±6.7 | 15 | 10.7±0.69 | 6.0±6.5 | 171.3±7.3 | 160.5±6.6 | 15 | 10.8±0.69 | 6.3±6.4 | 149.8±6.6 | 139.4±5.9 | 17 | 10.4±0.70 | 6.9±6.7 |
| **Sum** | **285** | **389.8±15.0** | **362.8±14.4** | **43** | **27.0±1.7** | **7.4±6.3** | **374.5±15.9** | **345.0±14.1** | **52** | **29.5±1.8** | **7.9±6.2** | **335.1±14.9** | **301.3±12.8** | **59** | **33.8±2.1** | **10.1±6.2** |
| **Eastern Caucasus** | | | | | | | | | | | | | | | | |
| Northern slope (Tergi headwaters, Sunja Right tributaries, Sulak) | **130** | **49.1±2.5** | **35.4±1.7** | **54** | **13.7±0.84** | **27.9±6.2** | **41.3±2.5** | **27.2±1.6** | **56** | **14.1±0.86** | **34.1±.6.1** | **32.1±2.0** | **16.3±1.1** | **59** | **15.8±0.90** | **49.2±5.7** |
| **Elbrus Massif** | 21 | 125.4±5.3 | 123.1±5.1 | 9 | 2.3±0.16 | 1.8±6.9 | 120.9±4.6 | 117.2±4.3 | 13 | 3.7±0.25 | 3.1±6.8 | 118.4±4.2 | 112.4±3.8 | 18 | 6.0±0.4 | 4.6±6.6 |
| **All selected glaciers** | **659** | **691.5±29.0** | **643.2±25.9** | **122** | **48.3±3.1** | **7.0±6.4** | **656.5±27.9** | **601.9±24.5** | **143** | **54.6±3.4** | **9.1±6.3** | **590.0±25.8** | **511.0±20.9** | **172** | **79.0±4.9** | **13.4±6.2** |

3  * % of the total glacier area

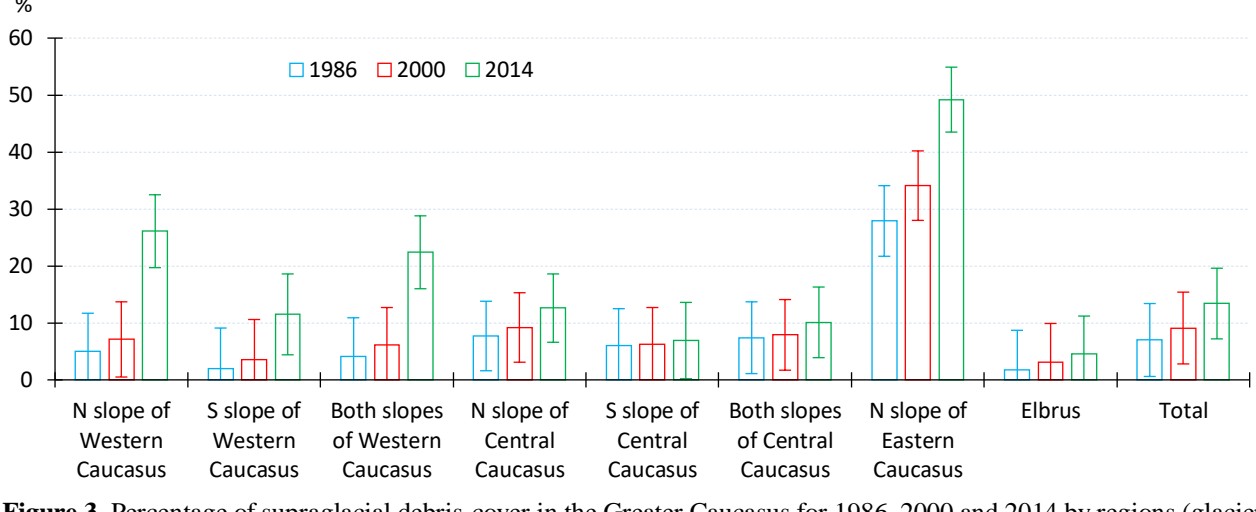

**Figure 3.** Percentage of supraglacial debris-cover in the Greater Caucasus for 1986, 2000 and 2014 by regions (glaciers are non-existent on southern slopes of the eastern Greater Caucasus).

Supraglacial debris-cover area for (the largest) Bezingi Glacier in the Greater Caucasus increased from $4.4\pm0.3$ km$^2$ or $11.0\pm5.9\%$ to $7.5\pm0.4$ km$^2$ or $20.0\pm6.0\%$ (~0.32% yr$^{-1}$) between 1986 and 2014 in contrast with a reduction of the total glacier area from $40.0\pm0.9$ km$^2$ to $37.5\pm0.9$ km$^2$ ($6.3\pm2.2\%$ or ~0.22% yr$^{-1}$) during the same period with a terminus retreat of ~374 m. Comparison with the debris-free Karaugom Glacier (third largest glacier of the Greater Caucasus), located in the same region (northern slope of central Greater Caucasus), shows that the percentage area reduction was almost three times greater than for the debris-covered Bezingi Glacier: from $29.2\pm0.6$ km$^2$ to $24.0\pm0.4$ ($17.8\pm2.1\%$ or ~0.63% yr$^{-1}$) with a terminus retreat of ~1366 m (Fig. S2).

The mean upper limit of the supraglacial debris-cover migrated from ~3015 m to ~3130 m between 1986 and 2014 for all selected glaciers of the Greater Caucasus. The highest mean upper limit lies on the northern slopes of eastern Greater Caucasus (~3626 m), while the lowest limit is on the southern slopes of western Greater Caucasus (~2850 m). On the Elbrus Massif the mean upper limit of the supraglacial debris-cover migrated from ~3345 m to ~3520 m between 1986 and 2014 (Fig. 7, Table S1).

**5 Discussion**

**5.1 Comparison with previous investigations**

Our results are in agreement with studies assessing changes in supraglacial debris-cover for smaller areas in the Greater Caucasus. For example, Stokes et al. (2007) calculated that supraglacial debris-cover increased by 3%-6% (~0.20% yr$^{-1}$) between 1985 and 2000 on several glaciers in the central Greater Caucasus. On individual glaciers, supraglacial debris-cover increased by 25%-30% (e.g. Shkhelda) in the same period. We found an increase of supraglacial debris-cover from $21.3\pm6.0\%$ to $30\pm5.8\%$ (~0.65% yr$^{-1}$) for Shkhelda Glacier between 1986 and 2000. Popovnin et al. (2015), reported a supraglacial debris-cover increase from 2% to 13% (~0.23% yr$^{-1}$) between 1968-2010, based on direct field monitoring for the Djankuat Glacier (northern slope of the central Greater Caucasus). This compares with our result of an area increase of supraglacial debris-cover for Djankuat Glacier from $2.6\pm6.9\%$ to $9.8\pm6.1\%$ (~0.25% yr$^{-1}$) between 1986 and 2014. This difference can be explained in that the detailed field measurement would have picked out smaller spots of debris-cover which were beyond the resolution of the satellite imagery. Lambrecht et al. (2011) estimated that the supraglacial debris-cover distribution remained nearly constant at ~16% between 1971 and 1991 in the Adyl-su River basin (northern slope of the central Greater Caucasus). Between 1991 and 2006, the supraglacial debris-cover started to increase noticeably reaching 23% (~0.46% yr$^{-1}$) within 15 years. For the Zopkhito and Laboda glaciers (southern slope of the central Greater Caucasus), supraglacial debris-

cover increase was lower in the same period (from 6.2% to 8.1% or ~0.12% yr⁻¹). We measured the supraglacial debris-cover increase in the same glaciers from 4.9±6.5% to 6.1±6.4% or ~0.08% yr⁻¹ between 1986 and 2000.

**Figure 4.** Supraglacial debris-cover increase (yellow) and glacier area decrease (green) rates in the Greater Caucasus by slopes, sections and mountain massifs in 1986–2000 (a), 2000–2014 (b) and 1986–2014 (c).

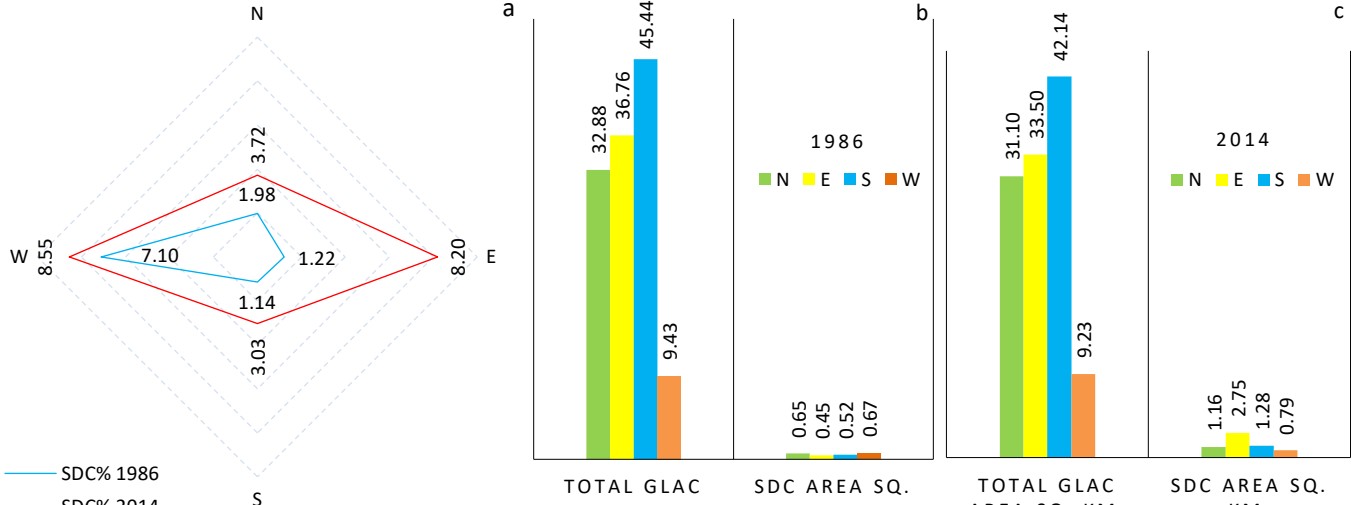

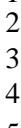

**Figure 5.** Percentage of supraglacial debris-cover (SDC) for the Elbrus slopes in 1986 and 2014 (a). Total glacier area (km$^2$) and supraglacial debris-cover area changes (km$^2$) in 1986 (b) and 2014 (c) for the Elbrus slopes.

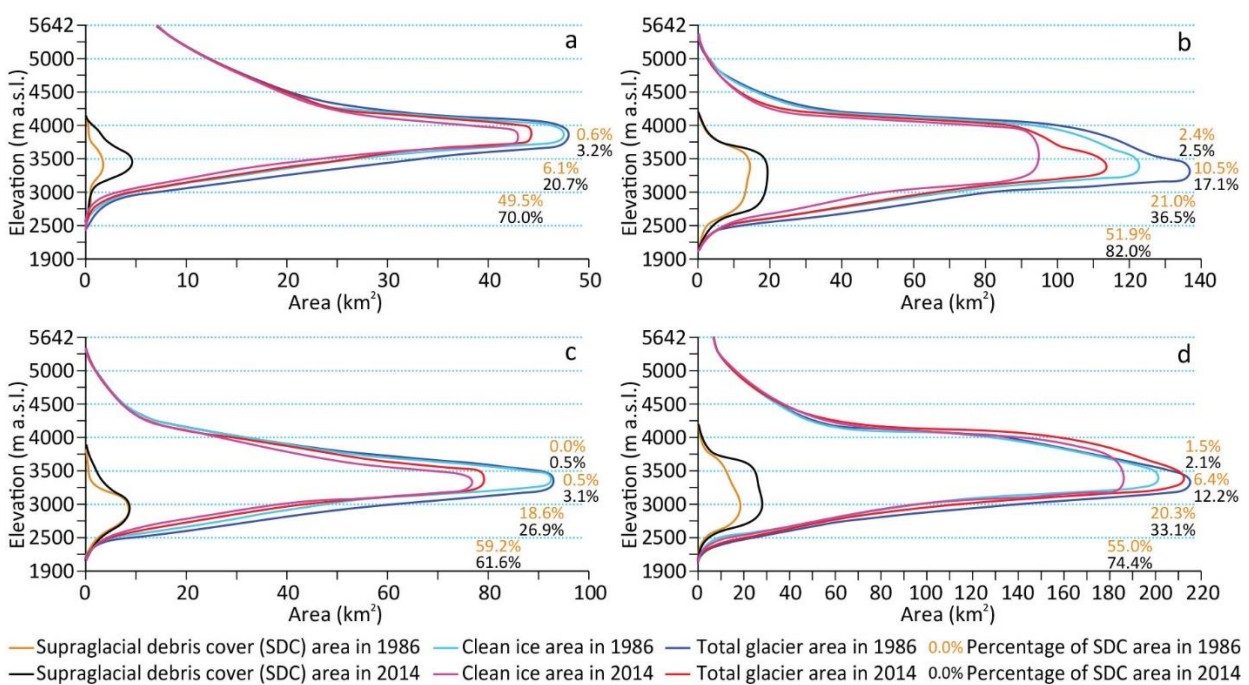

**Figure 6.** Hypsometry of supraglacial debris-cover, clean-ice and total glacier area, of the four study regions in 1986 and 2014. a – Elbrus, b – Northern Slope of the Greater Caucasus, c – Southern Slope, d – All selected glaciers. Supraglacial debris-cover percentage is given according to the different elevation zones in 1986 (brown digits) and 2014 (black digits).

We extracted both supraglacial debris-cover and clean-ice outlines from Scherler et al. (2018) for our glacier sample from 2014 to compare these results of our regional study with those from the global study. We found that a large portion of these glaciers in the Greater Caucasus are covered by supraglacial debris-cover. While for two regions the results match within the uncertainty (western Greater Caucasus: 22.4±6.4% vs. 23.7% and Elbrus 4.6±6.6% vs. 7.7%), our values are lower than the results of Scherler, et al. (2018) for the other two regions (10.1±6.2% vs. 30.8%

for central, 49.2±5.7% vs. 84.9% for eastern) (Fig. S3). These differences can mostly be explained by i) the RGI v6 used by Scherler, et al. (2018), is characterized by some inconsistent co-registration for the Greater Caucasus region which stems from the use of inadequately orthorectified satellite imagery to generate the inventory in contrast to the improved orthorectification of the Landsat L1T data (Fig. S4a); and ii) the RGI v6 contains nominal glaciers (i.e. ellipses around glacier label points) for the Greater Caucasus region which originate from the use of the world glacier inventory (WGI, Haeberli et al., 1989) to fill gaps with no data for earlier versions of the RGI (Pfeffer et al., 2014). According to Scherler et al. (2018), all nominal glaciers were classified as debris-covered (Fig. S4b). We note that the scope of the study by Scherler et al. (2018) was an automatized global assessment of supraglacial debris-cover from optical satellite data, without correcting any outlines in the RGI.

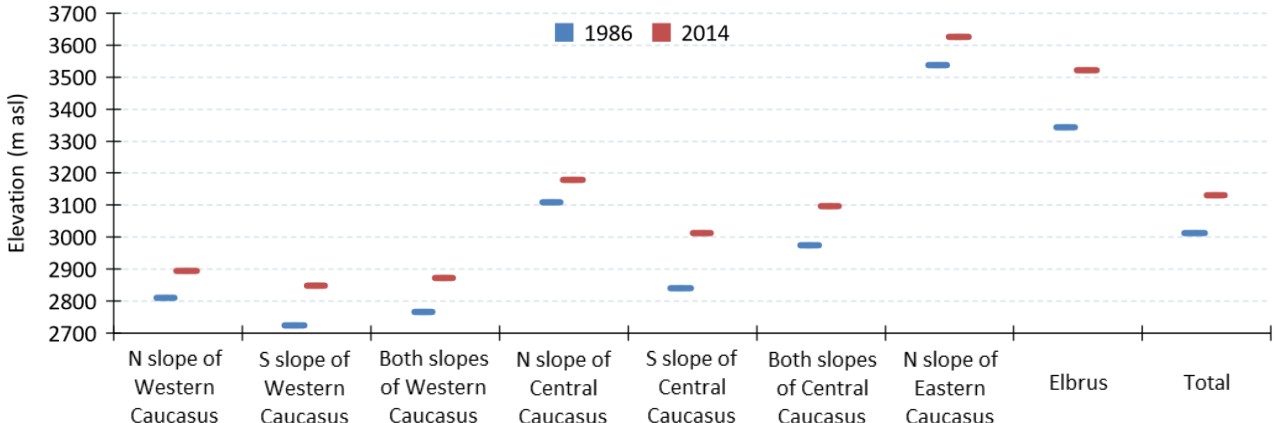

**Figure 7.** Mean upper limit (m asl) of the supraglacial debris-cover for 1986 and 2014 by different regions (glaciers are non-existent on southern slopes of the eastern Greater Caucasus).

## 5.2 Possible reasons for supraglacial debris-cover changes

We observed a clear increase in supraglacial debris-cover in all investigated regions, which became more pronounced after 2000. Based on our investigation, the upper limit of supraglacial debris-cover migrated up-glacier (Fig. 7, 8; Table S2) as a response to glacier retreat thinning and reduced mass flux, as described by Stokes et al. (2007) and defined as 'backwasting' by Benn and Evans (1998). A similar pattern of up-glacier migration has also been described on Tasman Glacier, New Zealand (Kirkbride and Warren, 1999), and on Zmuttgletscher Glacier, Swiss Alps (Mölg et al., 2019).

The results presented in this study indicate that the clean-ice area for all selected glaciers decreased from about 93% to 87% between 1986 and 2014 (Table 2). This reduction was caused by both glacier retreat and an increase in total supraglacial debris-cover (Table 2, Fig. 3-6, S5). This finding is also consistent with field measurements on Djankuat Glacier (Popovnin et al., 2015).

Glacier thinning and a warming atmosphere can lead to permafrost thawing and slope instability at higher altitudes (Deline et al., 2015). Rock avalanches after 2000 on some glaciers in the Greater Caucasus (particularly in the eastern section) might be one of the reasons why the increase rate was higher during our second time period (2000-2014). For example, a rock–ice avalanche onto the Devdoraki Glacier on 17 May 2014 (Tielidze et al., 2019) caused an area increase of supraglacial debris-cover from 5.9±6.0% to 19.1±5.6% or about 0.95% $yr^{-1}$ and a landslide after 2000 onto the Suatisi Glacier produced supraglacial debris-cover area increase from 2.1±6.1% to 17.6±5.7% or about 1.10% $yr^{-1}$ (Fig. S6).

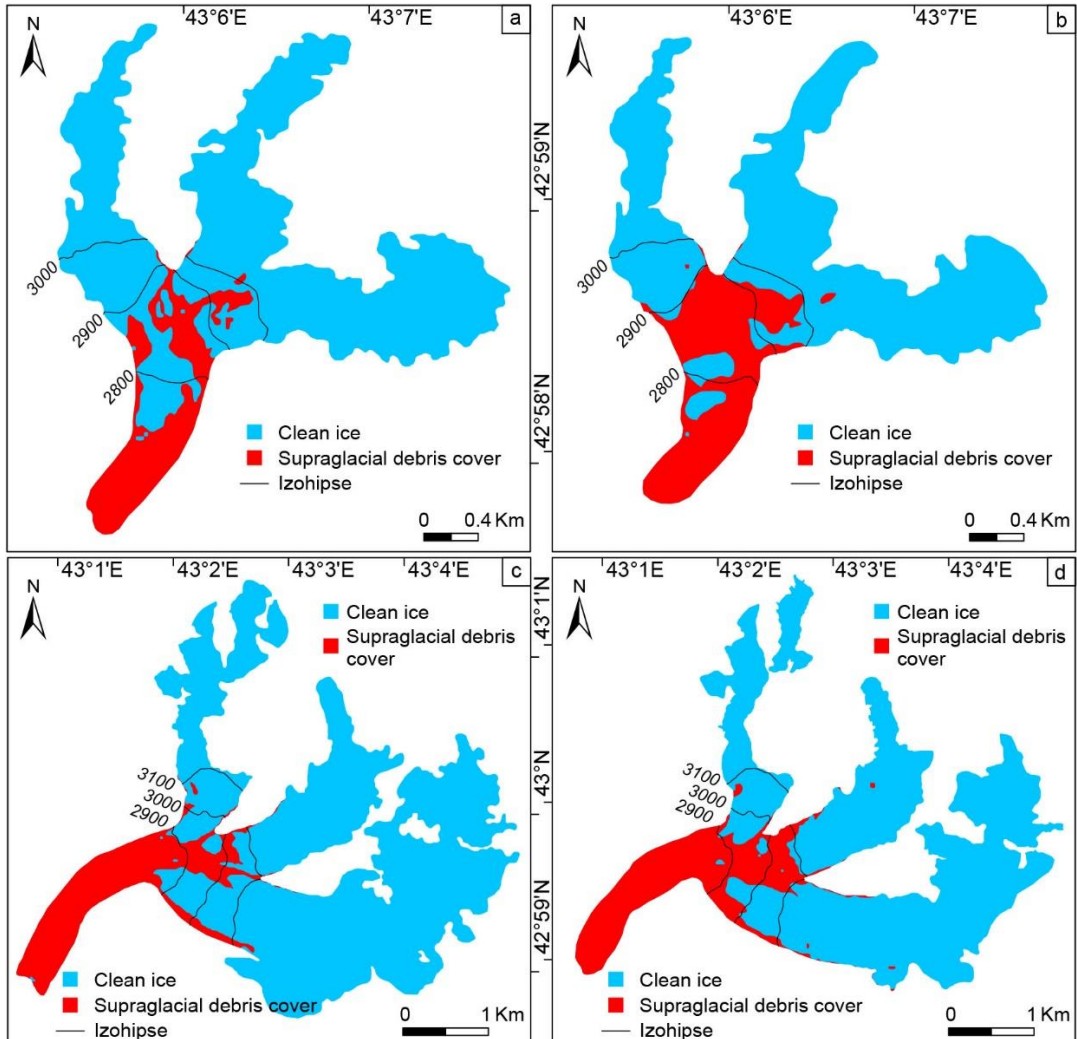

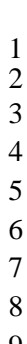
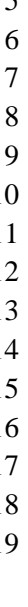

**Figure 8.** An example of the supraglacial debris-cover up-glacier migration onto the Shkhara Glacier: a – 1986, b – 2014 and Khalde Glacier: c – 1986, d – 2014.

Our investigation shows also that the supraglacial debris-cover increases more quickly on the northern slopes of the Greater Caucasus than on the southern, where higher solar radiation input commonly results in smaller glaciers than on the northern slopes. Furthermore, smaller glaciers on the southern slope exist in high cirques with much steeper surface. Glaciers on the northern slopes are on average less steep than on the south mainly because most valley glacier tongues in the north are longer and reach lower altitudes than the south-facing glaciers. This conclusion is supported by Lambrecht et al. (2011) who observed a more rapid increase of supraglacial debris-cover on the northern slopes than the southern.

The highest mean upper limit of the supraglacial debris-cover in the eastern Greater Caucasus can mostly be found at small cirque and simply-valley type glaciers preserved at high altitudes and surrounded by rock walls.

The variation of supraglacial debris-cover area in the eastern, central and western Greater Caucasus could be mostly caused by climate, lithology and morphological peculiarities of the relief. Some river basins in the eastern Greater Caucasus are built on Jurassic sedimentary rocks, which are characterized by relatively high denudation rates (Gobejishvili et al., 2011; Bochud, 2011) supporting supraglacial debris-cover formation. The relief of the central Greater Caucasus is mainly constructed from Proterozoic and Lower Paleozoic plagiogranites, plagiogneisses, quartz diorites and crystalline slates, which are more resistant and are less prone to the formation of rock avalanches. In

addition, the central Greater Caucasus is the highest section of the main range and glacier tongues are relatively steeper and hence, less favourable for debris-cover accumulation. The western Greater Caucasus is hypsometrically lower with less steep glaciers. This section is characterised by the highest glacier retreat after the eastern Greater Caucasus. It is therefore possible that glaciers are also rapidly thinning favouring debris-covered over the ablation area (Benn et al., 2012; Pratap et al., 2015). The dome of the anticlinorium of the western Greater Caucasus (crest of the main water divide) is built on Proterozoic and Paleozoic plagiogneisses, granites, amphibolites, and crystalline slates. This provides the framework for overall denudation of the high mountainous relief (over 3000 m) (Gobejishvili et al., 2019). Furthermore, this area is characterized by active tectonic and ongoing mountain building (uplifting) processes (Tsereteli et al., 2016), which might be a further reason for increasing supraglacial debris-cover. We note that all these reasons need confirmation by detailed field measurements and could be part of a separate investigation since there is no accurate geographical pattern which otherwise explains the clear differences of the increase in supraglacial debris cover.

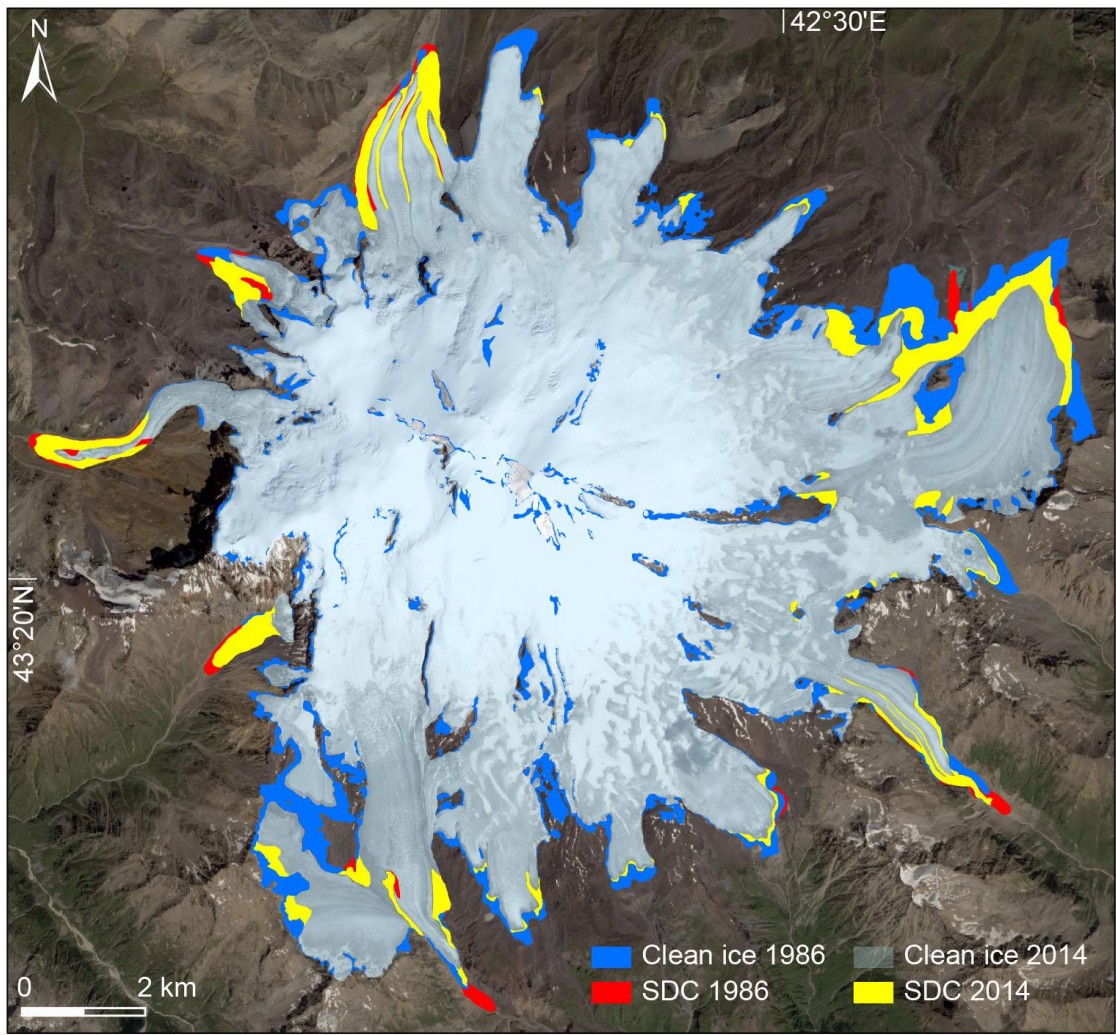

**Figure 9.** Supraglacial debris-cover (SDC) increase on the Elbrus Massif from 1986 to 2014. SPOT-7 image 20/08/2016 is used as the background. Blue color shows retreat of clean-ice parts. Clean-ice in 1986 consists of the clean-ice 2014 (light blue, transparent) plus clean-ice area that retreated between 1986 and 2014 (dark blue).

Our results indicate more than doubling of supraglacial debris-cover area for Elbrus glaciers from 1986 to 2014 with the highest increase rate between 2000 and 2014 (Fig. 4, 9), although the total estimated uncertainty is comparable to the obtained relative changes. Glaciers in the western slope of Elbrus are affected by avalanches and thus are partially debris-covered (Kutuzov et al., 2019). Glaciers on the eastern slope are characterized by high rates of retreat and great expansion in proglacial lake number and area (Petrakov et al., 2007). The most significant increase of supraglacial debris-cover occurred on the eastern oriented glaciers of Elbrus, where glaciers are characterized by the highest thinning rates in recent years (Kutuzov et al., 2019). Detailed Ground-Penetrating Radar (GPR) survey helps to more accurately identify supraglacial debris-cover extent in this area (e.g. GPR measurements by Kutuzov et al. (2019) showed that ~30 m of ice may be present under the previously considered ice-free area on the eastern slope of Elbrus).

The glaciers in the Greater Caucasus have retreated continuously since 1960 (Tielidze and Wheate, 2018), suggesting that the shielding effect of increased supraglacial debris-cover at the glacier surface may only partly offset the retreat trend. This is in line with detailed observations of the evolution of Zmuttgletscher, in the Swiss Alps (Mölg et al., 2019) and that mass changes of debris-covered and debris-free glaciers in the Himalaya are similar (e.g. Berthier et al., 2007; King et al., 2019). Direct field measurements show that thermal resistance of the < 20 cm supraglacial debris-cover for some glaciers (e.g. Djankuat and Zpkhito) in the Greater Caucasus is relatively higher (0.07-0.15°C and 0.05-0.08°C m²/W) than in other glacierised regions of the world (e.g. Baltoro, Karakoram 0.02-0.07°C and Maliy Aktru, Altay 0.02-0.09°C m²/W) (Lambrecht et al., 2011), preventing a more rapid retreat. This process is consistent with our observations of the largest debris-covered (Bezingi) and debris-free (Karaugom) glaciers of the Greater Caucasus, where the latter is characterized with higher area shrinkage and terminus retreat. Numerous authors have found similar model results in the Himalaya (e.g. Scherler et al., 2011; Rowan et al., 2015; Jiang et al., 2018).

**6 Conclusions**

We have presented supraglacial debris-cover change over the last 30 years in the Greater Caucasus region. Manual and semi-automated digitization from satellite imagery (Landsat, SPOT, Google Earth) were used to distinguish the supraglacial debris-cover area in 1986, 2000 and 2014. We expect that this study will significantly improve existing knowledge for this region.

The main conclusions of this research are summarized as follows:

i) An overall glacier shrinkage by 15.8±4.1% in the Greater Caucasus was accompanied by an increase of supraglacial debris-cover from 7.0±6.4% to 13.4±6.2% between 1986 and 2014. Clean-ice area reduced by 20.1±4.0% during the same time.

ii) Supraglacial debris-cover increase rate was relatively lower between 1986 and 2000 (~0.15% yr$^{-1}$), while it doubled in the second investigated period (~0.30% yr$^{-1}$) partly explained by rock avalanches after 2000 on some glaciers (particularly in the eastern section).

iii) The number of debris-covered glaciers (debris larger than 10%) increased from 122 in 1986, to 143 in 2000, and to 172 in 2014.

iv) Increase of supraglacial debris-cover was observed more rapidly on the northern slopes than on the southern ones during the investigated periods.

v) The eastern Greater Caucasus represented the largest percentage of supraglacial debris-cover (49.2±5.7%) for 2014, while the Elbrus Massif contained the lowest percentage of supraglacial debris-cover (4.6±6.6%) in all our study regions.

Given the increasing degree of supraglacial debris-cover in the Greater Caucasus region, by monitoring should be continued, as it constitutes an important control on glacier response to climate change. The recent significant increase of the supraglacial debris-cover area in this region may alter the glacier mass balance in different ways depending on

debris thickness and properties. Such feedbacks can affect future glacier evolution and should be considered in glacier modeling.

Future work should focus on using high resolution aerial/satellite imagery and more detailed field measurements, e.g. glaciological mass balance measurements, characteristics of debris and debris thickness measurement by Ground-Penetrating Radar, or incoming and reflected solar radiation, long-wave terrestrial and returned radiation. This will reduce uncertainties connected with supraglacial debris-cover assessment and glacier mapping accuracy in this region.

**Acknowledgements:**
This work was supported by Shota Rustaveli National Science Foundation of Georgia (SRNSFG) [YS17_12]. We gratefully acknowledge the support of the editor, Etienne Berthier, and two reviewers, Dirk Scherler and Sam Herreid, for useful suggestions and detailed comments which clearly enhanced the quality of the paper. We thank Shaun Eaves for cooperation during the preparation of the paper.

**Information about the Supplement**
The supplement includes glacier size classes with debris-covered and debris-free glaciers distributions for northern and southern slopes (Fig. 1); Comparison of Debris-covered (Bezingi) and Debris-free (Karaugom) glaciers retreat between 1986 and 2014 (Fig. 2); Mean up-glacier migration of the upper limit of the supraglacial debris-cover for 1986 and 2014 by regions (Table 1); Relative supraglacial debris-cover for the western, central, and eastern Greater Caucasus as well as for Elbrus based on the current study and in comparison to Scherler et al. (2018) (Figs. 3-4); Increased supraglacial debris-cover area for Devdoraki and Suatisi glaciers before and after rock-ice avalanches (Fig. 5); A comparison of supraglacial debris-cover and clean-ice area distribution in 1986 and 2014 for south and north facing glaciers (Fig. 6). Total glacier area and supraglacial debris-cover change on the Elbrus Massif from 1986 to 2014 (animation map).

**Data availability**
The new supraglacial debris-cover database created during the investigated period has been submitted to GLIMS, and can be used as a basis dataset for future studies.

*Supplement.* The supplement related to this article is available online at: https://doi.org/…..

*Competing interests.* The authors declare that they have no conflict of interest.

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
