# Peer review of "Brief communication: Supraglacial debris-cover changes in the Caucasus Mountains"

_The Cryosphere, 2018_

## Referee Comment (RC1) · Sam Herreid (Referee) · 6 Feb 2019

General Comments

This paper presents changes in glacier and debris-covered area for several subregions in the Greater Caucasus mountains. The results are accompanied by an error analysis that uses two approaches to quantify mapping error. This work is relevant both by expanding the spatial domain over which debris-covered area changes are measured and by providing a comparison against larger, global scale debris cover mapping efforts. Overall, I think there are methodological deficiencies that need to be addressed, figures that need to be both added and removed and a general improvement in the clarity of the writing.

A key component of any study investigating debris-covered area change is a consistent and meaningful spatial domain. Transient snowfall (possible at any time of year) can cover debris resulting in an underestimation of debris cover that is actually present in a glaciers ablation zone. If a later map of debris cover is generated from an image with a higher snowline, a false debris-covered area change signal will be measured, even in a setting where the position of the equilibrium line is stable. In order to eliminate these errors, a spatial domain can be set at the aggregate lowest minimum snowline from all of the images used to map debris cover. Tracking of the up-glacier migration of debris cover in a phase of glacier shrinkage will require additional attention/data/criteria. If the debris-covered area changes mapped in this study were well below snowline, then showing this will negate the concern. If mapped debris-cover shared a boundary with snow rather than ice or firn, I do not think debris area change measurements can be trusted without more information.

One of the two approaches used in this study to estimate mapping errors is a buffer method which I do not think is sufficiently supported to meaningfully quantify error. I would like to see some evidence supporting the two buffer distances that were selected. Further, it is unclear in the results presented with error bounds which approach they are derived from or if the two approaches are in some way combined. It seems feasible to use the detailed manual error assessment at six glaciers to calibrate a more meaningful buffer approach applied to the entire study area, but I do not believe this was done.

An example of where article clarity could be improved is in the description of the debris cover mapping methods. The methods section is somewhat confusing to follow, yet follows the widely used approach of finding the residual of bare-ice area classified with a band ratio threshold and manual debris cover outlines. The threshold(s) used should also be stated for future studies that might want to repeat/continue this work. An additional focusing of the article is needed to address/remove results and figures that are not supported with motivation in the introduction or methods (e.g. ice thickness measurements).

After considering these general comments and the specific comments below I think this paper would make a nice contribution to the glacier research community.

Specific Comments

P1L20-21: Is it a fact that debris coverage typically increases with shrinking glaciers? I would think this is more of a hypothesis that studies like this will either support or reject.

P1L23: "throughout" or "across" rather than "different regions" might give more information to the reader, better still would be the fraction of the total glacier area that you consider.

P1L25: I think "-0.52% yr^-1"

P1L25: Is glacier area change a result from this work or a result from previously published work?

P1L25: This is not a "Thereby" statement.

P1L25-26: northern and southern slopes of what? Unclear if reading only the abstract.

P1L26-28: The last sentence of the abstract is unclear, unsupported in the text and should be removed.

P1L34: considered to be significant by whom?

P1L34: Isn't the debris cover generally a passive element in a sediment transport system? The sediment is of course a significant part of a sediment transport system, but its role in the efficiency isn't clear to me.

P1L39: "exact evaluation," do you mean precise or accurate?

P2L2-3: "methods for satellite mapping of supraglacial debris remain in development (Zhang et al., 2016)" Do you mean debris thickness, debris-covered area or both? This statement might need additional reference(s).

P2L4: "Several studies" but cite one, add "e.g." or other citations.

P2L6-7: This sentence should be restructured to make clear SDC is one of the complexities in the relation between climate and glacier mass budget.

P2L9-10: add a citation for how we know SDC is an important control for ice ablation.

P2L14: What does "SDC is abundant" mean? Make this statement in objective relative terms or merge with next sentence on P2L15.

P2L15: Can you please specifically state the contradiction? Did earlier studies claim the Caucasus and Middle East region did not have the highest percent debris coverage worldwide?

P2L18-19: This might not need to be changed but what a "region" is isn't very clear.

P2L18-21: This once sentence paragraph needs to be rewritten, further, I don't think a discussion of controlling factors is adequately discussed to be mentioned here. "in light of" isn't clear scientific language and partly suggests there might be some global context when really a global product is sampled to match your spatial domain.

P2L25: Did you select these glaciers individually or did you select whole regions?

P2L25: could you add a citation or a sentence and a citation describing what differences in climate conditions we can have in mind while reading this article?

P2L34-35: "glacier margins digitized manually" I am confused, I thought the glacier outlines are taken from Tielidze and Wheate, 2018. Could you please make it very clear what data exist previously, what work was done for this study, and if the quality/data timing was not sufficient in earlier work, what alterations were made?

P2L37: Again, it is unclear if mapping glaciers is an objective of this study.

P2L39: "All imagery was captured from the 28th of July to the 12th of September." Why? This sentence is unclear and unrelated to the following sentence. My guess is that this the argument used for not considering seasonal snowline (see main comment above)?

P2L40: This is not a sufficient explanation of GPR data acquisition and processing and these data have not been motivated in the introduction.

P3L1: The ASTER GDEM was also used in orthorectification (P2L35).

P3L1: How do these stated errors effect this study?

P3L4: This is a very confusing title and I'm not so sure if there is a comparison described in this section.

P3L5-12: The framework of written "steps" is ineffective here. For example the "then" on P3L8 implies a 3rd step but is not called as such. I think there are more than two clear steps and therefore suggest restructuring the presentation of information.

P3L6: I believe you identified "clean-ice", not "clean-ice glaciers."

P3L6-7: I don't think it is useful to the reader to know about data formats (raster polygons and vector data)

P3L7: I think it is better to say "removed misclassified area" rather than "deleted misclassified polygons" If a polygon was half correct would you still delete it? It was unclear in earlier sections that mapping glaciers was an objective of this study, it seemed like that task was complete and now the debris cover would be found as a residual from identifying bare ice only. Does this mean that Tielidze and Wheate, 2018 did not consider debris cover and therefore significantly underestimated glacier area?

P3L8: "as accurately as possible" this is not meaningfully to the reader, please be more specific.

P3L9: can you please clarify what you are assessing here? Did you classify thin medial moraines as debris covered instead of bare ice? I applaud this effort to consider medial moraines below the detection limit but a drawback of this is your end results become more difficult to reproduce in future comparison studies.

P3L10-12: This sentence is a bit awkward and confusing when really you are applying

a very common technique used to map debris cover. A more simplistic description is "debris cover is classified as the residual between an automatically derived bareice map and a manually generated glacier extent map." With a citation usually to: Paul, Frank, Christian Huggel, and Andreas Kääb. "Combining satellite multispectral image data and a digital elevation model for mapping debris-covered glaciers." Remote sensing of Environment 89.4 (2004): 510-518.

P3L13: The difficult boundaries are not clearly explained. You list SDC, moraines and debris in shadow, in my opinion there should be no boundaries between these three. Do you mean moraines off the glacier? The writing of this list is also awkward.

P3L15: I would be interested to know if the glacier edge picked from very high resolution agreed (independently) with your GPS measurements. In other words, Between Landsat derived outlines and field measurements, how much aid is heightened resolution?

P3L16-17: This GPS data sounds very useful for validation of this work but if you are going to present it you will need to describe the sensor, field and processing methods, time of acquisition and location. Are you referring to the one point in Figure S4? Can you convince us readers that being in the field actually enables locating the terminus position better than high resolution imagery? I think for some glaciers this is true but for others it is so unclear that an aerial perspective is the best for outlining a glacier's edge.

P3L16: One-half pixel is not a helpful unit of measurement, please present in meters and describe how this error "[was] assumed."

P3L17-19: Cases of uncertainty are very nice for the reader if they are shown in an example. E.g. in your Figure 1 it would help us understand the limitations you encountered to have one of the examples be at a location of uncertainty. This may also help inspire future work to solve the difficulties faced here.

P4L2: Do you mean buffer distances when you say "sizes"

P4L2: "a sample of manually digitization" corrected to "manual digitization" news a few more details to link to uncertainty estimation.

P4L4: Use meters rather than pixel for units.

P4L4: How did you pick these buffer values? At P4L22 you cite an article reporting "five pixels" of error, does this article also support your using 2? Or 1?

P4L5-6: "an average ratio between the original glacier areas and the areas with a buffer increment." It is unclear what is meant by original. This is also stated as a singular average, are you considering debris and bare ice separately? Do you include bare-ice/debris boundaries internal to the glacier? If so are you double counting error at these locations?

P4L7: I don't think the percentage error should be a function of area. I also would anticipate errors to be larger for earlier sensors and improve with the higher radiometric resolution of Landsat 8.

P4L10: If a method does not produce realistic results, as stated, you should not include it! This is anyhow an interpretation that belongs in the discussion. The two methods also have various strengths and weaknesses, there is an advantage to an error estimate that considers the whole area rather than six glaciers only. However, I do not see much value in the buffer method error estimates.

P4L9-16: It's not clear if we are only talking about the outline of the glacier or the outline of the glacier and the outline of the debris. Also here you establish an unspecified classification "debris covered glacier." What is a debris covered glacier? What criteria did you make this classification on? Was it an automated or manual classification? Is there a physical or processed based motivation for this Boolean classification? A figure showing examples of the error analysis should be at least in the SI.

P4L17-19: The GPR data looks very interesting but it is not appropriately developed in

this article. It is unclear if this is new work done for this paper or existing work presented in a different publication. If it is new and being presented here first there needs to be motivation in the introduction, methods and stand-alone results. If citing existing work, I don't think it is necessary to have Figure S6, and each statement regarding GPR work should be appropriately cited. A GPR trace that shows an ice thickness of zero off glacier that transitions to a non-zero ice thickness under debris is very interesting and relevant for glacier and debris mapping, however, as it is now, Figure S6 is beyond the scope of this article.

P4L22-24: I don't find this argument for using a 30 m buffer to sound.

P4L27: Please clarify what you mean by "a significant increase." Do you mean within statistical significance there is a change (this would be the most meaningful use of the word) or do you mean you consider the amount of increase to be significant based on some unstated prior understanding? Considering your upper estimate for 1986 (12.6% debris-covered) and your lower estimate for 2014 (11.6 debris-covered), your results imply a decrease in SDC. Considering this, your results do not show a significant change.

P4L28: I do not think it is established that debris area changes are concomitant with glacier area change.

P4L32: If you have solved for errors numerically, why are you using a tilde here? (comment extends also to the abstract).

P5L8: The up-glacier migration is not shown in any figures or presented in the results, please include this along with evidence that is not due to seasonal snow variability. Showing that all mapped debris cover is well below the seasonal snow line is sufficient. If, however, the debris cover extends to the seasonal snowline convincing the reader the signal is up-glacier migration will become more difficult but it is essential to make any statement about up-glacier migration of mapped debris cover.

P5L12 (and Figure S2): A image showing the glacier before and after rock avalanche deposits would make this point much more clear. I would like to see some quantification of "dramatically increased SDC" or it is not adding information. Mapping and quantifying the area of SDC from rock avalanches would add a nice additional dimension to this article without requiring much extra work and might help you address the title of this section "SDC increase possible reasons" which should be rewritten as "Possible reasons for an increase in SDC."

P5L14: "..recently for some glaciers" I believe the reference you cite considers one glacier not several.

P5L16: "the reduction of glaciers is mainly at the expense of clean ice" This is both unclear and possibly not correct. Are you talking about changes in x,y or z? Please defend this statement if you elect to keep it here.

P5L22-23: Does local mean at an ice cliff scale or do you mean rocks are sliding down large portions of a glacier? At what glacier slope do you think rocks are able to accumulate?

P6L4-6: this information on lateral moraines either needs to be cited or the measurements be motivated in the introduction, described in the methods and presented in the results.

P6L8-9: Can you please offer support to the statement that "a large percentage of the debris cover is a result of the lithology" What percentage? A glacier surrounded by highly erosive rock at a very low angle might not generate any debris cover.

P6L14: In the framework of glaciology, 20-40 m of ice is not "substantial". 20-40 m of ice is likely not deforming internally and could be stagnant rendering it not part of a glacier following the classical definition of a glacier.

P6L17: "DC" not defined.

P6L38-41: Are you sure there are not other reasons for a difference between Scherler

et al., 2018 and the two points you describe?

P7L10-13: A 50% increase in debris cover is not reported in your results. Please clarify how this value is calculated.

Table 1: For future work that might want to cite this article, it might be convenient for additional rows that give the sum of all of these sub-regions. I believe these are also the values you cite in the article. I also would suggest showing the changes in a time series plot with error bars.

Figure 1: e,f and g need to be shown in the upper (unlabeled) panel of this figure. This is may be personal preference, but I think the top panel could be a stand-alone location figure with panels e,f,g being their own figure or coupled to a Figure similar to Figure S7.

Figure 2: According to the text all of the differences between your results and Scherler et al., 2018 is datum shifts and erroneously classified nominal glacier ellipses. Do these sources really explain all of the differences shown in this figure?

Figure S1: How did you define debris covered and debris free glaciers? I think this classification should be shown in Figure 1 or elsewhere so it is clear for future work what was considered "debris-covered." What glacier criteria did you apply to classify 0.01-0.05 km^2 land surfaces as independent glaciers? I would be interested to see some of these glaciers along with a satellite image and their debris maps.

Figure S2: This figure does not provide much if any information to the article and does not fit the scope of the work that was done.

Figure S4: It is not clear what is meant by "semi-automated." The whole approach to mapping debris cover could be called semi-automated, but what non-automated work went into the classification of bare ice alone? The trace of the longitudinal profile needs to be shown in a or b. Dashed line in a and b should probably be defined.

Figure S5: Rather than what is essentially a repeated figure S4 in a different location, I

would like to see more of the changes. Oblique photography is nice, but here does not offer much information.

Figure S6: Panels b and c showing ice thickness measurements have no established relevance to this article. I would suggest removing this figure.

Figure S7: I think an altered and expanded upon version of this figure is the key figure of this work and should be in the main article. Changes in glacier area and debris covered area are somewhat difficult to see side by side, I would recommend taking an overlap/transparency approach similar to the following two articles for visually showing changes in glacier and debris-covered areas:

Glasser, Neil F., et al. "Recent spatial and temporal variations in debris cover on Patagonian glaciers." Geomorphology 273 (2016): 202-216.

Herreid, Sam, et al. "Satellite observations show no net change in the percentage of supraglacial debris-covered area in northern Pakistan from 1977 to 2014." Journal of Glaciology61.227 (2015): 524-536.

I would also recommended a plot in this style be shown for all of the regions you consider (or at least large portions of them). To keep the article a brief communication maybe add one of these to the main article and have several more in the SI.

Figure S8: It is confusing to discuss "Debris cover outline[s]" as well as "Bare ice outline[s]" as you are using different words to reference effectively the same thing.

Technical corrections

P1L30: I think "SDC" should be defined at the first mention of debris cover.

P1L33: Remove "the" before "SDC" and "glacier ablation."

P1L37: Add "summarized in Kirkbride and.."

P1L40: "The difficulty. . ." I would say "One difficulty.."

P2L1: Use SDC consistently if defined, e.g. here: "properties of debris"

P2L1: Change to "of a debris layer has"

P2L10: Change "as it is similarity in" to "as it is similar to"

P2L10: Comma after citation

P2L11: Consider changing "key player" to "glacier-wide component"

P2L12-13: "as surface mass balance. . .is different from that of bare ice" this has already been established earlier in your introduction and I don't think it needs to be restated.

P2L19: Change to "and a recently"

P2L29: Change to "as the largest"

P2L31-33: Awkward sentence, please break into two

P2L35: "imagery from 2016. The SPOT" not clear if one or several images were used

P3L14: I would remove "Relatively heavily"

P3L15: add "..to distinguish the glacier boundary" or "glacier terminus"

P3L18: change to "..might result in a potential.."

P4L1: change to "i) a buffer method"

P4L2 "manual"

P4L9: correct English in this sentence

P4L13: end parenthesis after NSD

P6L5: please add "Glacier" after named glaciers, here and throughout the article (or "glaciers" after a list)

P6L30: This is not a "whereas" statement.

P6L30: "increment" should be "increase."

P7L14: "vital" seems like too strong of language to me.

---

## Referee Comment (RC2) · Dirk Scherler (Referee) · 7 Feb 2019

General comments

This contribution presents satellite imagery derived changes in supraglacial debris cover in the Caucasus Mountains between 1986, 2000, and 2014, based on Landsat and Spot imagery. The paper presents interesting data although I find some of the methodology unclear. The analysis of the data could be extended to support, or refute, some of the inferences in the discussion, which is sometimes rather speculative. When addressing these issues, the paper should be a relevant contribution, but may be better published as a standard format paper, instead of a brief communication?

Main comments

The description of the methods takes up a considerable fraction of the entire paper, but there still is some information missing. I had the biggest difficulties to understand how the GPR data was used. Hardly any information is given about it, but it appears to be relevant for "correcting" the debris covered area on the Elbrus Massif. If the GPR data was used in this study, I think the authors should provide much more information about the data and the results. At present, I've only seen one figure in the supplementary material (Figure S6). How reliable are the GPR measurements? Is there room for interpretation or was it all straight forward? I don't question the observations; I would just like to see more of the data the authors collected.

Overall, I found the discussion quite confusing. It starts with a chapter on possible reasons for the observed SDC increase, but this chapter also addresses the question of spatial differences in SDC, without any temporal aspect. The authors mention many potential reasons for either spatial differences or temporal changes (and I think many of them are truly meaningful), but they remain speculative. I see potential to address some of these reasons with the current data, but that would require additional analysis. For example, the authors suggest that rock avalanches after 2000 may be one of the reasons why SDC increased more during the period 2000-2014 and they provide examples in the supplementary material. Wouldn't it be possible to quantify these in order to assess their relevance? From the supplementary figure, it wasn't clear to me if all of the rock avalanches were deposited in the ablation zone. If not, some of them may be gone again when buried under snow and ice in a few years. The authors also suggest that topographic differences between northern and southern slopes are responsible for spatial differences in debris cover. Instead of opposing just the two regions, wouldn't it make sense to analyze the glaciers and their topographic setting for testing this idea? If true, there should be a correlation between the topographic reason and the observed difference. There exist other inferences or statements ("Little Ice Age moraine can affect the SDC increase on the glacier tongue, as debris often falls from lateral moraines onto the glacier surface"; "in the eastern Greater Caucasus, a large percentage of the debris cover is a result of the lithology") that should be better backed up by observations, for example, from the spatial distribution of debris cover and its increase on the glacier surfaces.

The comparison with previous estimates of debris cover in the Caucasus Mountains is useful and worth reporting. However, when comparing results with the values reported in our recent paper (Scherler et al., 2018), it should be explicitly stated that the scope of our study was a different one. We attempted an automatized global assessment, knowing and discussing the issues of erroneous glacier outlines in the RGI and we explicitly stated that we did not correct any outlines in the RGI. In other words, we did not pretend to get the debris cover correct, if the RGI outlines are not correct. This is an important point that should be acknowledged to avoid making a straw man argument! It is also not clear to me how the data of Figure 2 was put together. Were only those glaciers compared that were analyzed in both studies? And what about the circular glaciers? Overall, I find Figure 2 not relevant. More relevant would be the comparison of the clean ice-debris cover boundary between our and this study, as in our automatized mapping, we were relying on a single threshold value for the entire Earth.

Specific comments

P1L24: "Thereby": I don't see the causal connection to the foregone sentence P1L26-28: It would be good if you could provide an explanation, or your favorite explanation, why this is the case.

P2L2: "supraglacial debris thickness" P2L8: "Europe": Nothing important, but I'm wondering if all of the Caucasus Mountains and thus all of the glaciers and their area are part of Europe? P2L10: Delete "similarly" P2L15: How does it contradict earlier studies? Please specify. P2L25: What determined your selection of glaciers? Does this mean there exist glaciers that you did not consider? Please clarify. P2L29: "a largest" -> "the largest" P2L39: "Additionally, . . ." -> Please specify how you included what kind of GPR results in which way. Also give references if it is published. If it is not published,

I think you need to provide much more information on the GPR data.

P3L1: ASTER GDEM asks for a certain way of acknowledgement that is missing. P3L5: What threshold value did you use to distinguish between ice and debris? P3L13: "SDC is" -> "SDC are" P3L14: Delete "Relatively" P3L16: Again, it is unclear how you used GPR data (see comment above). Figure 1: I think this figure should be larger – but maybe that's just due to the formatting of the PDF. P3L22: "sections" -> "regions" P3L23: "Elbrus Massif"

P4L9: Delete "however" & "critical to" -> "critical for" P4L10: "performed" -> "used" & Why is method 2 giving you a more realistic uncertainty estimate? P4L19: How is the geomorphology complicated? P4L31: "debris cover" -> "SDC" & comma after "increased"

P5L8: "Debris cover migrated up-glacier" -> I haven't seen any data on the spatial distribution of the SDC. If you have, it would be worth showing it.

P5L10-P6L11: This chapter sounds more like results. Also see the main comments above.

P6L14: Regarding the GPR results, see main comments above. P6L25: Not clear how the cited studies and this one are broadly consistent. In that there is an increase? It appears difficult to compare a regional study with individual glacier studies. Perhaps compare results from this studies with previous ones by limiting to those glaciers that are in common? P6L31: "141% increment"? P6L36-P7L3: See main comments above.

P7L10: SCD -> SDC P7L15: "periglacial debris cover": this term comes surprising and it's unclear how this conclusion came about. Also, why is the monitoring "vital"? P7L16: Delete "a"

---

## Author Comment (AC1) · 1 Jun 2019

**Authors reply to Dr. Sam Herreid's comments**

**"Brief communication: Supraglacial debris-cover changes in the Caucasus Mountains" by L. G. Tielidze, et al.**

The Cryosphere Discuss.,
https://doi.org/10.5194/tc-2018-259

Dear Dr. San Herried,

Thank you very much for your detail comments which we help to increase the quality of our manuscript. Please find in the following a point-by-point reply to your review.

All corrections and changes what we did in the text are in yellow.

| |
|---|
| **General Comments** |
| This paper presents changes in glacier and debris-covered area for several subregions in the Greater Caucasus mountains. The results are accompanied by an error analysis that uses two approaches to quantify mapping error. This work is relevant both by expanding the spatial domain over which debris-covered area changes are measured and by providing a comparison against larger, global scale debris cover mapping efforts. Overall, I think there are methodological deficiencies that need to be addressed, figures that need to be both added and removed and a general improvement in the clarity of the writing. |
| We agree that methodological deficiencies were one of the main problems of the first version of the manuscript. Considering this, we provided a new chapter of methodology, with a more detail description. Several figures were added/deleted as well. |
| A key component of any study investigating debris-covered area change is a consistent and meaningful spatial domain. Transient snowfall (possible at any time of year) can cover debris resulting in an underestimation of debris cover that is actually present in a glaciers ablation zone. If a later map of debris cover is generated from an image with a higher snowline, a false debris-covered area change signal will be measured, even in a setting where the position of the equilibrium line is stable. In order to eliminate these errors, a spatial domain can be set at the aggregate lowest minimum snowline from all of the images used to map debris cover. Tracking of the up-glacier migration of debris cover in a phase of glacier shrinkage will require additional attention/data/criteria. If the debris-covered area changes mapped in this study were well below snowline, then showing this will negate the concern. If mapped debris-cover shared a boundary with snow rather than ice or firn, I do not think debris area change measurements can be trusted without more information. |
| We have added new text and figures in order to approve the SDC up-glacier migration. Please see P6 L15-19, P7 L6-10, Fig. S3 (P2). |
| One of the two approaches used in this study to estimate mapping errors is a buffer method which I do not think is sufficiently supported to meaningfully quantify error. I would like to see some evidence supporting the two buffer distances that were selected. |
| Further, it is unclear in the results presented with error bounds which approach they are derived from or if the two approaches are in some way combined. It seems feasible to use the detailed manual error assessment at six glaciers to calibrate a more meaningful buffer approach applied to the entire study area, but I do not believe this was done. |

We excluded multiple digitization method in this version of manuscript.

There is not yet an ideal method for estimating mapping errors for debris-covered glaciers, even though many attempts have been made from various studies.

The buffer method is widely used and adopted for mapping errors of debris-covered/debris-free glaciers by many scientists (Bolch et al., 2010; Frey et al., 2012; Mölg et al., 2018, and many others. See these references in the main manuscript), that's why we decided to use the buffer method in this study.

The uncertainty for debris-covered and debris-free glaciers was calculated separately based on this two different data-set (outlines).

An example of where article clarity could be improved is in the description of the debris cover mapping methods. The methods section is somewhat confusing to follow, yet follows the widely used approach of finding the residual of bare-ice area classified with a band ratio threshold and manual debris cover outlines. The threshold(s) used should also be stated for future studies that might want to repeat/continue this work. An additional focusing of the article is needed to address/remove results and figures that are not supported with motivation in the introduction or methods (e.g. ice thickness measurements).

We agree that it was somewhat confusing and considering this we provided a new description (including the threshold values) of all steps of the methodology. We deleted all extra figures e.g. ice thickness measurements, etc.

**Specific comments**

*P1L20-21: Is it a fact that debris coverage typically increases with shrinking glaciers? I would think this is more of a hypothesis that studies like this will either support or reject.*

*P1L23: "throughout" or "across" rather than "different regions" might give more information to the reader, better still would be the fraction of the total glacier area that you consider.*

*P1L25: I think "-0.52% yrˆ-1"*

*P1L25: Is glacier area change a result from this work or a result from previously published work?*

*P1L25: This is not a "Thereby" statement.*

*P1L25-26: northern and southern slopes of what? Unclear if reading only the abstract.*

*P1L26-28: The last sentence of the abstract is unclear, unsupported in the text and should be removed.*

We have completely changed an abstract text, please see new version P1 L24-34

*P1L34: considered to be significant by whom?*

*P1L34: Isn't the debris cover generally a passive element in a sediment transport system? The sediment is of course a significant part of a sediment transport system, but its role in the efficiency isn't clear to me.*

We have changed this sentence, please see P1 L39-40

*P1L39: "exact evaluation," do you mean precise or accurate?*

We have changed this sentence, please see P2 L4-6

*P2L2-3: "methods for satellite mapping of supraglacial debris remain in development (Zhang et al., 2016)" Do you mean debris thickness, debris-covered area or both? This statement might need additional reference(s).*

We meant the thickness of debris, please see P1 L7-9

*P2L4: "Several studies" but cite one, add "e.g." or other citations.*

We added the second reference here, please see P2 L9-10

*P2L6-7: This sentence should be restructured to make clear SDC is one of the complexities in the relation between climate and glacier mass budget.*

We have deleted this sentence

*P2L9-10: add a citation for how we know SDC is an important control for ice ablation.*

We have changed this sentence, please see P2 L16-18

*P2L14: What does "SDC is abundant" mean? Make this statement in objective relative terms or merge with next sentence on P2L15.*

| |
|---|
| *P2L15: Can you please specifically state the contradiction? Did earlier studies claim the Caucasus and Middle East region did not have the highest percent debris coverage worldwide?* |
| We have changed this sentence, please see P2 L20-22 |
| *P2L18-19: This might not need to be changed but what a "region" is isn't very clear.*

*P2L18-21: This once sentence paragraph needs to be rewritten, further, I don't think a discussion of controlling factors is adequately discussed to be mentioned here. "in light of" isn't clear scientific language and partly suggests there might be some global context when really a global product is sampled to match your spatial domain.* |
| We have changed this sentence, please see P2 L25-27 |
| *P2L25: Did you select these glaciers individually or did you select whole regions?*

*P2L25: could you add a citation or a sentence and a citation describing what differences in climate conditions we can have in mind while reading this article?* |
| We have changed this sentence, please see P2 L31-32 |
| *P2L34-35: "glacier margins digitized manually" I am confused, I thought the glacier outlines are taken from Tielidze and Wheate, 2018. Could you please make it very clear what data exist previously, what work was done for this study, and if the quality/data timing was not sufficient in earlier work, what alterations were made?*

*P2L37: Again, it is unclear if mapping glaciers is an objective of this study.* |
| We have changed this paragraph, please see P2 L38-42 |
| *P2L39: "All imagery was captured from the 28th of July to the 12th of September." Why? This sentence is unclear and unrelated to the following sentence. My guess is that this the argument used for not considering seasonal snowline (see main comment above)?* |
| We have changed this paragraph, please see P3 L3-4 |
| *P2L40: This is not a sufficient explanation of GPR data acquisition and processing and these data have not been motivated in the introduction.* |
| We have excluded GPR data in this study |
| *P3L1: How do these stated errors effect this study?* |
| We have changed this paragraph, please see P3 L5-9 |
| *P3L4: This is a very confusing title and I'm not so sure if there is a comparison described in this section.* |
| We have changed the title, please see P3 L20 |
| *P3L5-12: The framework of written "steps" is ineffective here. For example the "then" on P3L8 implies a 3rd step but is not called as such. I think there are more than two clear steps and therefore suggest restructuring the presentation of information.*

*P3L6: I believe you identified "clean-ice", not "clean-ice glaciers."*

*P3L6-7: I don't think it is useful to the reader to know about data formats (raster polygons and vector data)*

*P3L7: I think it is better to say "removed misclassified area" rather than "deleted misclassified polygons" If a polygon was half correct would you still delete it? It was unclear in earlier sections that mapping glaciers was an objective of this study, it seemed like that task was complete and now the debris cover would be found as a residual from identifying bare ice only. Does this mean that Tielidze and Wheate, 2018 did not consider debris cover and therefore significantly underestimated glacier area?*

*P3L8: "as accurately as possible" this is not meaningfully to the reader, please be more specific.*

*P3L9: can you please clarify what you are assessing here? Did you classify thin medial moraines as debris covered instead of bare ice? I applaud this effort to consider medial moraines below the detection limit but a drawback of this is your end results become more difficult to reproduce in future comparison studies.*

*P3L10-12: This sentence is a bit awkward and confusing when really you are applying a very common technique used to map debris cover. A more simplistic description is "debris* |

*cover is classified as the residual between an automatically derived bareice map and a manually generated glacier extent map." With a citation usually to: Paul, Frank, Christian Huggel, and Andreas Kääb. "Combining satellite multispectral image data and a digital elevation model for mapping debris-covered glaciers." Remote sensing of Environment 89.4 (2004): 510-518.*

*P3L13: The difficult boundaries are not clearly explained. You list SDC, moraines and debris in shadow, in my opinion there should be no boundaries between these three. Do you mean moraines off the glacier? The writing of this list is also awkward.*

*P3L15: I would be interested to know if the glacier edge picked from very high resolution agreed (independently) with your GPS measurements. In other words, Between Landsat derived outlines and field measurements, how much aid is heightened resolution?*

*P3L16-17: This GPS data sounds very useful for validation of this work but if you are going to present it you will need to describe the sensor, field and processing methods, time of acquisition and location. Are you referring to the one point in Figure S4? Can you convince us readers that being in the field actually enables locating the terminus position better than high resolution imagery? I think for some glaciers this is true but for others it is so unclear that an aerial perspective is the best for outlining a glacier's edge.*

*P3L16: One-half pixel is not a helpful unit of measurement, please present in meters and describe how this error "[was] assumed."*

*P3L17-19: Cases of uncertainty are very nice for the reader if they are shown in an example. E.g. in your Figure 1 it would help us understand the limitations you encountered to have one of the examples be at a location of uncertainty. This may also help inspire future work to solve the difficulties faced here.*

*P4L2: Do you mean buffer distances when you say "sizes"*

*P4L2: "a sample of manually digitization" corrected to "manual digitization" news a few more details to link to uncertainty estimation.*

*P4L4: Use meters rather than pixel for units.*

*P4L4: How did you pick these buffer values? At P4L22 you cite an article reporting "five pixels" of error, does this article also support your using 2? Or 1?*

*P4L5-6: "an average ratio between the original glacier areas and the areas with a buffer increment." It is unclear what is meant by original. This is also stated as a singular average, are you considering debris and bare ice separately? Do you include bare-ice/debris boundaries internal to the glacier? If so are you double counting error at these locations?*

*P4L7: I don't think the percentage error should be a function of area. I also would anticipate errors to be larger for earlier sensors and improve with the higher radiometric resolution of Landsat 8.*

*P4L10: If a method does not produce realistic results, as stated, you should not include it! This is anyhow an interpretation that belongs in the discussion. The two methods also have various strengths and weaknesses, there is an advantage to an error estimate that considers the whole area rather than six glaciers only. However, I do not see much value in the buffer method error estimates.*

*P4L9-16: It's not clear if we are only talking about the outline of the glacier or the outline of the glacier and the outline of the debris. Also here you establish an unspecified classification "debris covered glacier." What is a debris covered glacier? What criteria did you make this classification on? Was it an automated or manual classification? Is there a physical or processed based motivation for this Boolean classification? A figure showing examples of the error analysis should be at least in the SI.*

*P4L17-19: The GPR data looks very interesting but it is not appropriately developed in this article. It is unclear if this is new work done for this paper or existing work presented in a different publication. If it is new and being presented here first there needs to be motivation in the introduction, methods and stand-alone results. If citing existing work, I don't think it is necessary to have Figure S6, and each statement regarding GPR work*

*should be appropriately cited. A GPR trace that shows an ice thickness of zero off glacier that transitions to a non-zero ice thickness under debris is very interesting and relevant for glacier and debris mapping, however, as it is now, Figure S6 is beyond the scope of this article.*

*P4L22-24: I don't find this argument for using a 30 m buffer to sound.*

We improved methodology chapter and provided new text, please see P3 L21-34 and P 4L1-24

*P4L27: Please clarify what you mean by "a significant increase." Do you mean within statistical significance there is a change (this would be the most meaningful use of the word) or do you mean you consider the amount of increase to be significant based on some unstated prior understanding? Considering your upper estimate for 1986 (12.6% debris-covered) and your lower estimate for 2014 (11.6 debris-covered), your results imply a decrease in SDC. Considering this, your results do not show a significant change.*

*P4L28: I do not think it is established that debris area changes are concomitant with glacier area change.*

We have changed this sentence, please see P4 L27-30

*P4L32: If you have solved for errors numerically, why are you using a tilde here? (comment extends also to the abstract).*

We deleted all tildes in the manuscript

*P5L8: The up-glacier migration is not shown in any figures or presented in the results, please include this along with evidence that is not due to seasonal snow variability. Showing that all mapped debris cover is well below the seasonal snow line is sufficient. If, however, the debris cover extends to the seasonal snowline convincing the reader the signal is up-glacier migration will become more difficult but it is essential to make any statement about up-glacier migration of mapped debris cover.*

We provided new figures showing up-glacier migration, please see Fig. S3 (in supplement), where the SDC is well below to the snow line. In addition, we provided a new Fig. 3 (please see P7 L5-10) showing SDC vertical distributions, that approves SDC increase in upper elevations. Please see the appropriate text as well, P6 L15-19

*P5L12 (and Figure S2): A image showing the glacier before and after rock avalanche deposits would make this point much more clear. I would like to see some quantification of "dramatically increased SDC" or it is not adding information. Mapping and quantifying the area of SDC from rock avalanches would add a nice additional dimension to this article without requiring much extra work and might help you address the title of this section "SDC increase possible reasons" which should be rewritten as "Possible reasons for an increase in SDC."*

We provided new Fig. S4 (in supplement) showing SDC increase after rock avalanche

*P5L14: "..recently for some glaciers" I believe the reference you cite considers one glacier not several.*

*P5L16: "the reduction of glaciers is mainly at the expense of clean ice" This is both unclear and possibly not correct. Are you talking about changes in x,y or z? Please defend this statement if you elect to keep it here.*

*P5L22-23: Does local mean at an ice cliff scale or do you mean rocks are sliding down large portions of a glacier? At what glacier slope do you think rocks are able to accumulate?*

*P6L4-6: this information on lateral moraines either needs to be cited or the measurements be motivated in the introduction, described in the methods and presented in the results.*

*P6L8-9: Can you please offer support to the statement that "a large percentage of the debris cover is a result of the lithology" What percentage? A glacier surrounded by highly erosive rock at a very low angle might not generate any debris cover.*

*P6L14: In the framework of glaciology, 20-40 m of ice is not "substantial". 20-40 m of ice is likely not deforming internally and could be stagnant rendering it not part of a glacier*

| |
|---|
| *following the classical definition of a glacier.* |
| *P6L17: "DC" not defined.* |
| We deleted all these sentences and instead of this, we provided the new text and figures, please see P7 L12-28, P8 L1-20, and P9 L1-6 |
| *P6L38-41: Are you sure there are not other reasons for a difference between Scherler et al., 2018 and the two points you describe?* |
| As we mentioned, those two are the main reasons for the difference |
| *P7L10-13: A 50% increase in debris cover is not reported in your results. Please clarify how this value is calculated.* |
| We agree and changed this sentence, please see P9 L38-41 |
| *Table 1: For future work that might want to cite this article, it might be convenient for additional rows that give the sum of all of these sub-regions. I believe these are also the values you cite in the article. I also would suggest showing the changes in a time series plot with error bars.* |
| We agree and changed Table 1, please see P5 L1-2 |
| *Figure 1: e,f and g need to be shown in the upper (unlabeled) panel of this figure. This is may be personal preference, but I think the top panel could be a stand-alone location figure with panels e,f,g being their own figure or coupled to a Figure similar to Figure S7.* |
| We changed Fig. 1, please see P3 L14 |
| *Figure 2: According to the text all of the differences between your results and Scherler et al., 2018 is datum shifts and erroneously classified nominal glacier ellipses. Do these sources really explain all of the differences shown in this figure?* |
| We moved the Fig.2 in the supplement as Fig. S6. Using this figure we just show the percentage differences of SDC between of these two study results. More reasons of these two results are shown in Fig. S7 |
| *Figure S1: How did you define debris covered and debris free glaciers? I think this classification should be shown in Figure 1 or elsewhere so it is clear for future work what was considered "debris-covered." What glacier criteria did you apply to classify 0.01-0.05 km^2 land surfaces as independent glaciers? I would be interested to see some of these glaciers along with a satellite image and their debris maps.* |
| We provided an appropriate text about the debris-covered and debris-free glacier definitions, please see P3 L28-34, and glacier size classification P2 L37 |
| *Figure S2: This figure does not provide much if any information to the article and does not fit the scope of the work that was done.* |
| We agree, and deleted this figure |
| *Figure S4: It is not clear what is meant by "semi-automated." The whole approach to mapping debris cover could be called semi-automated, but what non-automated work went into the classification of bare ice alone? The trace of the longitudinal profile needs to be shown in a or b. Dashed line in a and b should probably be defined.* |
| We agree and deleted this figure |
| *Figure S5: Rather than what is essentially a repeated figure S4 in a different location, I would like to see more of the changes. Oblique photography is nice, but here does not offer much information.* |
| We have changes this image, please see Fig. S5, P3 L6 |
| *Figure S6: Panels b and c showing ice thickness measurements have no established relevance to this article. I would suggest removing this figure.* |
| We agree and deleted this figure |
| *Figure S7: I think an altered and expanded upon version of this figure is the key figure of this work and should be in the main article. Changes in glacier area and debris covered area are somewhat difficult to see side by side, I would recommend taking an overlap/transparency approach similar to the following two articles for visually showing changes in glacier and debris-covered areas:* |

Glasser, Neil F., et al. "Recent spatial and temporal variations in debris cover on Patagonian glaciers." Geomorphology 273 (2016): 202-216.

Herreid, Sam, et al. "Satellite observations show no net change in the percentage of supraglacial debris-covered area in northern Pakistan from 1977 to 2014." Journal of Glaciology61.227 (2015): 524-536.

We agree and changed the old figure with the new, please see Fig. 4, P8 L9-11

*Figure S8: It is confusing to discuss "Debris cover outline[s]" as well as "Bare ice outline[s]" as you are using different words to reference effectively the same thing.*

We agree and changed the old figure with the new, please see Fig. 4, P8 L9-11

**Technical corrections**
*P1L30: I think "SDC" should be defined at the first mention of debris cover.*

We agree, please see P1 L37

*P1L33: Remove "the" before "SDC" and "glacier ablation."*

We agree, please see P1 L39

*P1L37: Add "summarized in Kirkbride and.."*

There is no reason to add "summarized"

*P1L40: "The difficulty. . ." I would say "One difficulty.."*

We agree, please see P2 L6

*P2L1: Use SDC consistently if defined, e.g. here: "properties of debris"*

We agree, please see P2 L7

*P2L1: Change to "of a debris layer has"*

We agree, please see P2 L7

*P2L10: Change "as it is similarity in" to "as it is similar to"*

We changes this sentence, please see P2 L16-17

*P2L10: Comma after citation*

We agree, please see P2 L17

*P2L11: Consider changing "key player" to "glacier-wide component"*

We agree, please see P2 L17

*P2L12-13: "as surface mass balance. . .is different from that of bare ice" this has already been established earlier in your introduction and I don't think it needs to be restated.*

We deleted earlier sentence and disagree to delete here as well

*P2L19: Change to "and a recently"*

We changed this sentence, please see P2 L25-27

*P2L29: Change to "as the largest"*

We changed this sentence, please see P2 L35-36

*P2L31-33: Awkward sentence, please break into two*

We agree and changed this sentence, please see P2 L38-40

*P2L35: "imagery from 2016. The SPOT" not clear if one or several images were used*

We changed this sentence, please see P2 L42

*P3L14: I would remove "Relatively heavily"*
*P3L15: add "..to distinguish the glacier boundary" or "glacier terminus"*
*P3L18: change to "..might result in a potential.."*
*P4L1: change to "i) a buffer method"*
*P4L2 "manual"*
*P4L9: correct English in this sentence*
*P4L13: end parenthesis after NSD*
*P6L5: please add "Glacier" after named glaciers, here and throughout the article (or "glaciers" after a list)*

We deleted all these sentences

*P6L30: This is not a "whereas" statement.*

We agree, please see P9 L17

| |
|---|
| *P6L30: "increment" should be "increase."* |
| We changed this sentence, please see P9 L17 |
| *P7L14: "vital" seems like too strong of language to me.* |
| We agree and changed this sentence, please see P10 L1 |

---

## Author Comment (AC2) · 1 Jun 2019

**Authors reply to Dr. Dirk Scherler's comments**

**"Brief communication: Supraglacial debris-cover changes in the Caucasus Mountains" by L. G. Tielidze, et al.**

The Cryosphere Discuss.,
https://doi.org/10.5194/tc-2018-259

Dear Dr. Dirk Scherler,

First of all, we thank you for your careful reading of the paper and for the constructive review. In the following pages, we provide point-by-point responses following every comment.

All corrections and changes what we did in the text are in yellow.

| |
|---|
| **General comments**
This contribution presents satellite imagery derived changes in supraglacial debris cover in the Caucasus Mountains between 1986, 2000, and 2014, based on Landsat and Spot imagery. The paper presents interesting data although I find some of the methodology unclear. The analysis of the data could be extended to support, or refute, some of the inferences in the discussion, which is sometimes rather speculative. When addressing these issues, the paper should be a relevant contribution, but may be better published as a standard format paper, instead of a brief communication? |
| We agree that the methodology section was confusing in the previous version of the manuscript, and because of this, we are presenting much comprehensive methodology here. The analysis of the data has been expended as well. Please find the new methodology section P3 L20-34; P4 L1-24.
Due to many corrections and new text/figures, the manuscript was extended but we decided to keep it as a brief communication. |
| **Main comments**
The description of the methods takes up a considerable fraction of the entire paper, but there still is some information missing. I had the biggest difficulties to understand how the GPR data was used. Hardly any information is given about it, but it appears to be relevant for "correcting" the debris covered area on the Elbrus Massif. If the GPR data was used in this study, I think the authors should provide much more information about the data and the results. At present, I've only seen one figure in the supplementary material (Figure S6). How reliable are the GPR measurements? Is there room for interpretation or was it all straight forward? I don't question the observations; I would just like to see more of the data the authors collected. |
| Due to the GPR data caused some awkwardness, we excluded it in the current version of the manuscript. We agree that it required much more explanation and methodology definitions. |
| Overall, I found the discussion quite confusing. It starts with a chapter on possible reasons for the observed SDC increase, but this chapter also addresses the question of spatial differences in SDC, without any temporal aspect. The authors mention many potential reasons for either spatial differences or temporal changes (and I think many of them are truly meaningful), but they remain speculative. I see potential to address some of these reasons with the current data, but that would require additional analysis. For example, the authors suggest that rock avalanches after 2000 may be one of the reasons why SDC increased more during the period 2000-2014 and they provide examples in the supplementary material. |

Wouldn't it be possible to quantify these in order to assess their relevance? From the supplementary figure, it wasn't clear to me if all of the rock avalanches were deposited in the ablation zone. If not, some of them may be gone again when buried under snow and ice in a few years. The authors also suggest that topographic differences between northern and southern slopes are responsible for spatial differences in debris cover. Instead of opposing just the two regions, wouldn't it make sense to analyze the glaciers and their topographic setting for testing this idea? If true, there should be a correlation between the topographic reason and the observed difference. There exist other inferences or statements ("Little Ice Age moraine can affect the SDC increase on the glacier tongue, as debris often falls from lateral moraines onto the glacier surface"; "in the eastern Greater Caucasus, a large percentage of the debris cover is a result of the lithology") that should be better backed up by observations, for example, from the spatial distribution of debris cover and its increase on the glacier surfaces.

We provided a different version of the Discussion with many corrections and new texts and tried to avoid any speculations. Relevant figures have been provided in the main manuscript and supplement, that shows e.g. up-glacier migrations Fig. 3, S3, and SDC increase after rock-fall (related to permafrost) Fig. S4. etc.

The comparison with previous estimates of debris cover in the Caucasus Mountains is useful and worth reporting. However, when comparing results with the values reported in our recent paper (Scherler et al., 2018), it should be explicitly stated that the scope of our study was a different one. We attempted an automatized global assessment, knowing and discussing the issues of erroneous glacier outlines in the RGI and we explicitly stated that we did not correct any outlines in the RGI. In other words, we did not pretend to get the debris cover correct, if the RGI outlines are not correct. This is an important point that should be acknowledged to avoid making a straw man argument! It is also not clear to me how the data of Figure 2 was put together. Were only those glaciers compared that were analyzed in both studies? And what about the circular glaciers? Overall, I find Figure 2 not relevant. More relevant would be the comparison of the clean ice-debris cover boundary between our and this study, as in our automatized mapping, we were relying on a single threshold value for the entire Earth.

We certainly do not question work by Scherler et al. (2018) and we mentioned (end of the Discussion) that the goal of your study was an automatized global assessment of SDC from optical satellite data, without correcting any outlines in the RGI. The big difference between these two results is just caused by RGI inconsistent outlines (that we mentioned before as well, e.g. Tielidze and Wheate, 2018).

But, in fact, work by Scherler et al. (2018) is only one work including the SDC data-set for entire Greater Caucasus and we think that comparison of these two studies is important. Therefore, we decided to leave previous Fig. 2 but moved it in the supplement as Fig. S6. Also, we provided an explanation of how we compared these two data-set. Please see P9 L23-24.

**Specific comments**
*P1L24: "Thereby": I don't see the causal connection to the foregone sentence*
*P1L26- 28: It would be good if you could provide an explanation, or your favorite explanation, why this is the case.*

We agree and provided new Abstract. Please see P1 L24-34

*P2L2: "supraglacial debris thickness"*

We agree, please see P2 L8

*P2L8: "Europe": Nothing important, but I'm wondering if all of the Caucasus Mountains and thus all of the glaciers and their area are part of Europe?*

We changed this sentence, please see P2 L13

*P2L10: Delete "similarly"*

We agree, please see P2 L16-17

*P2L15: How does it contradict earlier studies? Please specify.*

| |
|---|
| We have changes this sentence, please see P2 L20-22 |
| *P2L25: What determined your selection of glaciers? Does this mean there exist glaciers that you did not consider? Please clarify* |
| We have changed this sentence and added appropriate citations, please see L2 L31-32 |
| *P2L29: "a largest" -> "the largest"* |
| We agree, please see L2 L35 |
| *P2L39: "Additionally, . . ." -> Please specify how you included what kind of GPR results in which way. Also give references if it is published. If it is not published, I think you need to provide much more information on the GPR data.* |
| We excluded the GPR data in this study |
| *P3L1: ASTER GDEM asks for a certain way of acknowledgement that is missing.* |
| We agree and an appropriate link was added P3 L6 |
| *P3L5: What threshold value did you use to distinguish between ice and debris?* |
| We used threshold value ≥2.0, please see P3 L21-22 |
| *P3L13: "SDC is" -> "SDC are"* |
| We deleted this sentence |
| *P3L14: Delete "Relatively"* |
| We deleted this sentence |
| *P3L16: Again, it is unclear how you used GPR data (see comment above).* |
| We excluded the GPR data in this study |
| *Figure 1: I think this figure should be larger – but maybe that's just due to the formatting of the PDF* |
| We have changed Fig. 1, please see P3 L14 |
| *P3L22: "sections" -> "regions"* |
| We agree, please see P3 L15 |
| *P3L23: "Elbrus Massif"* |
| We agree, please see P3 L16 |
| *P4L9: Delete "however" & "critical to" -> "critical for"* |
| We deleted this sentence |
| *P4L10: "performed" -> "used" & Why is method 2 giving you a more realistic uncertainty estimate?* |
| We deleted this paragraph |
| *P4L19: How is the geomorphology complicated?* |
| We deleted this sentence |
| *P4L31: "debris cover" -> "SDC" & comma after "increased"* |
| We agree, please see P4 L33 |
| *P5L8: "Debris cover migrated up-glacier" -> I haven't seen any data on the spatial distribution of the SDC. If you have, it would be worth showing it.* |
| We approved this sentence by providing new text (P6 L15-19) and images (Fig. 3 P7 L6-10 and Fig. S3 P2 L10-14) |
| *P5L10-P6L11: This chapter sounds more like results. Also see the main comments above.* |
| We have changed this chapter, please see new version from P6 L28 to P9 L6 |
| *P6L14: Regarding the GPR results, see main comments above.* |
| We excluded the GPR data in this study |
| *P6L25: Not clear how the cited studies and this one are broadly consistent. In that there is an increase? It appears difficult to compare a regional study with individual glacier studies. Perhaps compare results from this studies with previous ones by limiting to those glaciers that are in common?* |
| We have changed this sentence, please see P9 L9-11 |
| *P6L31: "141% increment"?* |
| We agree, please see P9 L17 |

| |
|---|
| *P6L36-P7L3: See main comments above.* |
| We have changed this paragraph, please see new version P9 L23-35 |
| *P7L10: SCD -> SDC* |
| We agree, please see P9 L38 |
| *P7L15: "periglacial debris cover": this term comes surprising and it's unclear how this conclusion came about. Also, why is the monitoring "vital"?* |
| We have changed this sentence, please see P10 L1-2 |
| *P7L16: Delete "a"* |
| We agree, please see P10 L3-4 |

---

## Author Comment (AC3) · 1 Jun 2019

This response includes the attached ZIP file - corrected and final version manuscript/supplement (pdf files)

Please also note the supplement to this comment:
https://www.the-cryosphere-discuss.net/tc-2018-259/tc-2018-259-AC3-supplement.zip

---

## Editor Comment (EC1) · Etienne Berthier (Editor) · 5 Jun 2019

Dear authors

I read your response to the reviewers and the revised manuscript (MS). I do not think they meet yet the requirements to deserve a full re-review.

1/ The rebuttal letter should be more self-consistent. Currently it does not contain scientifically justified-responses to the referee comments and lacks some explanations. For example, reviewer 1 (Sam Herreid) made some detailed comments about the need to examine a consistent and meaningful spatial domain. Your response to this comment is simply to refer to new text and figures without any scientific explanations. Similarly, later in the rebuttal, you justify the use of a method (the buffer method) by citing three

references from the co-authors of this study. This is not really convincing and you do not justify neither why you "excluded multiple digitization method in this version of manuscript". This is not sufficient for a rebuttal letter that needs to convince the editor/reviewers that all concerns have been properly addressed.

2/ The manuscript is not ready neither. During a quick read, I found several typos, results reported with varying number of decimals, issues with section numbering etc. . . At this stage I expect an almost typo-free manuscript so that the reviewers can focus on the scientific content and how their concerns were addressed.

I have a lot of respect for the time spent by referees to evaluate and improve a paper. I will only send them a revised version if the MS and the rebuttal letter meet the quality level expected in TC.

I will carefully checked that you reply to all comments in an adequate way and that you provide an improved version of the paper before sending it to the referees.

Best regards,

Etienne Berthier

---

## Author Response (AR1)

Dear Dr. Etienne Berthier,

We appreciate your help for the constructive comments which have significantly improved the quality of our manuscript. We have made our best effort to revise the manuscript based on your and the referee's comments and suggestions.

In addition, we have expanded and made more detail comments in the response letters.

Please see the new response letters and the revised manuscript below.

All corrections and changes what we did in the manuscript are in yellow (first revision) and green (second revision).

**Authors reply to Dr. Dirk Scherler's comments**

**"Brief communication: Supraglacial debris-cover changes in the Caucasus Mountains" by L. G. Tielidze, et al.**

The Cryosphere Discuss., https://doi.org/10.5194/tc-2018-259

Dear Dr. Dirk Scherler,

First of all, we thank you for your careful reading of the paper and for the constructive review. In the following pages, we provide point-by-point responses following every comment.

All corrections and changes what we did in the text are in yellow (first revision) and green (second revision).

**General comments**

This contribution presents satellite imagery derived changes in supraglacial debris cover in the Caucasus Mountains between 1986, 2000, and 2014, based on Landsat and Spot imagery. The paper presents interesting data although I find some of the methodology unclear. The analysis of the data could be extended to support, or refute, some of the inferences in the discussion, which is sometimes rather speculative. When addressing these issues, the paper should be a relevant contribution, but may be better published as a standard format paper, instead of a brief communication?

We agree that the methodology section was confusing in the previous version of the manuscript, and therefore, we are presenting more comprehensive methodology here. The analysis of the data has also been expanded. Please find the new methodology section P3 L18-34; P4 L1-21.

It is true, that because of the many corrections and new text/figures, the manuscript was slightly expanded but we decided to keep it as a brief communication, which is better suited to its scope and length. We think that the standard format article should be much more thorough, in-depth study that could contain more detailed data.

Main comments

The description of the methods takes up a considerable fraction of the entire paper, but there still is some information missing. I had the biggest difficulties to understand how the GPR data was used. Hardly any information is given about it, but it appears to be relevant for "correcting" the debris covered area on the Elbrus Massif. If the GPR data was used in this study, I think the authors should provide much more information about the data and the results. At present, I've only seen one figure in the supplementary material (Figure S6). How reliable are the GPR measurements? Is there room for interpretation or was it all straight forward? I don't question the observations; I would just like to see more of the data the authors collected.

As the GPR data caused some awkwardness, we have excluded it in the current version of the manuscript. We agree that it required much more explanation and methodology

**definitions.**

We note that, the removal has no impact of the overall results/message of the paper, since it was a more separate topic and not the main part of the manuscript. In addition it applied only to the Elbrus Massif and not the whole Caucasus.

Overall, I found the discussion quite confusing. It starts with a chapter on possible reasons for the observed SDC increase, but this chapter also addresses the question of spatial differences in SDC, without any temporal aspect. The authors mention many potential reasons for either spatial differences or temporal changes (and I think many of them are truly meaningful), but they remain speculative. I see potential to address some of these reasons with the current data, but that would require additional analysis. For example, the authors suggest that rock avalanches after 2000 may be one of the reasons why SDC increased more during the period 2000-2014 and they provide examples in the supplementary material. Wouldn't it be possible to quantify these in order to assess their relevance? From the supplementary figure, it wasn't clear to me if all of the rock avalanches were deposited in the ablation zone. If not, some of them may be gone again when buried under snow and ice in a few years. The authors also suggest that topographic differences between northern and southern slopes are responsible for spatial differences in debris cover. Instead of opposing just the two regions, wouldn't it make sense to analyze the glaciers and their topographic setting for testing this idea? If true, there should be a correlation between the topographic reason and the observed difference. There exist other inferences or statements ("Little Ice Age moraine can affect the SDC increase on the glacier tongue, as debris often falls from lateral moraines onto the glacier surface"; "in the eastern Greater Caucasus, a large percentage of the debris cover is a result of the lithology") that should be better backed up by observations, for example, from the spatial distribution of debris cover and its increase on the glacier surfaces.

We provided a new version of the Discussion with many corrections and new texts. We tried our best to avoid any speculations.

Overall we provided two sub-titles in Discussion section:

**4.1 Supraglacial debris-cover changes**

- Here we provided SDC percentage distribution by different elevation zones to justify the upglacier migration Fig. 3; and added one more image in the supplement (Fig. S3).

- We also discussed that, dramatic increase of the SDC after 2000 could be caused by the rock avalanches related to permafrost. We gave an example of the Devdoraki Glacier – Fig. S4 and added a new related reference (Tielidze et al., 2019).

- In addition, we added one more relevant figure in supplement (Fig. S5) in order to better understand topographic differences between northern and southern slopes related to SDC different distribution.

More detailed analysis regarding the topographic differences for the Greater Caucasus glaciers, has been already Published in Tielidze et al., 2018 (see reference list in the manuscript).

- In case of Elbrus, we discuss that the significant increase of SDC can be related to

resurfacing of the englacial debris as a result of glacier recession (specifically in the eastern slopes).

In the second subtitle **(4.2 Comparison with previous investigations)** we compared our results with previous studies in local and regional scale.

The comparison with previous estimates of debris cover in the Caucasus Mountains is useful and worth reporting. However, when comparing results with the values reported in our recent paper (Scherler et al., 2018), it should be explicitly stated that the scope of our study was a different one. We attempted an automatized global assessment, knowing and discussing the issues of erroneous glacier outlines in the RGI and we explicitly stated that we did not correct any outlines in the RGI. In other words, we did not pretend to get the debris cover correct, if the RGI outlines are not correct. This is an important point that should be acknowledged to avoid making a straw man argument! It is also not clear to me how the data of Figure 2 was put together. Were only those glaciers compared that were analyzed in both studies? And what about the circular glaciers? Overall, I find Figure 2 not relevant. More relevant would be the comparison of the clean ice-debris cover boundary between our and this study, as in our automatized mapping, we were relying on a single threshold value for the entire Earth.

We certainly do not question work by Scherler et al. (2018) and we mentioned (end of the Discussion) that the goal of your study was an automatized global assessment of SDC from optical satellite data, without correcting any outlines in the RGI. The big difference between these two results is just caused by RGI inconsistent outlines (that we mentioned before as well, e.g. Tielidze and Wheate, 2018).

But, in fact, the work by Scherler et al. (2018) is only one work including the SDC data-set for entire Greater Caucasus and we think that comparison of these two studies and highlighting the resultant overestimation caused by erroneous RGI outlines is important. Therefore, we decided to leave previous Fig. 2 but moved it in the supplement as Fig. S6.

In addition, we provided an explanation of how we compared these two data-set – "We extracted both supraglacial debris cover and clean-ice outlines from Scherler et al. (2018) for our glacier sample to compare these results of our regional study with those from the global study." Please see P9 L24-25.

We also provided a new figure in the supplement (Fig. S7) showing the differences between current study and study by Scherler et al. (2018)/RGI outlines.

Specific comments

P1L24: "Thereby": I don't see the causal connection to the foregone sentence

P1L26-28: It would be good if you could provide an explanation, or your favorite explanation, why this is the case.

We agree. We rewrote the abstract taken your comments into account. Please see P1 L24-34 *P2L2: "supraglacial debris thickness"*

Corrected as suggested, please see P2 L8

P2L8: "Europe": Nothing important, but I'm wondering if all of the Caucasus Mountains and thus all of the glaciers and their area are part of Europe?

Since there are several definitions about the location of the Greater Caucasus, we just mentioned it as a "one of the world's highest mountain systems".

Please see P2 L13

P2L10: Delete "similarly"

We agree and deleted this word, please see P2 L16

P2L15: How does it contradict earlier studies? Please specify.

We have changed this sentence, please see P2 L20-22

P2L25: What determined your selection of glaciers? Does this mean there exist glaciers that you did not consider? Please clarify

We have changed this sentence and added appropriate citations, as follows: "We selected four regions representing different climate conditions (Stokes, 2011) and glacier characteristics (Tielidze and Wheate, 2018) with a total of 659 glaciers:"

Please see P2 L30-31

P2L29: "a largest" -> "the largest"

Corrected as suggested, please see L2 L34

P2L39: "Additionally, . . . " -> Please specify how you included what kind of GPR results in which way. Also give references if it is published. If it is not published, I think you need to provide much more information on the GPR data.

As the GPR data caused some awkwardness, we excluded it in the current version of the manuscript. We think that it required much more explanation and methodology definitions. We note that, the removal has no impact of the overall results/message of the paper, since it was a separate topic and not the main part of the manuscript.

P3L1: ASTER GDEM asks for a certain way of acknowledgement that is missing.

We agree and an appropriate link was added P3 L5

P3L5: What threshold value did you use to distinguish between ice and debris?

We used threshold value ≥2.0. This information was now added in the manuscript, please see P3 L20

P3L13: "SDC is" -> "SDC are"

We deleted this sentence, so this comment is not relevant anymore.

P3L14: Delete "Relatively"

We agree and deleted this word

P3L16: Again, it is unclear how you used GPR data (see comment above).

As the GPR data caused some awkwardness, we excluded it in the current version of the manuscript. We think that it required much more explanation and methodology definitions. We note that, the removal has no impact of the overall results/message of the paper, since it was a separate topic and not the main part of the manuscript.

Figure 1: I think this figure should be larger – but maybe that's just due to the formatting of the PDF

**We have changed Fig. 1, please see P3 L12**

P3L22: "sections" -> "regions"

**Improved as suggested, please see P3 L13**

P3L23: "Elbrus Massif"

Corrected as suggested, please see P3 L14

P4L9: Delete "however" & "critical to" -> "critical for"

*P4L10: "performed" -> "used" & Why is method 2 giving you a more realistic uncertainty estimate?*

We used Google Earth software/images instead of the multiple digitization for uncertainty

estimation as a second method. The main reason for this decision was the high-resolution imagery from Google Earth that allowed us much precise uncertainty estimation. Thus, we deleted this paragraph and provided a new one. Please see P4 L7-11

P4L19: How is the geomorphology complicated?

We deleted this sentence because the differences in geomorphology of these parts have no impact on the debris-covered glaciers in this area

*P4L31: "debris cover" -> "SDC" & comma after "increased"*

Corrected as suggested, please see P4 L31

P5L8: "Debris cover migrated up-glacier" -> I haven't seen any data on the spatial distribution of the SDC. If you have, it would be worth showing it.

We approved this sentence by providing the new text (P6 L15-19) and images (P7 Fig. 3). In the supplement P2 L12-14; Fig. S3, showing the up-glacier migration evidence.

P5L10-P6L11: This chapter sounds more like results. Also see the main comments above.

We have changed this chapter, please see specific comments how exactly we improved it.

**4.1 Supraglacial debris-cover changes**

- Here we provided SDC percentage distribution by different elevation zones to justify the upglacier migration Fig. 3; and added one more image in the supplement (Fig. S3).

- We also discussed that, dramatic increase of the SDC after 2000 could be caused by the rock avalanches related to permafrost. We gave an example of the Devdoraki Glacier – Fig. S4 and added a new related reference (Tielidze et al., 2019).

- In addition, we added one more relevant figure in supplement (Fig. S5) in order to better understand SDC different distribution between northern and southern slopes related to topographic differences.

More detailed analysis regarding the topographic differences for the Greater Caucasus glaciers, has been already Published in Tielidze et al., 2018 (see reference list in the manuscript).

- In case of Elbrus, we discussed that the significant increase of SDC can be related to resurfacing of the englacial debris as a result of glacier recession (specifically in the eastern slopes).

- In addition, we discussed that shielding effect is not enough to offset the retreat trend in the Greater Caucasus.

Overall, please see new version of the chapter from P6 L28 to P9 L6

P6L14: Regarding the GPR results, see main comments above.

As the GPR data caused some awkwardness, we excluded it in the current version of the manuscript. We think that it required much more explanation and methodology definitions. We note that, the removal has no impact of the overall results/message of the paper, since it was a separate topic and not the main part of the manuscript.

P6L25: Not clear how the cited studies and this one are broadly consistent. In that there is an increase? It appears difficult to compare a regional study with individual glacier studies. Perhaps compare results from this studies with previous ones by limiting to those glaciers that are in common? We have changed this sentence, and mentioned that our result is in good agreement with general picture of other studies in this region, resulting the SDC increase in recent decades. Although, in the beginning of this chapter we also stated that direct comparisons of our study with previous investigations are difficult because most of them cover only a relatively small area.

Please see new version of these sentences P9 L9-11

P6L31: "141% increment"?

We have corrected this sentence"

"The debris layer became thicker and larger at some points near the terminus between 1983 and 2010, and the volume of the lithogenic matter over the whole glacier increased by ~140%."

Please see P9 L17-18

P6L36-P7L3: See main comments above.

Since the work by Scherler et al. (2018) is only one work including the SDC data-set for entire Greater Caucasus we think that comparison of these two studies and highlighting the resultant overestimation caused by erroneous RGI outlines is important.

Thus, we provided an explanation of how we compared these two data-set – "We extracted both supraglacial debris cover and clean-ice outlines from Scherler et al. (2018) for our glacier sample to compare these results of our regional study with those from the global study." Please see P9 L24-25.

We also provided a new figure in the supplement (Fig. S7) showing the clear differences between current study and study by Scherler et al. (2018)/RGI outlines.

Once again we emphasize that we do not question work by Scherler et al. (2018), and this is clearly state in the manuscript where we mentioned that the goal of study by Scherler et al. (2018) was an automatized global assessment of SDC from optical satellite data, without correcting any outlines in the RGI. The big difference between these two results is just caused by RGI inconsistent outlines.

Please see P9 L35-37.

Overall, we have changed this paragraph, please see new version P9 L24-37

P7L10: SCD -> SDC

Corrected, please see P9 L40

P7L15: "periglacial debris cover": this term comes surprising and it's unclear how this conclusion came about. Also, why is the monitoring "vital"?

We have changed this sentence as following: "Given the increasing degree of supraglacial debris cover in the Greater Caucasus region, it is worthwhile to maintain its monitoring, as it constitutes an important control on glacier response to climate change".

Please see P10 L4-5

P7L16: Delete "a"

Deleted, please see P10 L7

**Authors reply to Dr. Sam Herreid's comments**

**"Brief communication: Supraglacial debris-cover changes in the Caucasus Mountains" by L. G. Tielidze, et al.**

The Cryosphere Discuss., https://doi.org/10.5194/tc-2018-259

Dear Dr. San Herried,

Thank you very much for your detail comments which we help to increase the quality of our manuscript. Please find in the following a point-by-point reply to your review.

All corrections and changes what we did in the text are in yellow (first revision) and green (second revision).

**General Comments**

This paper presents changes in glacier and debris-covered area for several subregions in the Greater Caucasus mountains. The results are accompanied by an error analysis that uses two approaches to quantify mapping error. This work is relevant both by expanding the spatial domain over which debris-covered area changes are measured and by providing a comparison against larger, global scale debris cover mapping efforts. Overall, I think there are methodological deficiencies that need to be addressed, figures that need to be both added and removed and a general improvement in the clarity of the writing.

We agree that methodological deficiencies were one of the main issues of the first version of the manuscript. Considering this, we provided a new chapter of methodology, with a more detailed description.

- In the beginning we described the clean-ice outline delineating process with widely used band ratio segmentation method (RED/SWIR; Landsat OLI 4/6 or TM 3/5 with a threshold of  $\geq$ 2.0) and intensive manual improvements (removed misclassified areas, e.g. snow, shadows). In the end of the first paragraph, we mentioned our approach that "the supraglacial debris cover was classified as the residual between a semi-automatically derived clean-ice map and a manually improved glacier extent map", similar to the Paul et al. (2004). Supraglacial debris cover was extracted and saved as separate layers.

- Based on Global Land Ice Measurements from Space (GLIMS) Glacier Classification Guidance (Rau et al., 2005), we gave a more explanation in order to better understand the definitions of debris-covered and debris-free glaciers.

- for uncertainty estimation we used:

1. The buffer method;

For clean ice we used a 15 m (1/2 pixel) buffer (Bolch et al., 2010)

For debris-covered parts 60 m (two pixels) (Frey et al., 2012)

2. High resolution Google Earth imagery;

3. GPS measurement data which >1200 points

- regarding to all these changes several figures were added/deleted as well. e.g.

Fig. 2. Showing the increase of supraglacial debris cover according to the all selected regions and entire Greater Caucasus for 1986, 2000 and 2014.

Fig. 3. Showing the hypsometrical distribution of supraglacial debris cover, clean ice and total glacier area in 1986 and 2014.

Fig. 4. Showing the hypsometrical debris cover increase on the Elbrus Massif from 1986 to 2014.

Fig. S1. Showing the examples of glacier outline accuracy assessment by GPS measurements.

Fig. S3. Showing the example of the supraglacial debris cover up-glacier migration onto the Khalde Glacier.

Fig. S4. Showing the example of the supraglacial debris cover increase onto the Devdoraki Glacier after rock-ice avalanche in 2014. Possible related to permafrost.

Fig. S5. Showing the comparison of supraglacial debris cover and clean-ice area distribution between the southern and northern slopes.

**- Figure. 1 e, f, g, and Figure S2, S3, S4, S5, S6, S7, were deleted.**

A key component of any study investigating debris-covered area change is a consistent and meaningful spatial domain. Transient snowfall (possible at any time of year) can cover debris resulting in an underestimation of debris cover that is actually present in a glaciers ablation zone. If a later map of debris cover is generated from an image with a higher snowline, a false debris-covered area change signal will be measured, even in a setting where the position of the equilibrium line is stable. In order to eliminate these errors, a spatial domain can be set at the aggregate lowest minimum snowline from all of the images used to map debris cover. Tracking of the up-glacier migration of debris-covered area changes mapped in this study were well below snowline, then showing this will negate the concern. If mapped debris-cover shared a boundary with snow rather than ice or firn, I do not think debris area change measurements can be trusted without more information.

We provided new figures that shows up-glacier migration, please see Fig. S3 (in supplement), where the SDC is well below to the snow line. In addition, we provided a new Fig. 3 (please see P7) that shows SDC vertical distributions, that approves SDC increase in upper elevations.

We also mentioned that the similar pattern of up-glacier migration was detected on Tasman Glacier, New Zealand (Kirkbride and Warren, 1999), and on Zmuttgletscher Glacier, Swiss Alps (Mölg et al., 2019).

Please see P7 L1-12

One of the two approaches used in this study to estimate mapping errors is a buffer method which I do not think is sufficiently supported to meaningfully quantify error. I would like to see some evidence supporting the two buffer distances that were selected.

Further, it is unclear in the results presented with error bounds which approach they are derived from or if the two approaches are in some way combined. It seems feasible to use the

detailed manual error assessment at six glaciers to calibrate a more meaningful buffer approach applied to the entire study area, but I do not believe this was done.

- For more evidence of error assessment we used not just the buffer method but also Google Earth images. According to the buffer method the uncertainty was ~4.1% for clean-ice and ~6.3% for debris-covered ice, while the according to the high resolution Google Earth images it was ~3.4% and ~5.2% respectively. A comparison of these two approaches show a good agreement.

In addition the buffer method is widely used and adopted for mapping errors of debriscovered/debris-free glaciers in many studies (e.g. Frey et al., 2012; Shahgedanova et al., 2014; Khromova, et al., 2014; etc.), that allow us to use it in current study.

- Frey, H., Paul, F., and Strozzi, T.: Compilation of a glacier inventory for the western Himalayas from satellite data: methods, challenges, and results, Remote Sens. Environ., 124, 832–843, 2012.
- Shahgedanova, M., Nosenko, G., Kutuzov, S., Rototaeva, O., and Khromova, T.: Deglaciation of the Caucasus Mountains, Russia/Georgia, in the 21st century observed with ASTER satellite imagery and aerial photography, The Cryosphere, 8, 2367–2379, https://doi.org/10.5194/tc-8-2367-2014, 2014.
- Khromova, T., Nosenko, G., Kutuzov, S., Muravievand, A., and Chernova, L.: Glacier area changes in Northern Eurasia, Environ. Res. Lett., 9, 015003, doi:10.1088/1748-9326/9/1/015003, 2014.
- In in the results the error bounds are derived from buffer method. Please see P5 L2

- Since we used high-resolution imagery from Google Earth that allowed us more precise uncertainty estimation than multiple digitization, we excluded it as a second method.

- Even though many attempts have been made from various studies, there is not yet an ideal method for estimating mapping errors for debris-covered glaciers. Thus, any method that used in a various studies can be critical with many reasons.

**- Overall, please see new paragraphs from P3 L32 to P4 L11**

An example of where article clarity could be improved is in the description of the debris cover mapping methods. The methods section is somewhat confusing to follow, yet follows the widely used approach of finding the residual of bare-ice area classified with a band ratio threshold and manual debris cover outlines. The threshold(s) used should also be stated for future studies that might want to repeat/continue this work. An additional focusing of the article is needed to address/remove results and figures that are not supported with motivation in the introduction or methods (e.g. ice thickness measurements).

We agree that it was somewhat confusing. Considering this we provided a new description (including the threshold values) of all steps of the methodology. We deleted all extra figures that were not related to new version of the manuscript. Overall, the new methodology section follows as:

- In the beginning we described the clean-ice outline delineating process with widely used band ratio segmentation method (RED/SWIR; Landsat OLI 4/6 or TM 3/5 with a threshold of  $\geq$ 2.0) and intensive manual improvements (removed misclassified areas, e.g. snow, shadows). In the end of the first paragraph, we mentioned our approach that "the supraglacial debris

cover was classified as the residual between a semi-automatically derived clean-ice map and a manually improved glacier extent map", similar to the Paul et al. (2004). Supraglacial debris cover was extracted and saved as separate layers.

- Based on Global Land Ice Measurements from Space (GLIMS) Glacier Classification Guidance (Rau et al., 2005), we gave a more explanation in order to better understand the definitions of debris-covered and debris-free glaciers.

- for uncertainty estimation we used:

1. The buffer method;

For clean ice we used a 15 m (1/2 pixel) buffer (Bolch et al., 2010)

For debris-covered parts 60 m (two pixels) (Frey et al., 2012)

2. High resolution Google Earth imagery;

3. GPS measurement data which >1200 points

Specific comments

P1L20-21: Is it a fact that debris coverage typically increases with shrinking glaciers? I would think this is more of a hypothesis that studies like this will either support or reject.

P1L23: "throughout" or "across" rather than "different regions" might give more information to the reader, better still would be the fraction of the total glacier area that you consider.

P1L25: I think "-0.52% yr^-1"

P1L25: Is glacier area change a result from this work or a result from previously published work?

P1L25: This is not a "Thereby" statement.

P1L25-26: northern and southern slopes of what? Unclear if reading only the abstract.

P1L26-28: The last sentence of the abstract is unclear, unsupported in the text and should be removed.

We completely rewrote the abstract taking these comments into account. Please see new version P1 L24-34

P1L34: considered to be significant by whom?

P1L34: Isn't the debris cover generally a passive element in a sediment transport system? The sediment is of course a significant part of a sediment transport system, but its role in the efficiency isn't clear to me.

We have changed this sentence as follows: "It is relevant not only from its impact on glacier ablation but also because it is an important part of the sediment transport system (supraglacial, englacial, and subglacial) in cold and high mountains, which ultimately affect the overall dynamics, and energy mass balance of the glaciers."

Please see P1 L39-40

*P1L39: "exact evaluation," do you mean precise or accurate?*

We have changed this sentence as follows: "For regions where the local population is dependent on glacial meltwater supply, detailed knowledge of glacial hydrology is important to ensure the sustainable use of water resources (Baraer et al., 2012)."

Please see P2 L4-5

P2L2-3: "methods for satellite mapping of supraglacial debris remain in development (Zhang et al., 2016)" Do you mean debris thickness, debris-covered area or both? This statement might need additional reference(s).

We meant the thickness of debris, and changed this sentence as follows: "Field measurement of debris layers have practical difficulties on a large scale, and methods for estimating supraglacial debris thickness using remote sensing remain in development (Zhang et al., 2016)".

Please see P1 L7-9

P2L4: "Several studies" but cite one, add "e.g." or other citations.

We added the second reference here: "Several studies have also reported the role of debris cover in promoting the formation of supraglacial lakes (Thompson et al., 2016; Jiang et al., 2018),"

Please see P2 L9-10

P2L6-7: This sentence should be restructured to make clear SDC is one of the complexities in the relation between climate and glacier mass budget.

We simplified this sentence as follows: "Therefore, it is necessary to take supraglacial debris cover into account when assessing temporal change of mountain glaciers."

Please see P2 L11-12

P2L9-10: add a citation for how we know SDC is an important control for ice ablation.

We have changed this sentence as suggested: "In the Greater Caucasus, supraglacial debris cover is an important control for ice ablation (Lambrecht et al., 2011), and a component in glacier mass balance (Popovnin and Rozova, 2002)."

Please see P2 L15-17

P2L14: What does "SDC is abundant" mean? Make this statement in objective relative terms or merge with next sentence on P2L15.

We specified this sentence as follows: "A recent global study (Scherler et al., 2018) suggests that supraglacial debris cover is abundant in the Caucasus and Middle East (more than 25% glacier area) and that this region shows the highest percent of supraglacial debris cover worldwide."

Please see P2 L18-21

P2L15: Can you please specifically state the contradiction? Did earlier studies claim the Caucasus and Middle East region did not have the highest percent debris coverage worldwide?

We have changed these sentences as follows: "Earlier studies indicated lower relative supraglacial debris cover in the Greater Caucasus but extensive in smaller regions or individual glaciers (Stokes et al., 2007; Lambrecht et al., 2011; Popovnin et al., 2015)." Please see P2 L21-23

P2L18-19: This might not need to be changed but what a "region" is isn't very clear.

P2L18-21: This once sentence paragraph needs to be rewritten, further, I don't think a discussion of controlling factors is adequately discussed to be mentioned here. "in light of" isn't clear scientific language and partly suggests there might be some global context when really a global product is sampled to match your spatial domain.

We rewrote this paragraph as suggested:

"Based on a recently published glacier inventory (Tielidze and Wheate, 2018), we present the first regional assessment of the spatial distribution of supraglacial debris cover and related glacier changes between 1986, 2000 and 2014 for the Greater Caucasus." Please see P2 L24-26

P2L25: Did you select these glaciers individually or did you select whole regions?

P2L25: could you add a citation or a sentence and a citation describing what differences in climate conditions we can have in mind while reading this article?

We changed this sentence and cited appropriately: "We selected four regions representing different climate conditions (Stokes, 2011) and glacier characteristics (Tielidze and Wheate, 2018) with a total of 659 glaciers:"

**Please see P2 L30-31**

P2L34-35: "glacier margins digitized manually" I am confused, I thought the glacier outlines are taken from Tielidze and Wheate, 2018. Could you please make it very clear what data exist previously, what work was done for this study, and if the quality/data timing was not sufficient in earlier work, what alterations were made?

We changed and specified this sentence as follows: "Other datasets used in this study include the "Greater Caucasus Glacier Inventory" manually mapped dataset (Tielidze and Wheate, 2018)".

Please see P3 L9-10

P2L37: Again, it is unclear if mapping glaciers is an objective of this study.

We corrected this sentence as follows: "The Landsat scenes served as a basis for supraglacial debris cover assessment while the SPOT image was used for corrections of supraglacial debris cover areas of Elbrus."

Please see P2 L42 - P3 L1

P2L39: "All imagery was captured from the 28th of July to the 12th of September." Why? This sentence is unclear and unrelated to the following sentence. My guess is that this the argument used for not considering seasonal snowline (see main comment above)?

We explained this question: "All imagery was captured from the 28th of July to the 12th of September, when glacier tongues were mostly free of seasonal snow under cloud-free conditions."

Please see P3 L2-3

P2L40: This is not a sufficient explanation of GPR data acquisition and processing and these data have not been motivated in the introduction.

As the GPR data caused some awkwardness, we excluded it in the current version of the manuscript. We think that it required much more explanation and methodology definitions.

We note that, the removal has no impact of the overall results/message of the paper, since it was a separate topic and not the main part of the manuscript.

P3L4: This is a very confusing title and I'm not so sure if there is a comparison described in this section.

We have changed the title to simply "Methods", please see P3 L18

P3L5-12: The framework of written "steps" is ineffective here. For example the "then" on P3L8 implies a 3rd step but is not called as such. I think there are more than two clear steps and therefore suggest restructuring the presentation of information.

P3L6: I believe you identified "clean-ice", not "clean-ice glaciers."

- P3L6-7: I don't think it is useful to the reader to know about data formats (raster polygons and vector data)
- P3L7: I think it is better to say "removed misclassified area" rather than "deleted misclassified polygons" If a polygon was half correct would you still delete it? It was unclear in earlier sections that mapping glaciers was an objective of this study, it seemed like that task was complete and now the debris cover would be found as a residual from identifying bare ice only. Does this mean that Tielidze and Wheate, 2018 did not consider debris cover and therefore significantly underestimated glacier area?
- P3L8: "as accurately as possible" this is not meaningfully to the reader, please be more specific.
- P3L9: can you please clarify what you are assessing here? Did you classify thin medial moraines as debris covered instead of bare ice? I applaud this effort to consider medial moraines below the detection limit but a drawback of this is your end results become more difficult to reproduce in future comparison studies.
- P3L10-12: This sentence is a bit awkward and confusing when really you are applying a very common technique used to map debris cover. A more simplistic description is "debris cover is classified as the residual between an automatically derived bareice map and a manually generated glacier extent map." With a citation usually to: Paul, Frank, Christian Huggel, and Andreas Kääb. "Combining satellite multispectral image data and a digital elevation model for mapping debris-covered glaciers." Remote sensing of Environment 89.4 (2004): 510-518.

All these questions are related to the first paragraph of old version of manuscript. We have completely changed and provided new paragraph with more clear steps.

- In the beginning we described the clean-ice outline delineating process with widely used band ratio segmentation method (RED/SWIR; Landsat OLI 4/6 or TM 3/5 with a threshold of ≥2.0) and intensive manual improvements (removed misclassified areas, e.g. snow, shadows). Then, we mentioned our approach that "the supraglacial debris cover was classified as the residual between a semi-automatically derived clean-ice map and a manually improved glacier extent map", similar to the Paul et al. (2004). In the end of the first paragraph, we also mentioned that supraglacial debris cover was extracted and saved as separate layers and calculated the area of supraglacial debris cover for individual glaciers

Overall, please see new version of the paragraph P3 L19-25

P3L13: The difficult boundaries are not clearly explained. You list SDC, moraines and debris in shadow, in my opinion there should be no boundaries between these three. Do you mean moraines off the glacier? The writing of this list is also awkward.B

We deleted this sentence because it is no longer associated with this paragraph. This does not affect the content of the methodology section.

P3L15: I would be interested to know if the glacier edge picked from very high resolution agreed (independently) with your GPS measurements. In other words, Between Landsat derived outlines and field measurements, how much aid is heightened resolution?

- P3L16-17: This GPS data sounds very useful for validation of this work but if you are going to present it you will need to describe the sensor, field and processing methods, time of acquisition and location. Are you referring to the one point in Figure S4? Can you convince us readers that being in the field actually enables locating the terminus position better than high resolution imagery? I think for some glaciers this is true but for others it is so unclear that an aerial perspective is the best for outlining a glacier's edge.
- P3L16: One-half pixel is not a helpful unit of measurement, please present in meters and describe how this error "[was] assumed."
- P3L17-19: Cases of uncertainty are very nice for the reader if they are shown in an example. E.g. in your Figure 1 it would help us understand the limitations you encountered to have one of the examples be at a location of uncertainty. This may also help inspire future work to solve the difficulties faced here.

All these questions are related to old version of GPS measurement. In this contents we provided new data and more explanation of GPS measurement. In addition we provided new image (Fig. S1) showing an examples of glacier outline accuracy assessment by GPS measurements.

- For extra uncertainty assessment we used GPS (Garmin 62stc) measurement data which included glacier margins (>1200 points) with horizontal accuracy from ±4 to ±10 m, obtained during field investigations in 2014. In total seven glaciers (Ushba, Chalaati, Lekhziri, Adishi, Shkhara, Zopkhito, Kirtisho) were surveyed. Fig. S1 shows the results of comparison between GPS measurements and Landsat based supraglacial debris cover /clean ice outlines. The accuracy is ±30 m for supraglacial debris cover and ±15 m for clean ice.

**Please see P4 L12-17 and P1 Fig. S1 in supplement**

P4L2: "a sample of manually digitization" corrected to "manual digitization" news a few more details to link to uncertainty estimation.

We deleted this sentence because it is no longer associated with this paragraph. This does not affect the content of the methodology section.

P4L4: Use meters rather than pixel for units.

**Corrected as suggested, please see P3 L33-34**

P4L4: How did you pick these buffer values? At P4L22 you cite an article reporting "five pixels" of error, does this article also support your using 2? Or 1?

For clean-ice we used buffer values from Bolch et al., (2010) and for debris-covered ice, from Freay et al., (2012), We corrected these sentences as follows: "The buffer method (Granshaw and Fountain, 2006) was used for uncertainty estimation for both clean ice and debris-covered glacier parts. For clean ice we used a 15 m (1/2 pixel) buffer (Bolch et al., 2010) and for debris-covered parts 60 m (two pixels) (Frey et al., 2012)."

Corrected as suggested, please see P3 L32-34

- P4L5-6: "an average ratio between the original glacier areas and the areas with a buffer increment." It is unclear what is meant by original. This is also stated as a singular average, are you considering debris and bare ice separately? Do you include bare-ice/debris boundaries internal to the glacier? If so are you double counting error at these locations?
- P4L7: I don't think the percentage error should be a function of area. I also would anticipate errors to be larger for earlier sensors and improve with the higher radiometric resolution of Landsat 8.

We deleted this sentence because it is no longer associated with this paragraph. This does not affect the content of the methodology section.

- P4L10: If a method does not produce realistic results, as stated, you should not include it! This is anyhow an interpretation that belongs in the discussion. The two methods also have various strengths and weaknesses, there is an advantage to an error estimate that considers the whole area rather than six glaciers only. However, I do not see much value in the buffer method error estimates.
- P4L9-16: It's not clear if we are only talking about the outline of the glacier or the outline of the glacier and the outline of the debris. Also here you establish an unspecified classification "debris covered glacier." What is a debris covered glacier? What criteria did you make this classification on? Was it an automated or manual classification? Is there a physical or processed based motivation for this Boolean classification? A figure showing examples of the error analysis should be at least in the SI.

- In this case we gave priority to buffer method rather than multiple digitization, as we had an advantage to estimate uncertainty by using this. Considering this we replaced the multiple digitizations by Google Earth imagery that allowed us more precise uncertainty estimation. According to the buffer method the uncertainty was ~4.1% for clean-ice and ~6.3% for debriscovered ice, while the according to the high resolution Google Earth images it was ~3.4% and ~5.2% respectively. A comparison of these two approaches show a good agreement. Please see P4 L7-11

- Regarding to the debris-covered glacier definition. Based on Global Land Ice Measurements from Space (GLIMS) Glacier Classification Guidance (Rau et al., 2005), we gave a more explanation in order to better understand the definitions of debris-covered and debris-free glaciers.

Please see P3 L26-31

P4L17-19: The GPR data looks very interesting but it is not appropriately developed in this article. It is unclear if this is new work done for this paper or existing work presented in a different publication. If it is new and being presented here first there needs to be motivation in the introduction, methods and stand-alone results. If citing existing work, I don't think it is necessary to have Figure S6, and each statement regarding GPR work should be appropriately cited. A GPR trace that shows an ice thickness of zero off glacier that transitions to a non-zero ice thickness under debris is very interesting and relevant for glacier and debris mapping, however, as it is now, Figure S6 is beyond the scope of this article.

As the GPR data caused some awkwardness, we excluded it in the current version of the manuscript. We think that it required much more explanation and methodology definitions. We note that, the removal has no impact of the overall results/message of the paper, since it was a separate topic and not the main part of the manuscript.

- P4L27: Please clarify what you mean by "a significant increase." Do you mean within statistical significance there is a change (this would be the most meaningful use of the word) or do you mean you consider the amount of increase to be significant based on some unstated prior understanding? Considering your upper estimate for 1986 (12.6% debris-covered) and your lower estimate for 2014 (11.6 debris-covered), your results imply a decrease in SDC. Considering this, your results do not show a significant change.
- P4L28: I do not think it is established that debris area changes are concomitant with glacier area change.

We changed this sentence as follows: "We found an absolute increase of supraglacial debris cover for all investigated glaciers from 48.3±3.1 km2 in 1986, to 54.6±3.4 km2 in 2000 and 79.0±4.9 km2 in 2014, in contrast with a reduction of the total glacier area." Please see P4 L24-27

P4L32: If you have solved for errors numerically, why are you using a tilde here? (comment

extends also to the abstract).

We deleted all tildes in the manuscript

P5L8: The up-glacier migration is not shown in any figures or presented in the results, please include this along with evidence that is not due to seasonal snow variability. Showing that all mapped debris cover is well below the seasonal snow line is sufficient. If, however, the debris cover extends to the seasonal snowline convincing the reader the signal is up-glacier migration will become more difficult but it is essential to make any statement about up-glacier migration of mapped debris cover.

We provided new figures that shows up-glacier migration, please see Fig. S3 (in supplement), where the SDC is well below to the snow line. In addition, we provided a new Fig. 3 (please see P7) showing SDC vertical distributions, that approves SDC increase in upper elevations. Please see the appropriate text as well, P6 L15-19

P5L12 (and Figure S2): A image showing the glacier before and after rock avalanche deposits would make this point much more clear. I would like to see some quantification of "dramatically increased SDC" or it is not adding information. Mapping and quantifying the area of SDC from rock avalanches would add a nice additional dimension to this

article without requiring much extra work and might help you address the title of this section "SDC increase possible reasons" which should be rewritten as "Possible reasons for an increase in SDC."

**We provided new Fig. S4 (in supplement) showing SDC increase after rock avalanche**

P5L14: "..recently for some glaciers" I believe the reference you cite considers one glacier not several.

- P5L16: "the reduction of glaciers is mainly at the expense of clean ice" This is both unclear and possibly not correct. Are you talking about changes in x,y or z? Please defend this statement if you elect to keep it here.
- P5L22-23: Does local mean at an ice cliff scale or do you mean rocks are sliding down large portions of a glacier? At what glacier slope do you think rocks are able to accumulate?
- P6L4-6: this information on lateral moraines either needs to be cited or the measurements be motivated in the introduction, described in the methods and presented in the results.
- P6L8-9: Can you please offer support to the statement that "a large percentage of the debris cover is a result of the lithology" What percentage? A glacier surrounded by highly erosive rock at a very low angle might not generate any debris cover.
- P6L14: In the framework of glaciology, 20-40 m of ice is not "substantial". 20-40 m of ice is likely not deforming internally and could be stagnant rendering it not part of a glacier following the classical definition of a glacier.

P6L17: "DC" not defined.

We think that all these sentences were more confusing, rather than relevant to the manuscript. Considering this, we deleted all these sentences and instead of, we provided the new text and figures.

- In the beginning of the Discussion section we mentioned that supraglacial debris cover increase process became more pronounced after 2000. The up-glacier migration of the upper limit of supraglacial debris cover could be a response to glacier retreat thinning and reduced mass flux as described by Stokes et al. (2007) and defined as 'backwasting' by Benn and Evans (1998). We also mentioned that the similar pattern of up-glacier migration was detected on Tasman Glacier, New Zealand (Kirkbride and Warren, 1999), and on Zmuttgletscher Glacier, Swiss Alps (Mölg et al., 2019).

Please see P7 L1-12

- In the second paragraph, we discussed that the reduction of the clean glacier area  $(18.7\pm4.1\%)$  between 1986 and 2014) appears to be attributable to both glacier retreat and an increase in total supraglacial debris cover. In addition, we mentioned that this finding is supported by field measurements on Djankuat Glacier, which indicate that supraglacial debris cover area increased from 2% to 13% and become thicker between 1968 and 2010 during glacier retreat (Popovnin et al., 2015).

Please see P7 L14-18

- In the next paragraph, we expressed our opinion that the rock avalanches happened after 2000 on some glaciers could be related to permafrost. We provided Figure S4 that shows the SDC increase onto the Devdoraki Glacier before and after rock-ice avalanche. Please see P7 L19-22

- In case of Elbrus, the most significant increase of supraglacial debris cover occurred on the eastern oriented glaciers of Elbrus, can be explained by resurfacing of the englacial debris as a result of glacier recession. In fact, these glaciers are characterized by the highest thinning rates in recent years (Kutuzov, et al., 2019).

Please see P8 L6-9

- In the last paragraph of Discussion, we mentioned that the shielding effect of the increased supraglacial debris cover at the glacier surface in the Greater Caucasus is not enough to offset the retreat trend, but preventing by more rapid retreat trend.

Please see P8 L16-20; P9 L1-6

P6L38-41: Are you sure there are not other reasons for a difference between Scherler et al., 2018 and the two points you describe?

As we mentioned, these differences can mostly be explained by these two main reasons.

P7L10-13: A 50% increase in debris cover is not reported in your results. Please clarify how this value is calculated.

We provided more results in Table 1 (P 5), where we show the sum of debris-covered, cleanice and total glacier area by individual sections of the study area. Also we provided a new figure (Fig. 2. P6) that is related the results of this study.

In addition we changed first sentence of the Conclusion section.

Please see P9 L41; P10 L1-3

Table 1: For future work that might want to cite this article, it might be convenient for additional rows that give the sum of all of these sub-regions. I believe these are also the values you cite in the article. I also would suggest showing the changes in a time series plot with error bars.

We agree and changed Table 1, please see P5 L1-2. In addition we created the new Fig. 2 showing the supraglacial debris cover change according to the different years.

Figure 1: e,f and g need to be shown in the upper (unlabeled) panel of this figure. This is may be personal preference, but I think the top panel could be a stand-alone location figure with panels e,f,g being their own figure or coupled to a Figure similar to Figure S7.

We changed Fig. 1, please see P3 L12

Figure 2: According to the text all of the differences between your results and Scherler et al., 2018 is datum shifts and erroneously classified nominal glacier ellipses. Do these sources really explain all of the differences shown in this figure?

We moved the Fig.2 in the supplement as Fig. S6. Using this figure we show the percentage differences of SDC between these two study results; specific examples are clearly shown in Fig. S7

Figure S1: How did you define debris covered and debris free glaciers? I think this classification should be shown in Figure 1 or elsewhere so it is clear for future work what was considered "debris-covered." What glacier criteria did you apply to classify 0.01-0.05 km2 land surfaces as independent glaciers? I would be interested to see some of these glaciers along with a satellite image and their debris maps.

- We used Global Land Ice Measurements from Space (GLIMS) Glacier Classification Guidance (Rau et al., 2005), in order to better understand the definitions of debris-covered and debris-free glaciers.

Please see P3 L26-31,

For glacier size classification and surface are calculation we used Paul et al., (2009) (see in the reference list).

**Please see P2 L36**

Figure S2: This figure does not provide much if any information to the article and does not fit the scope of the work that was done.

**We agree, and this figure was deleted**

Figure S4: It is not clear what is meant by "semi-automated." The whole approach to mapping debris cover could be called semi-automated, but what non-automated work went into the classification of bare ice alone? The trace of the longitudinal profile needs to be shown in a or b. Dashed line in a and b should probably be defined.

We deleted this figure because it is no longer associated with this manuscript.

Figure S5: Rather than what is essentially a repeated figure S4 in a different location, I would like to see more of the changes. Oblique photography is nice, but here does not offer much information.

We have changed this image that clearly shows supraglacial debris cover and clean-ice area distribution in 1986-2014 for the southern and northern-facing glaciers.

**Please see Fig. S5 (P3 L6 in supplement).**

*Figure S6: Panels b and c showing ice thickness measurements have no established relevance to this article. I would suggest removing this figure.*

We deleted this figure, since we excluded GPR measurement. See comments above.

Figure S7: I think an altered and expanded upon version of this figure is the key figure of this work and should be in the main article. Changes in glacier area and debris covered area are somewhat difficult to see side by side, I would recommend taking an overlap/transparency approach similar to the following two articles for visually showing changes in glacier and debris-covered areas:

Glasser, Neil F., et al. "Recent spatial and temporal variations in debris cover on Patagonian glaciers." Geomorphology 273 (2016): 202-216.

Herreid, Sam, et al. "Satellite observations show no net change in the percentage of supraglacial debris-covered area in northern Pakistan from 1977 to 2014." Journal of Glaciology61.227 (2015): 524-536.

We agree and this figure was replaced by new one, please see P8 L12 (Fig. 4)

Figure S8: It is confusing to discuss "Debris cover outline[s]" as well as "Bare ice outline[s]" as you are using different words to reference effectively the same thing.

We replaced this figure by new one, please see Fig. S7 (P4 L9 in supplement)

**Technical corrections**

P1L30: I think "SDC" should be defined at the first mention of debris cover.

We spelt out "supraglacial debris cover" everywhere

P1L33: Remove "the" before "SDC" and "glacier ablation."

We changed this sentence, please see P1 L39

P1L37: Add "summarized in Kirkbride and.."

Rather than add "summarized", we also cited Glasser et al., (2016) P2 L3

P1L40: "The difficulty. . . " I would say "One difficulty.."

Done, please see P2 L6

P2L1: Use SDC consistently if defined, e.g. here: "properties of debris"

We spelt out "supraglacial debris cover" everywhere

P2L1: Change to "of a debris layer has"

We changed this sentence, please see P2 L7-8

P2L10: Change "as it is similarity in" to "as it is similar to"

We changed this sentence, please see P2 L16-17

P2L10: Comma after citation

Done, please see P2 L16

P2L11: Consider changing "key player" to "glacier-wide component"

We changed this sentence, please see P2 L16-17

P2L12-13: "as surface mass balance. . .is different from that of bare ice" this has already been established earlier in your introduction and I don't think it needs to be restated.

We changed this sentence, please see P2 L17-18

P2L19: Change to "and a recently"

We changed this sentence, please see P2 L24-26

P2L29: Change to "as the largest"

We changed this sentence, please see P2 L34-35

P2L31-33: Awkward sentence, please break into two

Done, please see P2 L37-40

P2L35: "imagery from 2016. The SPOT" not clear if one or several images were used

We changed this sentence, please see P3 L1

P3L14: I would remove "Relatively heavily"

P3L15: add "..to distinguish the glacier boundary" or "glacier terminus"

P3L18: change to "...might result in a potential..."

P4L1: change to "i) a buffer method"

P4L2 "manual"

P4L9: correct English in this sentence

P4L13: end parenthesis after NSD

*P6L5: please add "Glacier" after named glaciers, here and throughout the article (or "glaciers" after a list)*

We deleted all these sentences because they are no longer associated with this manuscript.

P6L30: This is not a "whereas" statement.

Done, please see P9 L17

P6L30: "increment" should be "increase."

Done, please see P9 L18

P7L14: "vital" seems like too strong of language to me.

We changed by "worthwhile", please see P10 L5

[revised manuscript text omitted]

---

## Author Response (AR2)

Dear editor,

We would like to thank the reviewers for their additional in-depth review and are glad to read that the reviewers agree that our manuscript has much improved. We have addressed the remaining comments thoroughly and would be happy if the manuscript can now be considered for publication. Please find below our detailed response to the reviewers' comments. We would also like to mention, that the manuscript was much expanded due to the many corrections and new text and figures. Therefore, we decided to convert the manuscript as "standard format article" and not a short communication as also suggested by Dirk Scherler. In addition, we checked the journal guidelines, that says the "Brief communication" should be short (2–4 journal pages: https://www.the-cryosphere.net/about/manuscript_types.html)
We hope you agree with this decision.

All corrections and changes what we did in the manuscript are in yellow.

Best regards,

Levan Tielidze on behalf of all co-authors

**Reply to Dr. Dirk Scherler's comments**

The Reviewer comments are shown in black while author responses are subsequently provided in red

The revised version of Tielidze et al.'s contribution is much improved over the first submission. The authors have clarified the methodology, improved the analysis and streamlined the discussion.
My main comments are related to the discussion, which I think still needs some work. First, I think the order of the two chapters should be reversed. It makes more sense to me to that you first compare your results with those of others in order to discuss how your results are similar or different and how they combine to provide a more complete picture of debris-cover change in this region. Second, the chapter on supraglacial debris cover change is in fact more about the reasons of change and that should be stated explicitly, also in the chapter title. I think this is a very useful chapter to have, but I found some of the reasoning not well supported. Perhaps you could expand the discussion here by comparing your regions. If the regions are indeed different in topography and climate, you could provide

measurements (e.g., mean catchment slope, mean annual precipitation, etc.) and figures that support some of your arguments.

We thank you for offering constructive and thorough comments on our paper. Each concern is addressed below and in the manuscript, and we believe these changes have improved the clarity and quality of the paper.

First of all we would like to mention, that due to many corrections and new text/figures, the manuscript was much expanded. Therefore, we decided to convert it as the "standard format article".
We also took in to account your previous suggestion, that *the paper should be a relevant contribution, but may be better published as a standard format paper, instead of a brief communication?"*

We also changed the term - "Caucasus Mountains" to the "Greater Caucasus" in the title. The "Caucasus Mountains" is a broad concept and consist of two separate mountain systems: the Greater Caucasus and the Lesser Caucasus. As the current study was conducted just in the Greater Caucasus the new title stands as the "**Supraglacial debris-cover changes in the Greater Caucasus**"

We have also taken care to address discussion section. First of all we changed the structure of the discussion and it starts from "5.1 Comparison with previous investigations" and continues to "5.2 Possible reasons of supraglacial debris-cover changes".
We also provided a new (edited) paragraph in order to improve the difference between the slopes:
"Our investigation shows also that the supraglacial debris cover increases more quickly in the northern slopes of the Greater Caucasus than in the southern. Due to the climatic (more radiation input on the southern side) and orographic conditions, glaciers on the southern slopes have relatively smaller size compared to their northern equivalents, although smaller glaciers exist as well in high cirques. Glacier surfaces on the northern slopes are less steep than the south. Most valley glacier tongues in the north are longer and reach lower altitudes than the southern-facing glaciers. But there are some exceptions, where the northern-facing glaciers are shorter and steeper, and here, the glaciers of the southern slope are characterized with relatively more supraglacial debris cover. An example is Georgia's largest glacier Lekhziri and its northern counterparts, with the exception of the Bashkara Glacier (Fig. S6). This conclusion is supported by Lambrecht et al. (2011) who observed increase of supraglacial debris cover more rapidly in the northern slopes, than the southern."
Please see P11 L10-16 and P13 L1-4

In addition, we provided better evidence for the differences of supraglacial debris cover formations in the western, central and eastern sections/regions:
"The variation of supraglacial debris cover area in the eastern, central and western Greater Caucasus could mostly be conditioned by climate, lithology and morphological peculiarities of the relief. Some river basins in the eastern Greater Caucasus are built on

Jurassic sedimentary rocks, which suffer consistent denudation (Gobejishvili et al., 2011; Bochud, 2011) suitable for supraglacial debris cover formation. Furthermore, high erodability of the rocks may be a major reason why rock glaciers are widespread in the eastern Greater Caucasus (Tielidze, et al., 2019b). The relief of the central Greater Caucasus is mainly constructed from Proterozoic and Lower Paleozoic plagiogranites, plagiogneisses, quartz diorites and crystalline slates, which present poor conditions for the formation of rock avalanches in this area. In addition, the central Greater Caucasus is the highest section of the main watershed range and glacier surfaces are relatively steeper making less favourable conditions for supraglacial debris cover accumulation. The western Greater Caucasus is hypsometrically lower with less steep glaciers. This section is distinguished with the highest glacier reduction after the eastern Greater Caucasus and it is possible that thinning glaciers rapidly become debris-covered over the ablation area (Pratap, et al., 2015). This might be confirmed by detailed field measurements and could be part of a separate investigation."
Please see P13 L5-18

Regarding to your comment: *"you could provide measurements (e.g., mean catchment slope, mean annual precipitation, etc.) and figures that support some of your arguments."* The manuscript and supplement overall contains 14 images, and we believe that this is quite enough for the one publication.

Specific comments
P1L37-39: The effect of debris thickness on melt rates has first been described by Östrem (1959): Geografiska Annaler, Vol. 41, No. 4 (1959), pp. 228-230.
Suggested reference has been added in the text. Please see P1 L41

P2L1-3: Here too, I think you could refer to earlier studies that document changes in debris cover. For example, Deline (2005; The Holocene 2005 15: 302 DOI: 10.1191/0959683605hl809rr) showed an increase of debris-cover for Mont Blanc glaciers, and Stokes et al. (2007) for the Caucasus.
Done, suggested references have been added in the text. Please see P2 L3

P2L18-21: In your discussion (P9L34-37), you mention the issue of nominal glaciers in the RGI classified as debris-covered in Scherler et al. (2018). Why not mention it here already? It would be a great motivation for your study to provide an improved estimate of supraglacial debris cover for the Caucasus. Holding back the information until the discussion makes no real sense, because your results are not instrumental in detecting this issue.
Special thank for this comment. We added our motivation here and overall this section stands as a:
"A recent global study (Scherler et al., 2018) measured that supraglacial debris cover is abundant in the Caucasus and Middle East (more than 25% of glacier area) and that this region shows the highest percent of supraglacial debris cover worldwide. However, Scherler et al. (2018) used the RGI v6 database with some inconsistent co-registration and

nominal glaciers. That makes a good motivation for us to provide an improved estimate of supraglacial debris cover for this region."
Please see P2 L18-22

P2L29: It would be useful if you could mention in this chapter what fraction of all glaciers in the Caucasus you have analyzed in this study.
We provided appropriate sentence that making more clear what fraction of all glaciers we analyzed:
"Overall, this equals 49.5% and 32.6% of the Greater Caucasus total glacier area and number respectively."
Please see P3 L25-26

P2L30: "different climate conditions": Can you provide more details on how the climate differs in the studied regions? This may also help you in the discussion of why the regions differ in the rate of debris cover change.
The new format of the manuscript allowed us to provide more detail information not just about the climatic, but also orographic and current glaciation differences between the selected regions. Consequently, we provided the new paragraph - "**2 Study area**". We think that it makes the manuscript more comprehensive. In addition, it helps the reader to better understand the climatic and topographic differences between the selected regions:
"**2 Study area**
The Greater Caucasus is one of the world's highest mountain systems, and the major mountain unit of the Caucasus region. The range stretches for about 1300 km from west-northwest to east-southeast, between the Taman Peninsula of the Black Sea and the Absheron Peninsula of the Caspian Sea. Using morphological and morphometric characteristics, the Greater Caucasus can be divided into three parts - Western, Central and Eastern. At the same time, the terms northern and southern Greater Caucasus are commonly used. The central Greater Caucasus is the highest part of the main watershed range represented by summits exceeding 5000 m: Dykh-Tau - 5205 m, Shkhara - 5203 m, Jangha - 5058 m, and Pushkin Peak - 5034 m. The western and eastern sections are relatively lower with highest summits of Mt. Dombai-ulgen (4046 m) and Mt. Bazardüzü (4466 m) respectively. Elbrus is the highest summit of the Greater Caucasus with two peaks - western (5642 m) and eastern (5621 m).

According to the recent inventory, this mountain range contains over 2000 glaciers with a total area of about 1200 km$^2$. The northern slopes of the Greater Caucasus contain more glaciers than the southern slopes (Tielidze and Wheate, 2018). The altitude of the glacier equilibrium line (ELA), increases from 2500–2700 m in the west to 3700-3950 m in the eastern sector of the northern slope of the Greater Caucasus (Mikhalenko et al., 2015). The ELA was determined to range from ~3030 m in the west to ~3480 m in the eastern section of the southern slope of the Greater Caucasus (Tielidze, 2016). The ELA is ~1000 m higher on the northern slopes of the Elbrus than the southern slopes of the central Greater Caucasus (Mikhalenko et al., 2015).

As the greater Caucasus range is located on the boundary between temperate and subtropical climatic zones, the orientation and height of the range determines the contrasts between the northern and southern slopes. The mean annual temperatures at the northern

slopes are usually 1-2°C cooler than those in the south (Tielidze and Wheate, 2018). The average regional lapse rate is minimum in winter (2.3°C per 1000 m) and maximum (5.2°C per 1000 m) in summer (Kozachek et al., 2016).

Precipitation arrives from the west, in storm systems that replenish the waters of the Black Sea, driving the contrasts between the eastern and western of the southern slope, as well as between the southern and the northern slopes. Annual precipitation ranges between 2000-2500 mm in the west and declines to 800-1150 mm in the east on the northern slope of the Greater Caucasus. The central section of the southern slope receives over 2000 mm of precipitation while in the east, the annual total is 1000 mm. The south-western section of the region is very humid with annual precipitation about 3200 mm (Volodicheva, 2002; Mikhalenko et al., 2015)."
Please see P2 L29-42 and P3 L1-17

P4L19-21: Is this sentence complete? Sounds like words are missing.
We corrected this sentence as:
"The normalized standard deviation (NSD – based on delineations by two digitizations divided by the mean area) (Paul et al. 2013) between two datasets (Landsat and SPOT) was ±7.4%."
Please see P6 L24-25

P6L2-3: The bars: You write in the caption they reflect the ratio of clean ice to debris cover. Don't they rather reflect the percentage of debris cover?
We corrected this sentence as:
"Percentage increase of supraglacial debris cover in the Greater Caucasus for 1986, 2000 and 2014 by different regions (glaciers are non-existent on southern slopes of the eastern Greater Caucasus)."
Please see P8 L13-14

P6L11: "rate of supraglacial debris cover increase": Are these really rates? The units do not say so. Did you divide the percentage or area changes by the time period to come up with units %/yr or km^2/yr?
We calculated supraglacial debris cover increase and glacier area decrease rates for all selected regions and entire study area. All these calculations are included in the manuscript. For more clarifications, we provided new figure:
"**Figure 4.** Supraglacial debris cover increase (yellow) and glacier area decrease (green) rates in the Greater Caucasus by slopes, sections and mountain massifs in 1986–2000, 2000–2014 and 1986–2014."
Please see P9 L7-8

P6L20-26: I wonder if you want to show a figure relating to this paragraph, either in the paper or in the supplementary material? The question is what information you take away from mentioning the big glaciers and how that ties in to your discussion.
We provided a new figure in the supplement, showing the Bezingi (debris covered) and Karaugom (debris free) glaciers comparison - Terminus retreat differences. This figure

supports our discussion that debris-covered glaciers may not be as sensitive to climate change as debris-free glaciers.
Please see Figure S2 in the supplement.

Also, if the numbers in parentheses are rates, make sure to provide the correct units.
We corrected all rate numbers as a (-6.3% or -0.22% $yr^{-1}$) and (-17.8% or -0.63 $yr^{-1}$).
Please see P8 L28 and P9 L2

P7L2-4: Is the upglacier enlargement of the debris cover not simply a consequence of the rise of the ELA and the enlargement of the ablation area at the cost of the accumulation area?
Indeed, all these events are closely related each other, but we do not provide the ELA measurement in this manuscript. We think it requires deep and complex analyses with more calculations. This could be an interesting topic of further research and separate investigation.

P7Figure 3: Why did you choose 5642 m as the upper limit of your y-axis? Also, some of the curves are truncated with this choice.
We choose this elevation as a highest point of the investigated region. Glaciers on Elbrus are situated in the altitudinal range of 2500 to 5642 m. The truncated curve caused by choosing of an altitudinal zones with 500 m ranges (e.g. 4000-4500, 4500-5000, 5000-5642). In case of choosing an altitudinal zones with 100 m ranges, this curve would be close to the y-axis. But we note that all curves on this figure are shown correctly.

P7L15: "appears": Why appears? You have the data to quantify the attribution of the reduction in clean ice areas to retreat and debris-cover expansion.
We have changes this sentence and new sentence stands as: "This reduction was caused by both glacier retreat and an increase in total supraglacial debris cover (Table 2, Fig. 3-6)."
Please see P11 L34-35

P7L19-22: Can you provide more quantitative data to support your suggestion that the increase rate of debris cover during the 2000-2014 period was higher due to rock avalanches? The changes shown in Fig. 4 do not appear to be due to rock avalanches falling on glaciers.
We provided more calculation and quantitative data for some glaciers. Overall it stands: "Rock avalanches after 2000 on some glaciers in the Greater Caucasus (particularly in the eastern section), have strongly increased supraglacial debris cover (Tielidze, et al., 2019a). Supraglacial debris cover area increased from 2.1±6.1% to 17.6±5.7% or +1.09% $yr^{+1}$ for the Suatisi Glacier and from 5.9±6.0% to 19.1±5.6% or +0.94% $yr^{+1}$ for the Devdoraki Glacier between 2000 and 2014 (Fig. S5). This might be one of the reasons why the increase rate was higher during the second period (2000-2014)."
Please see P11 L40-41 and P12 L1-4

In addition, we provided more suitable images in the supplement, showing SDC area increase after the rock avalanches in the second investigated period
Please see Figure S5 in the supplement

P7L23: Your explanation is not readily evident to me. Can you please explain in more detail? I would have guessed that steeper slopes would result in more debris cover (see discussion in Scherler et al., 2011, JGR, VOL. 116, F02019, doi:10.1029/2010JF001751).
We would like to clarify that we meant the "steep surface of the glacier" and not "steep slope". We provided new sentence here:
"Our investigation shows also that the supraglacial debris cover increases more quickly in the northern slopes of the Greater Caucasus than in the southern. Due to the climatic (more radiation input on the southern side) and orographic conditions, glaciers on the southern slopes have relatively smaller size compared to their northern equivalents, although smaller glaciers exist as well in high cirques. Glacier surfaces on the northern slopes are less steep than the south."
Please see P12 L10-14

P8L2: "increase rate": I believe you the rate was higher between 2000-2014 compared to before, but it is hard to read from Fig. 4. Perhaps add the rate to your table or add another table with rates?
Accepted. We calculated supraglacial debris cover increase and glacier area decrease rates not just for the Elbrus but also for all selected regions and entire study area. For more clarifications, we provided a new figure:
"**Figure 4.** Supraglacial debris cover increase (yellow) and glacier area decrease (green) rates in the Greater Caucasus by slopes, sections and mountain massifs in 1986–2000, 2000–2014 and 1986–2014."
Please see P9 L1-8

P8L6-8: This statement is not really evident to me. Looking at Fig. 4, I see both large and small increases in debris cover in all directions. You could provide a rose diagram to analyze how changes in the rate depend on orientation. Also, if you want to link the thinning with the expansion, you should show that these quantities correlate.
Accepted. For this discussion we provided new text and the appropriate image in the result section:
"Supraglacial debris cover distribution according to the different slopes of the Elbrus was not homogenous. The increase rate was highest on the eastern slope from 1.22% to 8.20% or +0.25% $yr^{+1}$ between 1986 and 2014, while the western slope had lowest increase rate from 7.10% to 8.55% or +0.05% $yr^{+1}$. In the same time, glacier area decrease was lowest on the western slope from 9.43 $km^2$ to 9.23 $km^2$ or -0.08% $yr^{-1}$ and highest on the eastern slope from 36.76 $km^2$ to 33.50 $km^2$ or -0.31% $yr^{-1}$ (Fig. 5a-c)."
Please see P8 L5-10
In addition, please see new figure: "**Figure 5.** a – Supraglacial debris cover (SDC) area increase for the Elbrus slopes between 1986 and 2014. b and c – Total glacier area ($km^2$) and supraglacial debris cover percentage distribution between 1986 and 2014."
Please see P10 L1-3

P8L16-18: Perhaps; but this statement may be too general. Glaciers in Fig. 4 that had already some debris cover in 1986 appear to have retreated less than others.

Agreed. We have modified this sentence to read: "The glaciers in the Greater Caucasus have retreated continuously since 1960 (Tielidze and Wheate, 2018), suggesting that the shielding effect of increased supraglacial debris cover at the glacier surface may only partly offset the retreat trend."

Please see P13 L30-32

P8L19-20: Why is the thermal resistance of the debris on Caucasus glaciers different from that on other glaciers? And why is it relevant for your discussion here?

The detailed comparison of thermal resistance between Caucasus and other mountain glaciers was described by Lambrecht et al. 2011 (see reference in the manuscript), discussing that it should be caused by local geology, different grain size distribution (air content) and water saturation, etc. We don't think it is basic to describe all these reasons in the Discussion here. Furthermore, it still requires more detail investigation, in order to better understand all these causes.

We include some results of thermal resistance measurement here, because the "**relatively higher thermal resistance**" makes debris-covered glaciers even less sensitive to climate change in the Caucasus than other mountainous regions. In addition, it is good consistent with our measurements that debris-covered glaciers characterized lower retreat (e.g. Bezingi) than high sensitive debris-free glacier (e.g. Karaugom).

P9L6: Changes in retreat rate between glaciers with different amounts of debris cover as well as different bed slopes have earlier been reported for Himalayan glaciers in Scherler et al. (2011, Nature Geoscience, DOI: 10.1038/NGEO1068).

Done. Suggested reference has been added accordingly P13 L41

P10L15: Sure you mean "gradually"?

Done. Suggested sentence has been changed accordingly P15 L7

**Reply to Dr. Sam Herreid's comments**

Reviewer comments are shown in black while author responses are subsequently provided in red.

I would like to applaud the authors, this revised version of the article is substantially improved. Below are minor comments with two specific points raised about overlap between this work and Tielidze and Wheate, 2018, and the ability to detect a debris cover change signal. After a consideration of the comments below, I would recommend publication of this article in The Cryosphere.

We are very grateful for this very positive feedback! And we thank you for the helpful comments, which have helped to improve the manuscript. We address all your comments below.

P1L31: Missing "per" year superscript "-1"
Done. Superscript has been added in the text. Please see P1 L29

P1L38: Add this citation along with Nicholson et al., 2018: Østrem, G., 1959. Ice melting under a thin layer of moraine, and the existence of ice cores in moraine ridges. Geografiska Annaler, 41(4), pp.228-230.
Done. Suggested reference has been added in the text. Please see P1 L41

P2L1: I think it's either "energy balance" or "mass balance" not "energy mass balance".
Done. Suggested sentence has been changed as a "mass balance". Please see P2 L2

P2L15: "Ice and snow melt in these mountains are major sources of runoff for populated places" Is this common knowledge or is there a citation for this?
Appropriate reference has been added in the text (Tielidze, 2017).
Tielidze L.: Introduction. In: Glaciers of Georgia. Geography of the Physical Environment. Springer, Cham, https://doi.org/10.1007/978-3-319-50571-8_1, 2017.
Please see P2 L15

P2L19: I think Scherler et al., 2018 "measured" that debris cover is abundant, not "suggest."
Done. Suggested sentence has been changed accordingly. Please see P2 L18

P2L20: more than 25% [of] glacier area
Done. Suggested sentence has been changed accordingly. Please see P2 L19

P2L21: Add citations for the earlier studies that indicate low relative supraglacial debris cover in the Greater Caucasus, or if already present in the three citations at the end of the

sentence, move the relevant citations to the middle of the sentence so it is clear which reference is to which statement.

Done. Suggested sentence has been changed accordingly. Please see P2 L23-24

P2L33: For consistency with how you set up this list, give the ratio (130/0).

Done. Suggested sentence has been changed accordingly. Please see P3 L23

P2L34: northern/southern slope ratio is not given for the Elbrus massif and not divided by aspect in later analysis (although the eastern slopes are referenced). Please explain why or add this.

We have divided Elbrus glaciers according to different slopes:

"In addition, all 21 glaciers on Elbrus (5 - northern slope, 8 - southern slope, 5 - western slope, 3 - eastern slope) - the largest glacierised massif in the whole region - were selected"

Please see P3 L23-25

In addition, we added more results in the "Result" section:

"Supraglacial debris cover distribution according to the different slopes of the Elbrus was not homogenous. The increase rate was highest on the eastern slope from 1.22% to 8.20% or +0.25% $yr^{+1}$ between 1986 and 2014, while the western slope had lowest increase rate from 7.10% to 8.55% or +0.05% $yr^{+1}$. In the same time, glacier area decrease was lowest on the western slope from 9.43 $km^2$ to 9.23 $km^2$ or -0.08% $yr^{-1}$ and highest on the eastern slope from 36.76 $km^2$ to 33.50 $km^2$ or -0.31% $yr^{-1}$ (Fig. 5a-c)."

Please see P8 L6-10

In addition, please see new figure: "**Figure 5.** a – Supraglacial debris cover (SDC) area increase for the Elbrus slopes between 1986 and 2014. b and c – Total glacier area ($km^2$) and supraglacial debris cover percentage distribution between 1986 and 2014."

Please see P10 L1-3

Also, we added one more sentence in the "Discussion" section as a

Glaciers in the western slopes are affected by avalanches and thus are partially debris covered (Kutuzov, et al., 2019).

Please see P13 L21-22

P3L7: is 1986 to 2014 explicit or do you mean 1985/86 to 2013/14?

We meant 1985/86 to 2013/14 and this sentence has been changed accordingly.

Please see P7 L1

P3L10: It's not clear to me what work is new here and what is from Tielidze and Wheate, 2018. Looking at this earlier article, effort was taken to include debris cover. If you are using the same work then you do not need to describe the mapping methods in this article, simply cite your earlier work and present your new analysis of debris cover changes. If you have made changes to the earlier inventory, then I think a missing result is the difference between these glacier outlines and those of Tielidze and Wheate, 2018 for the subset regions considered here.

The inventory by Tielidze and Wheate (2018) is based only "manually mapped" outlines and does not contain any dataset (result) about the supraglacial debris cover distribution (except the only one Shkhelda Glacier - 43°100 N, 42°380 E).

Furthermore, we have not used a "semi-automated" mapping method (outlines) and any threshold values in the mentioned inventory. All selected glaciers were mapped just manually there.

In the current study, we only use these manually mapped glacier outlines from Tielidze and Wheate (2018), nothing else from there. And that's why the "manual mapping" method was not described here again.

All other works were performed for this new study as it is described in the methodology section.

Overall, this sentence stands as:

"Other datasets used in this study include the "Greater Caucasus Glacier Inventory" manually mapped dataset (Tielidze and Wheate, 2018), high resolution images from Google Earth, and GPS measurement."

Please see P4 L1-2

P3L21-22: Manual improvements correcting for snow and shadows, does this step also include mapping debris cover?

This does not include mapping debris cover, although this step helped us to determine clean ice (semi-automated) outline correctly that allowed us to measure difference between entire glacier (manual outline from Tielidze and Wheate, 2018) and clean ice (semi-automated) outline properly or to calculate supraglacial debris cover area.

Thus, we have not done any corrections here.

P3L24: "Supraglacial debris cover was extracted and saved as a separate layer." You can remove this sentence, your readers know you did this implicitly.

Done. Suggested sentence has been deleted accordingly.

P4L16-17: "The accuracy is +/-30 m for debris and +/-15 m for clean ice" How did you derive this result? Is this an average value for all seven glaciers?

We clarified this sentence: "Based on all seven glacier measurements, the average accuracy was calculated as ±30 m for supraglacial debris cover and ±15 m for clean ice."

Please see P6 L21-22

P6L11: make it a little more clear that these results are no longer per your subregions. Something like "For all regions investigated in the Greater Caucasus mountains, the rate of supraglacial debris cover was different between northern and southern aspects."

Done. Suggested sentence has been changed accordingly. Please see P8 L16-17

P6L12: "debris-covered"

Done. Suggested sentence has been changed accordingly. Please see P8 L17

P6L22: missing superscript "-1"

Done. Suggested superscript has been added accordingly. Please see P8 L28

P6L23: (and P6L26): rewrite; something like "[..same period] with a terminus retreat of"
Done. Both suggested sentence have been changed accordingly. Please see P8 L29

P7L24: "northern slopes are less steep" do you mean the mountain slope above the glaciers? How would this couple with more debris cover? Can you take this one logic step further?
We meant the surface of the glacier. We clarified this sentence:
"Our investigation shows also that the supraglacial debris cover increases more quickly in the northern slopes of the Greater Caucasus than in the southern. Due to the climatic (more radiation input on the southern side) and orographic conditions, glaciers on the southern slopes have relatively smaller size compared to their northern equivalents, although smaller glaciers exist as well in high cirques. Glacier surfaces on the northern slopes are less steep than the south."
Please see P12 L10-14

P8L2-3: "..although the total uncertainty is comparable to the obtained relative changes" I have a lot of respect for honest results that fall within suitable error bounds, however, there are clear changes between 1986 and 2014. Why was your method unable to confidently resolve this? In this discussion can you propose a method that might be more successful?
We clarify that this high error caused by using the buffer method for the small size of the supraglacial debris-covered outlines.
We clarify that this high error was caused by using the buffer method for the small size of the supraglacial debris-covered outlines.
The error is mostly inversely proportional to the length of the outline margin. So it depends strongly on the size of the outline. In this sense, the area error assessed by this study is rational because it accounts for the length of the SDC outline perimeter.
The more successful method could be multiple digitization that was excluded recently. Overall we don't think that the "high error values" have vital importance in this case and have not done any changes here.

P8L3-4 "Comparison with semi-automated methods shows that debris cover may be considerably underestimated." First, it is unclear what you are referring to as the semi-automated method, if I follow correctly, this is the bare ice area that was subtracted from manual outlines to derive a debris cover so how could this dataset provide the comparison to make this statement? Beyond descriptive clarity, I'm inclined to agree with the general statement. I had a quick look on Earth Explorer and compared the image from 1986 against your Figure 4 (see attached figure). I added some arrows where Figure 4 misleadingly shows changes, e.g. an entirely new medial moraine network (labeled 'a'), area that is shown to gain debris cover but if this is the case then the glacier would have had to have grown not shrunk over the investigation period ('b'), and what looks like two rock avalanche deposits that are missed ('c'). I think it is a fine argument that these points are within your error bounds and I acknowledge that it is difficult to maintain consistent methods with debris-covered area change measurements because earlier sensors do not have the radiometric resolution that the later sensors do. However, the images do show true

changes and it is the challenge of this line of research to be able to extract a meaningful signal.

[Figure]

LT05_L1TP_171030_19860806_20170217_01_T1

We would like to say special thanks for this comment here.

Apparently, we technically missed some debris-covered spots while making a new overlapping picture for the Elbrus massif. Please take in to account, this does not mean that we missed these areas while the calculating overall supraglacial debris cover area. For more clarification please see image below, from previous version of the manuscript (that was published in The Cryosphere Discussion) showing the SDC area changes between 1986 and 2014. All missed spots selected by you, are shown in the panel "a".

To double check please see: https://www.the-cryosphere-discuss.net/tc-2018-259/tc-2018-259-supplement.pdf

[Figure]

We have replaced a wrong version of the figure with new one. Please see P14 L1-3
In order to avoid any confusion, we made the "gif" file additionally showing clear differences between clean-ice and SDC in 1986- 2014.

The "gif" file is attached as a separate file in the supplement.

In addition, we have changed this sentence to: "Glaciers in the western slope of Elbrus are affected by avalanches and thus are partially debris covered (Kutuzov, et al., 2019). Glaciers on the eastern slope are characterized by high rates of retreat and great expansion in proglacial lake number and area (Petrakov et al., 2007)."
Please see P13 L21-24

P8L5: "lake number and area"
Done. Suggested sentence has been changed accordingly. Please see P13 L23

P8L7: I do not follow the evidence for an increase of debris cover being explained by resurfacing of englacial debris. Is there a pattern in how the debris cover looks that would lead you to make this assumption?
Accepted. Indeed, this statement is probably speculative and was not supported by strong evidence. Text was revised:
"The most significant increase of supraglacial debris cover occurred on the eastern oriented glaciers of Elbrus, where glaciers are characterized by the highest thinning rates in recent years (Kutuzov, et al., 2019)."
Please see P13 L24-26

P8L9: what boundaries in the subsurface are you looking for and why?
We meant that the GPR survey could be a good opportunity to recognising the ice under the SDC. We have changed this sentence and now stands as a:
"Detailed Ground-Penetrating Radar (GPR) survey may help to more accurately identify supraglacial debris cover extent in this area (e.g. unpublished GPR measurements by S. Kutuzov and I. Lavretiev showed that ~30 m of ice may be present under the previously considered ice-free area on the eastern slope of Elbrus)."
Please see P13 L26-29

P9L1: please make statements without ambiguities like "somewhat"
We use word "relatively" instead of "somewhat". Overall, we changed the structure of this sentence to:
"Direct field measurements show that thermal resistance of the <20 cm supraglacial debris cover for some glaciers (e.g. Djankuat and Zpkhito) in the Greater Caucasus is relatively higher (0.07-0.15°C and 0.05-0.08°C m²/W) than in other glacierised regions of the world (e.g. Baltoro, Karakoram 0.02-0.07°C and Maliy Aktru, Altay 0.02-0.09°C m²/W) (Lambrecht et al., 2011)"
Please see P13 L33-36

P9L6: end the sentence after "(2008)", and I believe you meant to say Rowan et al. (2005) found similar model results.
Done. Suggested sentence has been changed accordingly. Please see P13 L40-41

P10L10: the list of detailed field measurements: "debris thickness, GPR, radiation" give a

physical quantity, a geophysical tool and an emission of energy, please rewrite to list measurements.

Accepted and we provided new sentence here:

"Future work should focus on using high resolution aerial/satellite imagery and more detailed field measurements. e.g. debris thickness measurement by Ground-Penetrating Radar using a Sensors and Software pulseEKKO 1000, with 225, 450, 900 and 1200 MHz antennas (Mccarthy, et al., 2017), or incoming and reflected solar radiation; long-wave terrestrial and returned radiation by Kipp & Zonen CRN1 net radiometer (Lambrecht, et al., 2011)."

Please see P14 L16-18 and P15 L1-2

Figure 2. This figure is very nice. Could you please add the error bounds? Also, the wording "clean ice to supraglacial debris cover [ratio]" should be reversed, the ordering implies numerator and denominator.

Done. Suggested figure and sentence have been changed accordingly. Please see P8 L12-14

Figure 4. I'm not questioning the area you show as clean ice in 1986 that is no longer clean ice in 2014 high on the massif, but area that is former glacier and now bedrock is distinctly different from a discussion of debris cover changes on a glacier's surface.

In order to better clarify, we provided new sentence below to Fig. 7: "Blue color shows retreat of clean ice parts."

Please see P14 L3

Figure S4: A figure showing a permafrost dataset with no real context or discussion seems out of the scope of this paper.

We have changed this figure. First of all we deleted "panel c" from the figure. We provided more suitable images in the supplement, showing SDC area increase after the rock avalanches in the second investigated period.

[revised manuscript text omitted]

2 **Figure S6.** A comparison of supraglacial debris cover (SDC) and clean-ice area distribution in 1986-2014 for the
3 southern (a – Lekhziri) and northern-facing (b – Kashkatash, c – Bashkara and d – Djankuat) glaciers. Landsat 8
4 (panchromatic band 8), 03/08/14 was used as background.

---

## Author Response (AR3)

Dear editor,

We would like to once again thank you for your time and helpful suggestions to make the manuscript more accessible and accurate.
We took in to account all your comments and provided our corrections point by point.

All corrections and changes what we did in the manuscript are in yellow.

Best regards,

Levan Tielidze on behalf of all co-authors

Title: Adding the range of date of the study period would be useful to readers.
Done, we added the study periods in the title and now it is stands as:
**Supraglacial debris-cover changes in the Greater Caucasus from 1986 to 2014**
Please see P1 L1

2.21. "nominal" is RGI jargon, but could you clarify this concept in full words for all readers.
Done, we clarified it accordingly:
"Scherler et al. (2018) used the glacier outlines from the Randolph Glacier Inventory (RGI) v6 database (RGI consortium, 2017) with some geolocation errors and nominal glaciers, representing glacier area by an ellipse and included in RGI to achieve global coverage in case no other information were available (Pfeffer et al., 2014)."
Please see P2 L18-21

2.23. "lower" than what?
We clarified this sentence as follows:
"However, earlier studies indicated lower relative supraglacial debris-cover than our study in the Greater Caucasus but are restricted to smaller regions (Stokes et al., 2007) or individual glaciers (Lambrecht et al., 2011; Popovnin et al., 2015)."
Please see P2 L16-18

2.30. Would be good to state what percentage of the overall Caucasus glaciers (in area) are found in the Greater Caucasus. To illustrate that you map debris for a large fraction of the Caucasus.
Done, we added an appropriate sentence accordingly:
"The Greater Caucasus is one of the world's highest mountain systems, and the major mountain unit of the Caucasus region containing about 96% of contemporary glacier area of the Caucasus and Middle East glacier region according to RGI (Pfeffer et al., 2014)."
Please see P2 L28-30

2.34. Why stating "At the same time, the terms northern and southern Greater Caucasus are commonly used."? It complicates the picture and confused me a bit.
We clarified this sentence as follows:
"Moreover, the Greater Caucasus can also be subdivided the northern and southern part according to the location relative to the main crest of the range (Fig. 1)."
Please see P2 L33-34

2.38. In the end we do not know in which of the three parts of Caucasus, Elbrus is located. Is Elbrus part of Greater Caucasus. Why is it treated separately?

We clarified this sentence as follows:

"Contemporary glaciers are concentrated mostly along the main mountain range, as well as in the sub-ranges of the Greater Caucasus and separate massifs such as Elbrus and Kazbegi-Jimara (Fig. 1). Elbrus is a separate dormant volcanic mountain on the border between western and central Greater Caucasus about 10 km north from the main watershed divide. It is the highest summit of the region with two peaks - western (5642 m) and eastern (5621 m)."

Please see P2 L37-41

2.40. (Tielidze and Wheate, 2018) should be moved just after "recent inventory"

Done.

"According to the recent inventory (Tielidze and Wheate, 2018), this mountain range contains over 2000 glaciers with a total area of about 1200 km$^2$."

Please see P2 L42

3.9. Values of the temperature lapse rate should be negative. (temperature decrease while z increase)

Done.

"The average regional lapse rate is minimum in winter (-2.3°C per 1000 m) and maximum (-5.2°C per 1000 m) in summer (Kozachek et al., 2016)."

Please see P3 L17-18

3.11. Do they bring water to the back sea? Or do you want to say that evaporation from the back sea re-inforce the depression systems? Unclear.

We improved this paragraph and changed accordingly:

"Annual precipitation varies between 2000-2500 mm in the west and declines to 800-1150 mm in the east on the northern slope. The central section of the southern slope receives over 2000 mm of precipitation while in the east, the annual total is 1000 mm. The south-western section of the region is humid with annual precipitation about 3200 mm (Volodicheva, 2002; Mikhalenko et al., 2015). The south-western slopes of the Greater Caucasus experience heavy snowfall and snow avalanches (from November to April). Snow cover may reach 5-7 meters in several regions of the western part of the Greater Caucasus, such as the Upper Svaneti and northern Abkhazeti regions in Georgia (Sylvén et al., 2008)."

Please see P3 L19-25

3.21. Need to tell for what year.

We clarified year in brackets:

"We selected 659 glaciers with a total area of 590.0±25.8 km$^2$ (based on the "Greater Caucasus Glacier Inventory" and representing the year 2014 (Tielidze and Wheate, 2018)):"

Please see P4 L3-4

3.27. I must say I am surprised that you need to quote a paper for the area of polygon. Not needed probably. Or if there is some understatement/complication hidden by this, make it clear.

We removed an extra citation and changed sentence accordingly.

"The size of the selected glaciers varied between 37.5 km$^2$ and 0.01 km$^2$."

Please see P4 L8-9

Figure 1a. "percentage increase". I think this is not the increase but just the percentage of debris cover. Double check.

Done. We corrected accordingly.

Please see new Figure 1.

P3 L5

6.5 Update the Mölg et al. Reference. Now publish. One of the co-author should be aware of that...

We cited correctly Mölg et al. (2018) here that was published in 2018 (https://doi.org/10.5194/essd-10-1807-2018, 2018).

6.33. suggested change to "The number of debris-covered glaciers, i.e. with an areal percentage of debris larger than 10%,"

Accepted. We changed this sentence as it was suggested:

"The number of debris-covered glaciers (i.e. with an areal percentage of debris larger than 10% - see methods) also increased from 122 in 1986, to 143 in 2000, and to 172 in 2014."

Please see P6 L22-24

8.16. This is a bit confusing because the role of aspect as already been discussed. Make one paragraph to compare the different part of Greater Caucasus (east to west) and one paragraph for north/south differences.

Accepted. We improved these paragraphs accordingly:

"For all regions investigated in the Greater Caucasus the rate of increase in supraglacial debris-cover varied between northern and southern slopes, and between the western, central and eastern sections of the mountain range (Table 2; Fig. 3, 4). The western Greater Caucasus experienced a supraglacial debris-cover area increase from 4.1±6.8% in 1986, to 22.4±6.4% in 2014, while the increase was significantly lower in the central Greater Caucasus over the last 30 years - from 7.4±6.3% to 10.1±6.2% (Table 2). The eastern Greater Caucasus with fewer glaciers almost doubled in supraglacial debris cover over the last 30 years from 27.9±6.2% to 49.2±5.7%.

Supraglacial debris-covered area increased from 13.5±6.3% to 29.3±6.0% for all three sections (western, central and eastern) on the northern slope (not including Elbrus), and from 4.0±6.8% to 9.2±6.9% on the southern slope of the Greater Caucasus between 1986 and 2014."

Please see P6 L25-33

8.23 "in the 3000-3500 m zone"

Done.

Please see P6 L40

Figure 4. Clarify what are panels a, b and c. I suggest adding directly the time span on each panel to make the reading easier.

Accepted. Time spams have been added on each panel. We also clarified all three panels (a, b, c) in the legend.

Please see P9 L4-7

Figure 5. The figure gives hard work to the readers in part because the legend is not detailed enough. Is it only for Elbrus for all panels? (probably not but this is confusing). Are not % of SDC sufficient for panels b and C. The figure need to be re-thought.

We provided new figures for all three panels (a, b, c) in the Figure 5 along with the more appropriate legend.

At the panel "a" it is shown just percentage of supraglacial debris-cover for the Elbrus slopes in 1986 and 2014.

Panel "b" and "c" shows total glacier area (km$^2$) and supraglacial debris cover area (km$^2$) changes between 1986 and 2014 for the Elbrus slopes.

Please see P10 L1-3

Figure 6. Again a lot of information in this figure so that the reader feels a bit lost. Cannot you find a way to simplify it to really make your point?

We slightly modified the Figure 6, but overall we think that basic changes are not necessary.

Please see new Figure 6. P10 L6-10

11.3. superscript should be -1 not +1 here. Can the authors quote their result for this particular glacier (even if the period is slightly different)? Otherwise the reader cannot judge whether the two studies agree. And this is true for all comparisons to previous studies (by Popovnin, Lambrecht). Overall, the manuscript needs to be improved on this aspect.

Accepted, we corrected all superscripts in the manuscript.

We also made additional comparison for some individual glaciers (e.g. Shkhelda, Djankuat) in order to better compare with previous studies. Overall, the new sentences stands as the:

"On individual glaciers, supraglacial debris-cover increased by 25%-30% (e.g. Shkhelda) in the same period. We found an increase of supraglacial debris-cover from 21.3±6.0% to 30±5.8% (~0.65% yr$^{-1}$) for Shkhelda Glacier between 1986 and 2000."

Please see P8 L22-24

"Popovnin et al. (2015), reported a supraglacial debris-cover increase from 2% to 13% (~0.23% yr$^{-1}$) between 1968-2010, based on direct field monitoring for the Djankuat Glacier (northern slope of the central Greater Caucasus). This compares with our result of an area increase of supraglacial debris-cover for Djankuat Glacier from 2.6±6.9% to 9.8±6.1% (~0.25% yr$^{-1}$) between 1986 and 2014. This difference can be explained in that the detailed field measurement would have picked out smaller spots of debris-cover which were beyond the resolution of the satellite imagery."

Please see P8 L25-30

11.17. For the glacier with outlines in RGI v6 (i.e. not nominal glaciers), it would be nice to have a 1 to 1 comparison of the % of debris cover from this study and Scherler et al. Do Scherler et al. overestimate the % of debris cover for all glaciers? What is the mean overestimation?, its standard deviation, etc...? A more thorough comparison with quantitative statistics would be welcome.

We provided new text here, in order to better understand the clear differences for exactly the same glaciers (not nominal glaciers):

"We found that a large portion of these glaciers in the Greater Caucasus are covered by supraglacial debris-cover. While for two regions the results match within the uncertainty (western Greater Caucasus: 22.4±6.4% vs. 23.7% and Elbrus 4.6±6.6% vs. 7.7%), our values are lower than the results of Scherler, et al. (2018) for the other two regions (10.1±6.2% vs. 30.8% for central, 49.2±5.7% vs. 84.9% for eastern) (Fig. S3)."

Please see P10 L13-16 and P 11 L1

In addition, there are Figure S3 and Figure S4a in the supplement, nicely showing these differences.

11.29. Can the authors quantify the mean up-glacier migration of the upper limit of the debris cover? It should be straightforward and it would be a useful numbers that future studies could quote and compare to.

Special thanks for this comment.

We calculated the mean up-glacier migration of the upper limit of the debris cover for all selected regions and provided new text, figure and table.

Please see the text: P8 L12-16

"The mean upper limit of the supraglacial debris-cover migrated from ~3015 m to ~3130 m between 1986 and 2014 for all selected glaciers of the Greater Caucasus. The highest mean upper limit lies on the northern slopes of eastern Greater Caucasus (~3626 m), while the lowest limit is on the southern slopes of western Greater Caucasus

(~2850 m). On the Elbrus Massif the mean upper limit of the supraglacial debris-cover migrated from ~3345 m to ~3520 m between 1986 and 2014 (Fig. 7, Table S1)"

Please see the new figure (Figure 7). P11 L11-13
And new Table in the supplement (Table S1).

11.33. Change "decreased by" to "changed by". If you keep "decreased" then change sign from "-" to "+"
We changed this sentence as follows:
"The results presented in this study indicate that the clean-ice area for all selected glaciers decreased from about 93% to 87% between 1986 and 2014 (Table 2)"
Please see P11 L22-23

11.35-38. The Popovnin results were already compare to. They agree but do not "support" the interpretation. No need to repeat the agreement here.
We shortened this sentence as follows:
"This finding is also consistent with field measurements on Djankuat Glacier (Popovnin et al., 2015)."
Please see P11 L24-25

12.1-4. Please be more specific in the text. Was there some rock avalanches on Suatisi and Devdoraki glaciers? Figure S5 indeed suggest so but it would be good to state it directly in the text with the date (or possible range of dates) of the rock avalanches.
Accepted, we changed this sentence as follows:
"Rock avalanches after 2000 on some glaciers in the Greater Caucasus (particularly in the eastern section) might be one of the reasons why the increase rate was higher during our second time period (2000-2014). For example, a rock–ice avalanche onto the Devdoraki Glacier on 17 May 2014 (Tielidze et al., 2019) caused an area increase of supraglacial debris-cover from 5.9±6.0% to 19.1±5.6% or about 0.95% yr$^{-1}$ and a landslide after 2000 onto the Suatisi Glacier produced supraglacial debris-cover area increase from 2.1±6.1% to 17.6±5.7% or about 1.10% yr$^{-1}$ (Fig. S6)."
Please see P11 L27-32

13.13. "although smaller glaciers exist as well in high cirques". Is it in the North? A bit ambiguous.
We clarified this sentence as follows:
"Our investigation shows also that the supraglacial debris-cover increases more quickly on the northern slopes of the Greater Caucasus than on the southern, where higher solar radiation input commonly results in smaller glaciers than on the northern slopes. Furthermore, smaller glaciers on the southern slope exist in high cirques with much steeper surface."
Please see P12 L5-8

13.16. The sentence "But there are some exceptions, where the northern-facing glaciers are shorter and steeper, and here, the glaciers of the southern slope are characterized with relatively more supraglacial debris cover" is confusing again. I do not follow the reasoning.
We improved this sentence as follows:
"Glaciers on the northern slopes are on average less steep than on the south mainly because most valley glacier tongues in the north are longer and reach lower altitudes than the south-facing glaciers. This conclusion is supported by Lambrecht et al. (2011) who observed a more rapid increase of supraglacial debris-cover on the northern slopes than the southern."
Please see P12 L8-11

13.9. The discussion is about increase in debris cover and here "rock glaciers" are mentioned. Again I do not follow the reasoning.

We deleted this sentence since it was not relevant here.

13.14. Because the lithology of the eastern and central Caucasus are described it should also be the case for the western Caucasus (and how it influences rock avalanches), otherwise the demonstration is not finished.

Accepted. We provided an appropriate text here partly explaining possible reasons (including lithology) for increasing supraglacial debris-cover:

"This section is characterised by the highest glacier retreat after the eastern Greater Caucasus. It is therefore possible that glaciers are also rapidly thinning favouring debris-covered over the ablation area (Benn et al., 2012; Pratap et al., 2015). The dome of the anticlinorium of the western Greater Caucasus (crest of the main water divide) is built on Proterozoic and Paleozoic plagiogneisses, granites, amphibolites, and crystalline slates. This provides the framework for overall denudation of the high mountainous relief (over 3000 m) (Gobejishvili et al., 2019). Furthermore, this area is characterized by active tectonic and ongoing mountain building (uplifting) processes (Tsereteli et al., 2016), which might be a further reason for increasing supraglacial debris-cover. We note that all these reasons need confirmation by detailed field measurements and could be part of a separate investigation since there is no accurate geographical pattern which otherwise explains the clear differences of the increase in supraglacial debris cover."

Please see P13 L3-12

13.37. The end of the sentence "as debris-covered glaciers may not be as sensitive to climate change as debris-free glaciers" is a general statement that has not much to do here and is vaguely connected to the rest of the sentence.

We corrected this sentence and now stands as a:

"Direct field measurements show that thermal resistance of the < 20 cm supraglacial debris-cover for some glaciers (e.g. Djankuat and Zpkhito) in the Greater Caucasus is relatively higher (0.07-0.15°C and 0.05-0.08°C m²/W) than in other glacierised regions of the world (e.g. Baltoro, Karakoram 0.02-0.07°C and Maliy Aktru, Altay 0.02-0.09°C m²/W) (Lambrecht et al., 2011), preventing a more rapid retreat."

Please see P14 L15-21

Figure 8. I must say that I find the figure quite confusing. Showing SDC only for the two dates would be best. I do not understand why the authors show clean ice if they do not really discuss it in the text.

Showing the clean ice area, helps reader to recognise full margins of the glaciers.

e.g. There was no SDC in 1986 in this area (which could be a red color) (on the image below), but the blue color help us to recognise how did the full glacier margin look like in 1986.

[Figure]

We also discuss about the clean ice area decrease in the text: "The results presented in this study indicate that the clean-ice area for all selected glaciers decreased from about 93% to 87% between 1986 and 2014 (Table 2). This reduction was caused by both glacier retreat and an increase in total supraglacial debris-cover" Please see P11 L22-24

We also agree that it is sometimes a bit confusing to show several layers in one image. That's why we created "gif amination", which is more clearly showing the basic principle of this image.
Accordingly, we decided to leave the Figure 8 without any changes.

Nicely written conclusion. However, I would not provide the name of specific instrument here (Kipp & Zonen or pulseEKKO. I do not thing this is the right place.
Done, suggested sentence has been changed accordingly:
"Future work should focus on using high resolution aerial/satellite imagery and more detailed field measurements, e.g. glaciological mass balance measurements, characteristics of debris and debris thickness measurement by Ground-Penetrating Radar, or incoming and reflected solar radiation, long-wave terrestrial and returned radiation. This will reduce uncertainties connected with supraglacial debris-cover assessment and glacier mapping accuracy in this region."
Please see P15 L3-7

15.13. "Supraglacial debris-cover changes in the Greater Caucasus" should be replaced by "the supplement"
Done, suggested sentence has been changed accordingly
Please see P15 L16

**Additional comments**
Rates presented as -XY% yr^(-1), as well as +XY% yr^(+1) (see for example P1, L31). It appears to me that the main author confuses the unit and uses % yr each time when there is an increase. This is also seen in Figure 4. I also think the wording is often somewhat weird, but this is not too big an issue."
Done, we corrected all rates and superscripts through the manuscript.

I will ask you to check very carefully the units throughout the manuscript. It is of course of paramount importance to have correct units.
Accepted, all units have been checked and corrected.

[revised manuscript text omitted]